# Chemoreceptor co-expression in *Drosophila melanogaster* olfactory neurons

**Darya Task[1†], Chun-Chieh Lin[1,2‡], Alina Vulpe[3§], Ali Afify[1], Sydney Ballou[3], Maria Brbic[4], Philipp Schlegel[5], Joshua Raji[1], Gregory SXE Jefferis[5,6], Hongjie Li[7#], Karen Menuz[3], Christopher J Potter[1]\***

[1]The Solomon H. Snyder Department of Neuroscience, Center for Sensory Biology, Johns Hopkins University School of Medicine, Baltimore, United States; [2]Mortimer B. Zuckermann Mind Brain Behavior Institute, Columbia University, New York, United States; [3]Physiology & Neurobiology Department, University of Connecticut, Mansfield, United States; [4]Department of Computer Science, Stanford University, Stanford, United States; [5]Drosophila Connectomics Group, Department of Zoology, University of Cambridge, Cambridge, United Kingdom; [6]Neurobiology Division, MRC Laboratory of Molecular Biology, Cambridge, United Kingdom; [7]Department of Biology, Howard Hughes Medical Institute, Stanford University, Stanford, United States

**\*For correspondence:**
cpotter@jhmi.edu

**Present address:** [†]Department of Biology, Johns Hopkins University, Baltimore, United States; [‡]The Department of Pathology and Laboratory Medicine, Geisel School of Medicine, Dartmouth-Hitchcock Medical Center, Lebanon, United States; [§]Department of Neuroscience, Yale University, New Haven, United States; [#]Huffington Center on Aging, Department of Molecular and Human Genetics, Baylor College of Medicine, Houston, United States

**Competing interest:** The authors declare that no competing interests exist.

**Abstract** *Drosophila melanogaster* olfactory neurons have long been thought to express only one chemosensory receptor gene family. There are two main olfactory receptor gene families in *Drosophila*, the odorant receptors (ORs) and the ionotropic receptors (IRs). The dozens of odorant-binding receptors in each family require at least one co-receptor gene in order to function: *Orco* for ORs, and *Ir25a*, *Ir8a*, and *Ir76b* for IRs. Using a new genetic knock-in strategy, we targeted the four co-receptors representing the main chemosensory families in *D. melanogaster* (*Orco*, *Ir8a*, *Ir76b*, *Ir25a*). Co-receptor knock-in expression patterns were verified as accurate representations of endogenous expression. We find extensive overlap in expression among the different co-receptors. As defined by innervation into antennal lobe glomeruli, *Ir25a* is broadly expressed in 88% of all olfactory sensory neuron classes and is co-expressed in 82% of Orco+ neuron classes, including all neuron classes in the maxillary palp. *Orco*, *Ir8a*, and *Ir76b* expression patterns are also more expansive than previously assumed. Single sensillum recordings from Orco-expressing *Ir25a* mutant antennal and palpal neurons identify changes in olfactory responses. We also find co-expression of *Orco* and *Ir25a* in *Drosophila sechellia* and *Anopheles coluzzii* olfactory neurons. These results suggest that co-expression of chemosensory receptors is common in insect olfactory neurons. Together, our data present the first comprehensive map of chemosensory co-receptor expression and reveal their unexpected widespread co-expression in the fly olfactory system.

## Editor's evaluation

A combination of methods, including a new method for tagging genes, demonstrates that the chemosensory co-receptors of *Drosophila melanogaster* (Orco, IR8a, IR25a, IR76b) are expressed widely and highly overlapping. These findings challenge a long-standing dogma in the field and suggest that different types of receptors, that is, olfactory and ionotropic receptors, can be co-expressed in the same chemosensory neuron. Moreover, optogenetics and single sensillum recordings provide evidence that IR25a co-receptor might modulate the activity of typical Orco-dependent

olfactory sensory neurons. The authors also provide evidence that this co-expression is conserved by examining two other fly species.

## Introduction

The sense of smell is crucial for many animal behaviors, from conspecific recognition and mate choice (*Dweck et al., 2015*; *Stengl, 2010*), to location of a food source (*Auer et al., 2020*; *Hansson and Stensmyr, 2011*), to avoidance of predators (*Ebrahim et al., 2015*; *Kondoh et al., 2016*; *Papes et al., 2010*) and environmental dangers (*Mansourian et al., 2016*; *Stensmyr et al., 2012*). Peripheral sensory organs detect odors in the environment using a variety of chemosensory receptors (*Carey and Carlson, 2011*; *Su et al., 2009*). The molecular repertoire of chemosensory receptors expressed by the animal, and the particular receptor expressed by any individual olfactory neuron, define the rules by which an animal interfaces with its odor environment. Investigating this initial step in odor detection is critical to understanding how odor signals first enter the brain to guide behaviors.

The olfactory system of the vinegar fly, *Drosophila melanogaster*, is one of the most extensively studied and well understood (*Depetris-Chauvin et al., 2015*). *D. melanogaster* is an attractive model for studying olfaction due to its genetic tractability, numerically simpler nervous system (compared to mammals), complex olfactory-driven behaviors, and similar organizational principles to vertebrate olfactory systems (*Ache and Young, 2005*; *Wilson, 2013*). Over 60 years of research have elucidated many of the anatomical, molecular, and genetic principles underpinning fly olfactory behaviors (*Gomez-Diaz et al., 2018*; *Harris, 1972*; *Pask and Ray, 2016*; *Siddiqi, 1987*; *Stocker, 2001*; *Venkatesh and Naresh Singh, 1984*; *Vosshall and Stocker, 2007*; *Yan et al., 2020*). Recent advances in electron microscopy and connectomics are revealing higher brain circuits involved in the processing of olfactory information (*Bates et al., 2020*; *Berck et al., 2016*; *Frechter et al., 2019*; *Horne et al., 2018*; *Marin et al., 2020*; *Zheng et al., 2018*); such endeavors will aid the full mapping of neuronal circuits from sensory inputs to behavioral outputs.

The fly uses two olfactory appendages to detect odorants: the antennae and maxillary palps (*Figure 1A*; *Stocker, 1994*). Each of these is covered by sensory hairs called sensilla, and each sensillum houses between one and four olfactory sensory neurons (OSNs) (*Figure 1B*; *de Bruyne et al., 2001*; *Venkatesh and Naresh Singh, 1984*). The dendrites of these neurons are found within the sensillar lymph, and they express chemosensory receptors from three gene families: *odorant receptors* (*ORs*), *ionotropic receptors* (*IRs*), and *gustatory receptors* (*GRs*) (*Figure 1C*, left; *Benton et al., 2009*; *Clyne et al., 1999*; *Gao and Chess, 1999*; *Jones et al., 2007*; *Kwon et al., 2007*; *Vosshall et al., 1999*; *Vosshall et al., 2000*). These receptors bind odorant molecules that enter the sensilla from the environment, leading to the activation of the OSNs, which then send this olfactory information to the fly brain (*Figure 1D*), to the first olfactory processing center – the antennal lobes (ALs) (*Figure 1E*; reviewed in *Depetris-Chauvin et al., 2015*; *Gomez-Diaz et al., 2018*; *Pask and Ray, 2016*). The standard view regarding the organization of the olfactory system in *D. melanogaster* is that olfactory neurons express receptors from only one of the chemosensory gene families (either *ORs*, *IRs*, or *GRs*), and all neurons expressing the same receptor (which can be considered an OSN class) project their axons to one specific region in the AL called a glomerulus (*Figure 1C*, right; *Couto et al., 2005*; *Fishilevich and Vosshall, 2005*; *Gao et al., 2000*; *Laissue et al., 1999*; *Pinto et al., 1988*; *Vosshall et al., 2000*). This pattern of projections creates a map in which the OR+ (*Figure 1E*, teal), IR+ (*Figure 1E*, purple), and GR+ (*Figure 1E*, dark blue) domains are segregated from each other in the AL. The OR+ domains innervate 38 anterior glomeruli, while the IR+ (19 glomeruli) and GR+ (1 glomerulus) domains occupy more posterior portions of the AL. One exception is the Or35a+ OSN class, which expresses an *IR* (*Ir76b*) in addition to the *OR* and *Orco*, and innervates the VC3 glomerulus (*Figure 1E*, striped glomerulus; *Benton et al., 2009*; *Couto et al., 2005*; *Fishilevich and Vosshall, 2005*). Different OSN classes send their information to different glomeruli, and the specific combination of OSN classes and glomeruli that are activated by a given smell (usually a blend of different odorants) constitutes an olfactory 'code' that the fly brain translates into an appropriate behavior (*Grabe and Sachse, 2018*; *Haverkamp et al., 2018*; *Seki et al., 2017*).

The receptors within each chemosensory gene family form heteromeric ion channels (receptor complexes) (*Abuin et al., 2011*; *Butterwick et al., 2018*; *Sato et al., 2008*). The *ORs* require a single co-receptor, *Orco*, to function (*Figure 1C*, middle row; *Benton et al., 2006*; *Larsson et al., 2004*;

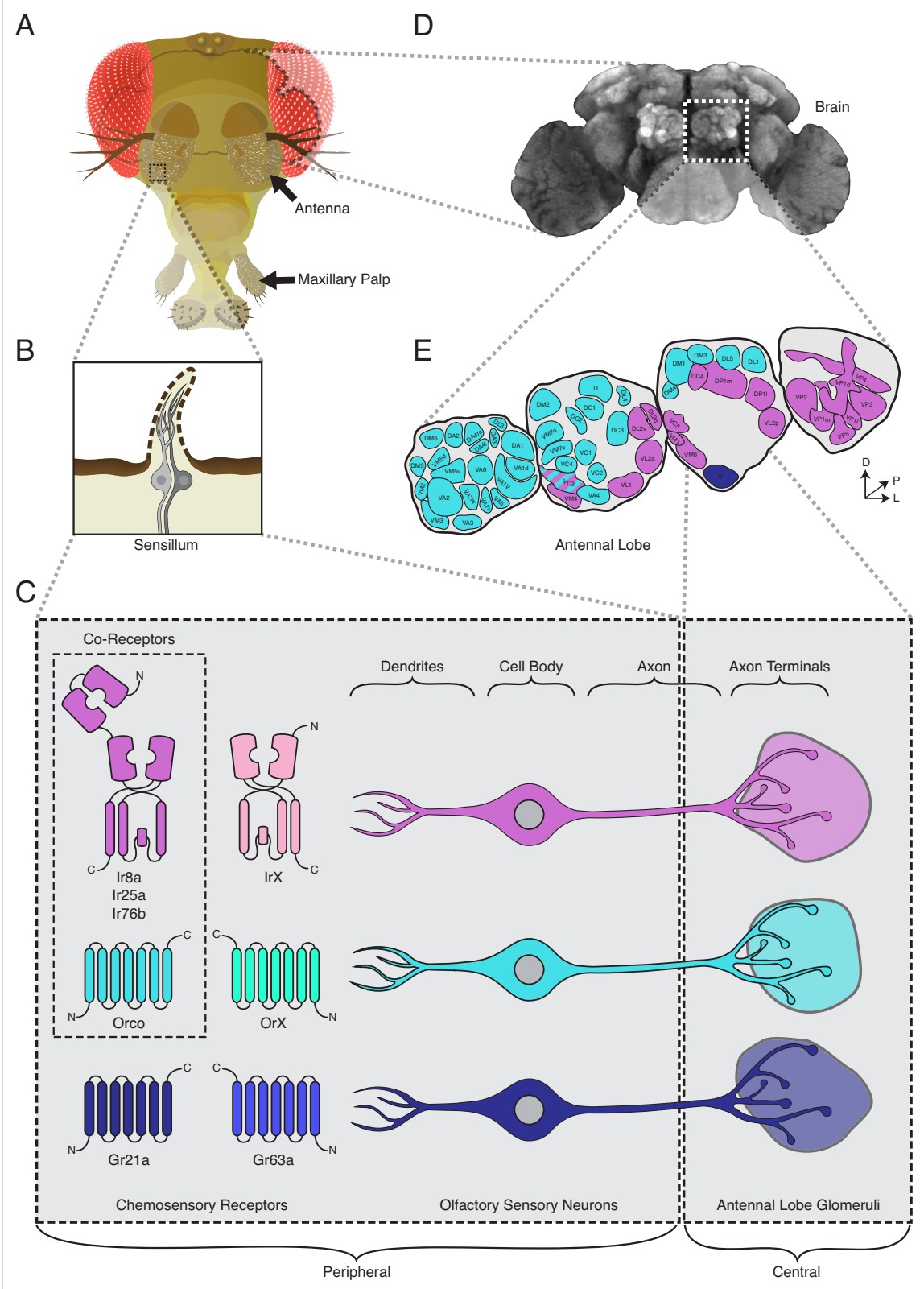

**Figure 1.** The standard view of olfactory receptor expression in *Drosophila melanogaster*. (**A**) The adult fly head (left) has two olfactory organs: the antennae and the maxillary palps (arrows). Olfactory neurons from these organs project to the fly brain (**D**), to the first center involved in processing of olfactory information, the antennal lobes (**E**). (**B**) The olfactory organs are covered by sensory hairs called sensilla (left). Each sensillum contains between one and four olfactory sensory neurons (two example neurons are shown in gray). The dendrites of these neurons extend into the sensilla, and the axons

*Figure 1 continued on next page*

**Figure 1 continued**

target discrete regions of the antennal lobes called glomeruli (**E**). Neuronal compartments (dendrites, cell body, axon, axon terminals) are labeled in (**C**). (**C**) Left: in the periphery, each olfactory sensory neuron is traditionally thought to express chemosensory receptors from only one of three gene families on its dendrites: ionotropic receptors (IRs, pink and purple), odorant receptors (ORs, teal and green), or gustatory receptors (GRs, light and dark blue). IRs and ORs require obligate co-receptors (dotted box outline) to form functional ion channels. All ORs utilize a single co-receptor, Orco (teal), while IRs can utilize one (or a combination) of three possible co-receptors (purple): Ir8a, Ir25a, or Ir76b. The two GRs form a functional carbon dioxide detecting channel expressed in only one class of neurons. All other olfactory neurons express one of the four co-receptors. Right: olfactory sensory neurons expressing ORs, IRs, and GRs are thought to project to mutually exclusive glomeruli in the antennal lobe (AL) of the central brain, forming the olfactory map shown in (**E**). (**D**) Fly brain stained with anti-brp synaptic marker (nc82), with left AL outlined by the dotted white box. (**E**) AL map with glomeruli color-coded by the chemosensory receptors (ORs, IRs, or GRs) expressed in the olfactory sensory neurons projecting to them. Only one glomerulus (VC3, striped) receives inputs from neurons expressing chemoreceptors from multiple gene families (ORs and IRs). Compass: D = dorsal, L = lateral, P = posterior.

*Vosshall and Hansson, 2011*). The ligand-binding *OrX* confers odorant specificity upon the receptor complex, while the co-receptor *Orco* is necessary for trafficking of the *OrX* to the dendritic membrane and formation of a functional ion channel (*Benton et al., 2006*; *Larsson et al., 2004*). Likewise, the ligand-binding *IrXs* require one or more *IR* co-receptors: *Ir8a*, *Ir25a*, and/or *Ir76b* (*Figure 1C*, top row). The *IR* co-receptors (IrCos) are similarly required for trafficking and ion channel function (*Abuin et al., 2011*; *Abuin et al., 2019*; *Ai et al., 2013*; *Vulpe and Menuz, 2021*). The *GR* gene family generally encodes receptors involved in taste, which are typically expressed outside the olfactory system (such as in the labella or the legs) (*Dunipace et al., 2001*; *Park and Kwon, 2011*; *Scott, 2018*; *Scott et al., 2001*); however, *Gr21a* and *Gr63a* are expressed in one antennal OSN neuron class and form a complex sensitive to carbon dioxide (*Figure 1C*, bottom row; *Jones et al., 2007*; *Kwon et al., 2007*).

The majority of receptors have been mapped to their corresponding OSNs, sensilla, and glomeruli in the fly brain (*Bhalerao et al., 2003*; *Couto et al., 2005*; *Fishilevich and Vosshall, 2005*; *Frank et al., 2017*; *Grabe et al., 2016*; *Hallem and Carlson, 2006*; *Hallem et al., 2004*; *Knecht et al., 2017*; *Marin et al., 2020*; *Ray et al., 2008*; *Silbering et al., 2011*). This detailed map has allowed for exquisite investigations into the developmental, molecular, electrophysiological, and circuit/computational bases of olfactory neurobiology. This work has relied on transgenic lines to identify and manipulate OSN classes (*Ai et al., 2013*; *Brand and Perrimon, 1993*; *Couto et al., 2005*; *Fishilevich and Vosshall, 2005*; *Kwon et al., 2007*; *Lai and Lee, 2006*; *Larsson et al., 2004*; *Menuz et al., 2014*; *Potter et al., 2010*; *Silbering et al., 2011*). These transgenic lines use regions of DNA upstream of the chemosensory genes that are assumed to reflect the enhancers and promoters driving expression of these genes. While a powerful tool, transgenic lines may not contain all of the necessary regulatory elements to faithfully recapitulate the expression patterns of the endogenous genes. In addition, the genomic insertional location of the transgene might affect expression patterns (positional effects). Some transgenic lines label a subset of the cells of a given olfactory class, while others label additional cells: for example, the transgenic *Ir25a-Gal4* line is known to label only a portion of cells expressing Ir25a protein (as revealed by antibody staining) (*Abuin et al., 2011*); conversely, *Or67d-Gal4* transgenes incorrectly label two glomeruli, whereas a *Gal4* knock-in at the *Or67d* genetic locus labels a single glomerulus (*Couto et al., 2005*; *Fishilevich and Vosshall, 2005*; *Kurtovic et al., 2007*). While knock-ins provide a faithful method to capture a gene's expression pattern, generating these lines has traditionally been cumbersome.

In this paper, we implement an efficient knock-in strategy to target the four main chemosensory co-receptor genes in *D. melanogaster* (Orco, Ir8a, Ir76b, Ir25a). We find broad co-expression of these co-receptor genes in various combinations in olfactory neurons, challenging the current view of segregated olfactory families in the fly. In particular, *Ir25a* is expressed in the majority of olfactory neurons, including most Orco+ OSNs. In addition, the *Ir8a* and *Ir25a* knock-in lines help to distinguish two new OSN classes in the sacculus that target previously unidentified glomerular subdivisions in the posterior AL. Recordings in *Ir25a* mutant sensilla in Orco+ neurons reveal subtle changes in odor responses, suggesting that multiple chemoreceptor gene families could be involved in the signaling or development of a given OSN class. We further extend our findings of co-receptor co-expression to two additional insect species, *Drosophila sechellia* and *Anopheles coluzzii*. These data invite a re-examination of odor coding in *D. melanogaster* and other insects. We present a comprehensive model of co-receptor expression in *D. melanogaster*, which will inform future investigations of combinatorial chemosensory processing.

**Table 1.** Summary of HACK knock-in efficiency (related to *Figure 2*).

There are two ways to generate knock-ins via the HACK technique: by direct injection or by genetic cross (see *Figure 2—figure supplement 1* and Materials and methods for details) (*Lin and Potter, 2016a*). All four co-receptor genes were targeted using the direct injection approach; additionally, the crossing approach was tested with *Orco* and *Ir25a*. Knock-in efficiency, as measured by the number of flies having the mCherry+ marker divided by the total number of potentially HACKed flies, was high for all genes tested and both approaches. Efficiency appears to depend on the genetic locus, as has been previously demonstrated (*Lin and Potter, 2016b*). To further estimate the effort required to generate a HACK knock-in, we calculated the percentage of founder flies producing knock-in lines; this gives an indication of the number of independent crosses needed to successfully create a knock-in line. Two to five individual $G_0$ starting crosses were sufficient to produce a knock-in. For each gene, a sample of individual knock-in lines was tested via PCR genotyping, sequencing, and by crossing to a reporter line to confirm brain expression (knock-ins sampled). For all knock-ins generated via the direct injection method, every fly tested represented a correctly targeted knock-in. However, for the cross method, some lines had the mCherry+ marker and yet did not drive GFP expression in the brain when crossed to a reporter line (labeled here as false positives). See also *Table 1—source data 1* and *Figure 2—figure supplement 1*.

| Gene | Approach | mCherry+ | mCherry- | Total | Efficiency (%) | Founders producing knock-in (#/total) | Knock-ins sampled | False positives | Confirmed | Correct (%) |
|------|----------|----------|----------|-------|----------------|---------------------------------------|-------------------|-----------------|-----------|-------------|
| *Orco* | Direct injection | 180 | 365 | 545 | 33 | 43% (3/7) | 30 | 0 | 30 | 100 |
| *Ir8a* | Direct injection | 53 | 609 | 662 | 8 | 20% (4/20) | 5 | 0 | 5 | 100 |
| *Ir76b* | Direct injection | 79 | 184 | 263 | 30 | 100% (2/2) | 10 | 0 | 10 | 100 |
| *Ir25a* | Direct injection | 82 | 268 | 350 | 23 | 40% (2/5) | 6 | 0 | 6 | 100 |
| *Orco* | Cross | 37 | 96 | 133 | 28 | 100% (3/3) | 2 | 1 | 1 | 50 |
| *Ir25a* | Cross | 30 | 95 | 125 | 24 | 100% (2/2) | 30 | 5 | 25 | 83 |

The online version of this article includes the following source data for table 1:

**Source data 1.** HACK knock-in screen.

## Results

### Generation and validation of co-receptor knock-in lines

We previously developed the HACK technique for CRISPR/Cas9-mediated in vivo gene conversion of binary expression system components, such as the conversion of transgenic *Gal4* to *QF2* (*Brand and Perrimon, 1993*; *Jinek et al., 2012*; *Lin and Potter, 2016a*; *Lin and Potter, 2016b*; *Potter et al., 2010*; *Riabinina et al., 2015*; *Xie et al., 2018*). Here, we adapt this strategy for the efficient generation of targeted knock-ins (see *Table 1* and *Table 1—source data 1* for details). We chose to target the four chemosensory co-receptor genes to examine unmapped patterns of co-receptor expression in *D. melanogaster*. We inserted a *T2A-QF2* cassette and mCherry selection marker before the stop codon of the four genes of interest (*Figure 2A*, *Figure 2—figure supplement 1*). By introducing the T2A ribosomal skipping peptide, the knock-in will produce the full-length protein of the gene being targeted as well as a functional QF2 transcription factor (*Figure 2A*, protein products). This approach should capture the endogenous expression pattern of the gene under the control of the gene's native regulatory elements while retaining the gene's normal function (*Baena-Lopez et al., 2013*; *Bosch et al., 2020*; *Chen et al., 2020*; *Diao et al., 2015*; *Diao and White, 2012*; *Du et al., 2018*; *Gnerer et al., 2015*; *Gratz et al., 2014*; *Kanca et al., 2019*; *Lee et al., 2018*; *Li-Kroeger et al., 2018*; *Lin and Potter, 2016a*; *Vilain et al., 2014*; *Xue et al., 2014*). We found that *T2A-QF2* knock-ins were functional with some exceptions (see *Figure 2—figure supplement 2* and *Figure 2—source data 1*). For example, *Orco-T2A-QF2* knock-in physiology was normal, while a homozygous *Ir25a-T2A-QF2* knock-in exhibited a mutant phenotype. This suggests that the addition of the T2A peptide onto the C-terminus of Ir25a might interfere with its co-receptor function.

We examined the expression of the co-receptor knock-in lines in the adult olfactory organs by crossing each line to the same *10XQUAS-6XGFP* reporter (*Figure 2B–I*). *Orco-T2A-QF2*-driven GFP expression was detected in the adult antennae and maxillary palps (*Figure 2B*), as previously described (*Larsson et al., 2004*). We validated the *Orco-T2A-QF2* knock-in line with whole-mount antibody staining of maxillary palps (*Figure 2C*) and found a high degree of correspondence between anti-Orco antibody staining and knock-in driven GFP in palpal olfactory neurons (quantified in *Table 2*; see also *Figure 2—figure supplement 3A–D* for PCR and sequencing validation of all knock-in lines).

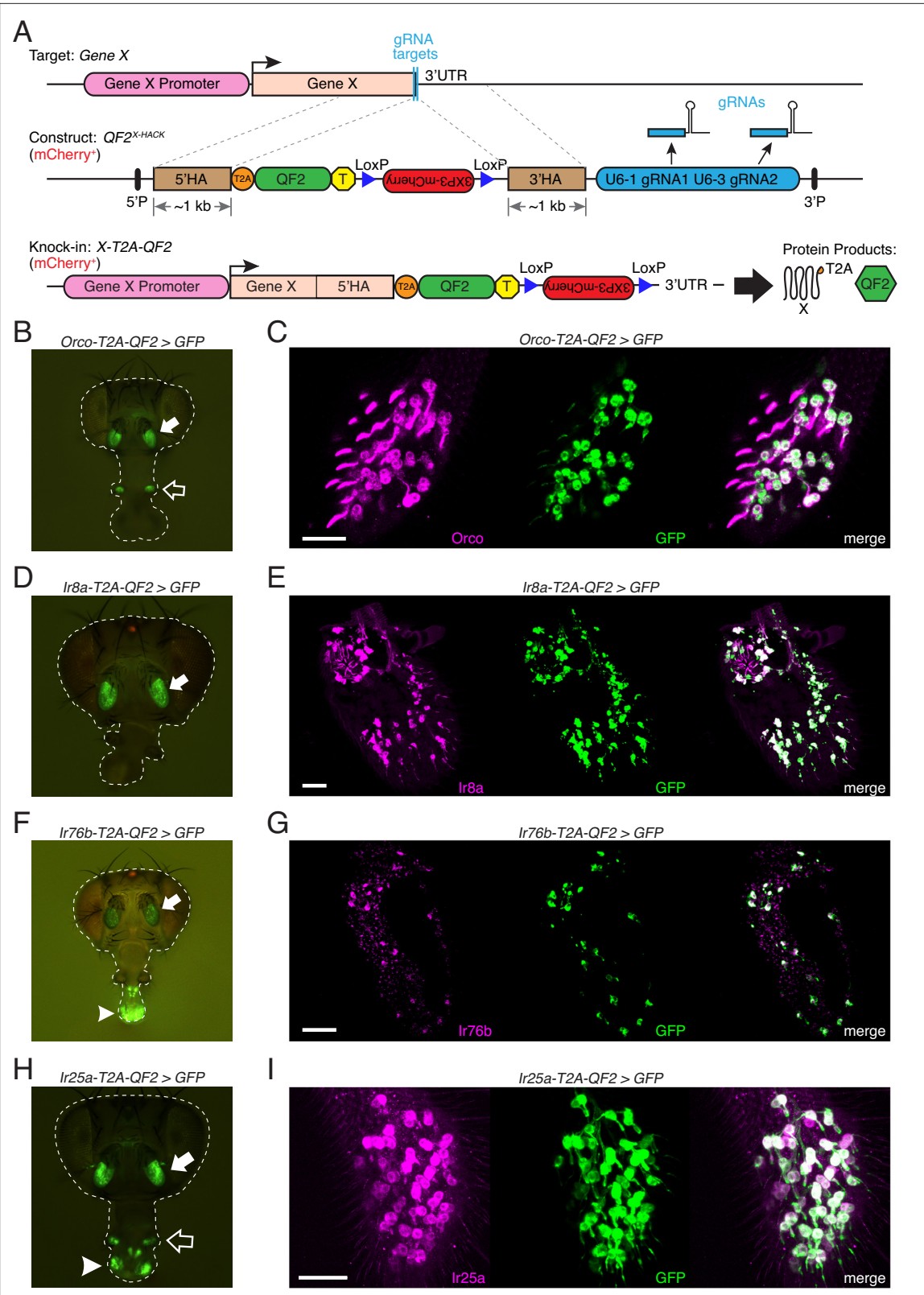

**Figure 2.** Generation and validation of chemosensory co-receptor knock-in lines. (**A**) Schematic of HACK knock-in approach. Top: two double-stranded breaks are induced on either side of the target gene stop codon with gRNAs (blue) expressed from the *QF2^X-HACK* construct (middle) in the presence of Cas9. The construct includes *T2A-QF2* and a floxed *3XP3-mCherry* marker. The knock-in introduces a transcriptional stop (yellow T) after QF2. Bottom: the knock-in produces two protein products (right) from the targeted mRNA: target X and the *QF2* transcription factor (***Diao and White, 2012***). The

*Figure 2 continued on next page*

*Figure 2 continued*

*X-T2A-QF2* knock-in can be crossed to a reporter (e.g., *QUAS-GFP*) to examine the endogenous expression pattern of the target gene. (**B**) *Orco-T2A-QF2* driving *QUAS-GFP* in adult fly head. GFP expression is found in the antennae (filled arrow) and maxillary palps (hollow arrow), as previously reported (*Larsson et al., 2004*). (**C**) Whole-mount anti-Orco antibody staining in *Orco-T2A-QF2>GFP* maxillary palps reveals a high degree of overlap of Orco+ and GFP+ cells. N = 3. (**D**) *Ir8a-T2A-QF2* drives GFP in the antennae (arrow), as previously reported (*Abuin et al., 2011*). (**E**) Anti-Ir8a antibody staining of *Ir8a-T2A-QF2>GFP* antennal cryosections shows high correspondence between Ir8a+ and GFP+ cells. N = 7. (**F**) *Ir76b-T2A-QF2* drives GFP expression in the antennae (filled arrow) and labella (hollow arrow), reflecting *Ir76b*'s role in olfaction and gustation, respectively (*Benton et al., 2009*; *Zhang et al., 2013*). (**G**) In situs on *Ir76b-T2A-QF2>GFP* antennal cryosections to validate that the knock-in faithfully recapitulates the endogenous expression pattern. N = 3. (**H**) *Ir25a-T2A-QF2* drives GFP in the antennae (filled arrow) and labella (hollow arrow), which has been reported previously (*Benton et al., 2009*; *Croset et al., 2010*). Expression in the maxillary palps (arrowhead) has not been previously reported. (**I**) Whole-mount maxillary palp staining with an anti-Ir25a antibody in *Ir25a-T2A-QF2>GFP* flies. The knock-in and Ir25a antibody co-labeled the majority of olfactory neurons in the palps. N = 5. Scale bars = 25 μm. In (**D**) and (**F**), the *3XP3-mCherry* knock-in marker can be weakly detected in the eyes and ocelli (red spot) of both *Ir8a-T2A-QF2* and *Ir76b-T2A-QF2*. See also *Figure 2—figure supplements 1–4*, *Tables 1 and 2*, *Table 1—source data 1*, *Figure 2—source data 1*, and Materials and methods.

The online version of this article includes the following source data and figure supplement(s) for figure 2:

**Source data 1.** Single sensillum recordings (SSRs) of knock-in lines.

**Figure supplement 1.** HACK crossing schematics, marker expression, and approach comparison.

**Figure supplement 2.** *T2A-QF2* HACK knock-in effects on target gene function.

**Figure supplement 3.** Additional validation of co-receptor knock-in lines.

**Figure supplement 4.** Knock-in expression in the larva.

We confirmed the specificity of the anti-Orco antibody by staining *Orco²* mutant palps and found no labeling of olfactory neurons (*Figure 2—figure supplement 3E*).

Unlike *Orco*, *Ir8a* expression has previously been localized only to the antenna, to olfactory neurons found in coeloconic sensilla and in the sacculus (*Abuin et al., 2011*). As expected, the knock-in line drove GFP expression only in the antenna (*Figure 2D*). To validate the *Ir8a-T2A-QF2* knock-in line, we performed antibody staining on antennal cryosections and found the majority of cells to be double labeled (*Figure 2E*, *Table 2*). There was no anti-Ir8a staining in control *Ir8a¹* mutant antennae (*Figure 2—figure supplement 3F*).

The *Ir76b* gene has previously been implicated in both olfaction and gustation and has been shown to be expressed in adult fly antennae, labella (mouthparts), legs, and wings (*Abuin et al., 2011*; *Chen and Amrein, 2017*; *Croset et al., 2010*; *Ganguly et al., 2017*; *Hussain et al., 2016*; *Sánchez-Alcañiz et al., 2018*; *Zhang et al., 2013*). We examined the *Ir76b-T2A-QF2* knock-in line and found a similar pattern of expression in the periphery, with GFP expression in the antennae and labella (*Figure 2F*). Because an anti-Ir76b antibody has not previously been tested in fly antennae, we performed in situs on *Ir76b-T2A-QF2>GFP* antennal cryosections to validate knock-in expression (*Figure 2G*) and confirmed the specificity of the probe in *Ir76b¹* mutant antennae (*Figure 2—figure supplement 3G*).

Of the four *D. melanogaster* co-receptor genes, *Ir25a* has been implicated in the broadest array of cellular and sensory functions, from olfaction (*Abuin et al., 2011*; *Benton et al., 2009*; *Silbering et al., 2011*) and gustation (*Chen and Amrein, 2017*; *Chen and Dahanukar, 2017*; *Jaeger et al., 2018*), to thermo- and hygro-sensation (*Budelli et al., 2019*; *Enjin et al., 2016*; *Knecht et al., 2017*; *Knecht et al., 2016*), to circadian rhythm modulation (*Chen et al., 2015*). In the adult olfactory system, *Ir25a* expression has previously been reported in three types of structures in the antenna: coeloconic sensilla, the arista, and the sacculus (*Abuin et al., 2011*; *Benton et al., 2009*). We examined the *Ir25a-T2A-QF2* knock-in line and found GFP expression in the adult antennae, labella, and maxillary palps (*Figure 2H*). This was surprising because no IR expression has previously been reported in fly palps. To verify Ir25a protein expression in the maxillary palps, we performed whole-mount anti-Ir25a antibody staining in *Ir25a-T2A-QF2>GFP* flies. We found broad *Ir25a* expression in palpal olfactory neurons (*Figure 2I*) and a high degree of overlap between knock-in driven GFP expression and antibody staining (*Table 2*). As expected, there was no anti-Ir25a staining in *Ir25a²* mutant palps (*Figure 2—figure supplement 3H*).

We also examined co-receptor knock-in expression in *D. melanogaster* larvae. As in the adult stage, larval GFP expression was broadest in the *Ir25a-T2A-QF2* and *Ir76b-T2A-QF2* knock-in lines, with GFP labeling of neurons in the head and throughout the body wall (*Figure 2—figure supplement 4*). The *Orco-T2A-QF2* knock-in line labeled only the olfactory dorsal organs in the larva, while

**Table 2.** Validation of *T2A-QF2* knock-in expression in the antennae and maxillary palps (related to *Figure 2*).
To verify that the knock-in lines recapitulate the endogenous expression patterns of the target genes, antennae or maxillary palps of flies containing the knock-ins driving GFP expression were co-stained with the corresponding antibody (Ab) (anti-Orco, anti-Ir8a, or anti-Ir25a). The overlap of Ab+ and GFP+ cells was examined, and a high correspondence between antibody staining and knock-in driven GFP was found. WM: whole-mount; cryo: cryosection. See also *Figure 2—figure supplement 3*.

| Knock-in | Sample | Antibody (Ab) | Ab+ cells | GFP+ cells | Double-labeled cells | Total cells |
|---|---|---|---|---|---|---|
| *Orco* | Palp 1 (WM) | Anti-Orco | 125 | 127 | 125 | 127 |
| *Orco* | Palp 2 (WM) | Anti-Orco | 112 | 111 | 108 | 115 |
| *Orco* | Palp 6 (WM) | Anti-Orco | 125 | 126 | 123 | 128 |
| | | Total across samples: | 362 | 364 | 356 | 370 |
| | | | Proportion of Ab+ cells that are GFP+: | Proportion of GFP+ cells that are Ab+: | Proportion of all cells that are double labeled: | |
| | | | 0.98 | 0.98 | 0.96 | |
| *Ir8a* | Antenna 1 (cryo) | Anti-Ir8a | 20 | 21 | 20 | 21 |
| *Ir8a* | Antenna 2 (cryo) | Anti-Ir8a | 24 | 24 | 24 | 24 |
| *Ir8a* | Antenna 6 (cryo) | Anti-Ir8a | 40 | 43 | 40 | 43 |
| *Ir8a* | Antenna 7 (cryo) | Anti-Ir8a | 12 | 13 | 12 | 13 |
| *Ir8a* | Antenna 8 (cryo) | Anti-Ir8a | 16 | 16 | 16 | 16 |
| *Ir8a* | Antenna 9 (cryo) | Anti-Ir8a | 42 | 42 | 41 | 43 |
| *Ir8a* | Antenna 10 (cryo) | Anti-Ir8a | 41 | 40 | 40 | 41 |
| | | Total across samples: | 195 | 199 | 193 | 201 |
| | | | Proportion of Ab+ cells that are GFP+: | Proportion of GFP+ cells that are Ab+: | Proportion of all cells that are double labeled: | |
| | | | 0.99 | 0.97 | 0.96 | |
| *Ir25a* | Palp 1 (WM) | Anti-Ir25a | 107 | 105 | 104 | 108 |
| *Ir25a* | Palp 2 (WM) | Anti-Ir25a | 86 | 85 | 85 | 86 |
| *Ir25a* | Palp 3 (WM) | Anti-Ir25a | 111 | 111 | 110 | 112 |
| *Ir25a* | Palp 4 (WM) | Anti-Ir25a | 94 | 94 | 94 | 94 |
| *Ir25a* | Palp 5 (WM) | Anti-Ir25a | 83 | 83 | 81 | 85 |
| | | Total across samples: | 481 | 478 | 474 | 485 |
| | | | Proportion of Ab+ cells that are GFP+: | Proportion of GFP+ cells that are Ab+: | Proportion of all cells that are double labeled: | |
| | | | 0.99 | 0.99 | 0.98 | |

the *Ir8a-T2A-QF2* knock-in line did not have obvious expression in the larval stage (*Figure 2—figure supplement 4*). All subsequent analyses focused on the adult olfactory system.

## Expanded expression of olfactory co-receptors

We next examined the innervation patterns of the four co-receptor knock-in lines in the adult central nervous system: the brain and ventral nerve cord (VNC) (*Figure 3*). Only two of the four lines (*Ir25a* and *Ir76b*) showed innervation in the VNC, consistent with the role of these genes in gustation in addition to olfaction (*Figure 3—figure supplement 1A*). In the brain, we compared the expression of each knock-in line (*Figure 3A–D*, green) to the corresponding transgenic *Gal4* line (*Figure 3A–D*, orange) to examine the differences in expression to what has previously been reported. Reporter-alone controls for these experiments are shown in *Figure 3—figure supplement 1B*. All four knock-in lines innervated the ALs, and the *Ir25a-T2A-QF2* and *Ir76b-T2A-QF2* lines additionally labeled the subesophageal zone (SEZ), corresponding to gustatory axons from the labella (*Figure 3C and D*, arrowheads; *Hussain et al., 2016*; *Zhang et al., 2013*). The co-labeling experiments revealed that all four knock-ins label more glomeruli than previously reported (see *Figure 3—source data 1* for AL

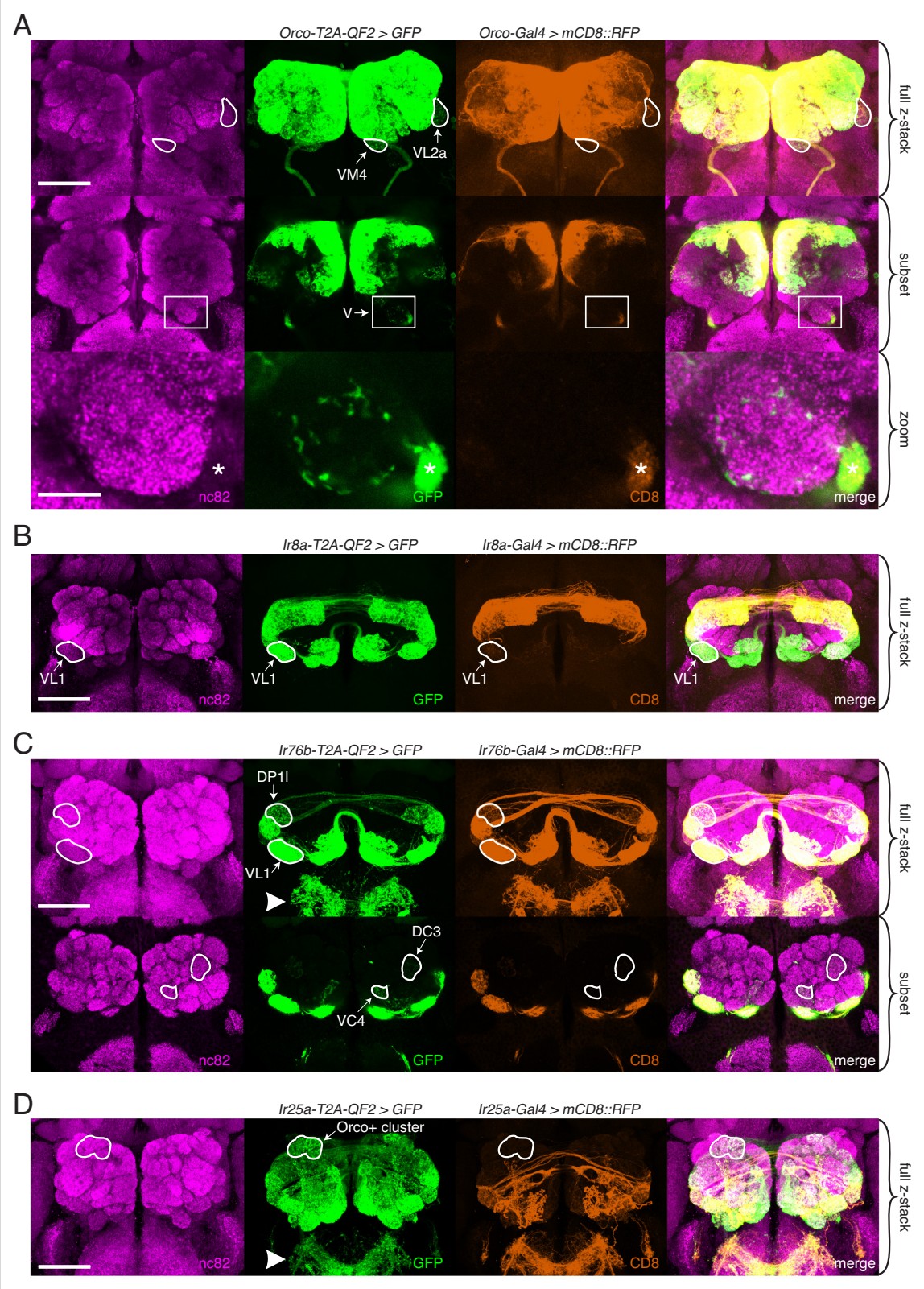

**Figure 3.** Expanded expression of olfactory co-receptors. (**A–D**) Comparing knock-in innervation patterns of the antennal lobe (AL) with what has previously been reported for each co-receptor. Co-labeling experiments with each co-receptor knock-in line driving *QUAS-GFP* (green) and the corresponding transgenic co-receptor *Gal4* line driving *UAS-mCD8::RFP* (anti-CD8, orange). The nc82 antibody labels synapses (magenta) and is used as a brain counterstain in these and all subsequent brain images. (**A**) The *Orco-T2A-QF2* knock-in labels more glomeruli than the *Orco-Gal4* line. Top:

*Figure 3 continued on next page*

*Figure 3 continued*

maximum intensity projection of full z-stack showing two additional glomeruli labeled by the knock-in, VM4 (Ir8a+/Ir76b+/Ir25a+) and VL2a (Ir8a+). Middle: subset of z-stack with a box around the V glomerulus. Bottom: zoom of boxed region showing sparse innervation of the V glomerulus (Gr21a+/ Gr63a+) by the knock-in but not the *Gal4* line. Asterisk indicates antennal nerve that is outside the V glomerulus. In the sub z-stack and zoom panel, gain has been increased in the GFP channel to visualize weak labeling more clearly. (**B**) The *Ir8a-T2A-QF2* knock-in also drives GFP expression in more glomeruli than previously reported, including the outlined VL1 glomerulus (Ir25a+). (**C**) In the brain, *Ir76b-T2A-QF2>GFP* olfactory neurons innervate the ALs, while gustatory neurons from the labella innervate the subesophageal zone (SEZ, arrowhead). Top: both the *Ir76b* knock-in and transgenic *Gal4* line label more glomeruli than previously reported, including VL1 (Ir25a+) and DP1l (Ir8a+). Bottom: the *Ir76b-T2A-QF2* knock-in labels several Orco+ glomeruli, such as DC3 and VC4 (outlined). In the subset, gain has been increased in the GFP channel to visualize weakly labeled glomeruli more clearly. (**D**) The *Ir25a-T2A-QF2* knock-in drives GFP expression broadly in the antennal lobes and SEZ (arrowhead). Ir25a+ neurons innervate many Orco+ glomeruli, such as those outlined. The transgenic *Ir25a-Gal4* line labels a subset of the knock-in expression pattern. N = 3–10 for co-labeling experiments, N = 5–15 for additional analyses of the knock-in lines alone. Scale bars = 25 μm, except zoom panel scale bar = 10 μm. See also *Figure 3—figure supplements 1 and 2*, *Table 3*, and *Figure 3—source data 1* and *Figure 3—source data 2*.

The online version of this article includes the following source data and figure supplement(s) for figure 3:

**Source data 1.** Knock-in antennal lobe analyses.

**Source data 2.** Examples of new glomerular expression in knock-in lines.

**Figure supplement 1.** Knock-in expression in the adult ventral nerve cord (VNC) and reporter expression in the brain.

**Figure supplement 2.** Transgenic co-receptor *Gal4* lines do not fully recapitulate knock-in expression.

analyses, *Figure 3—source data 2* for traced examples of newly identified glomeruli in each knock-in line, and *Table 3* for a summary of glomerular expression across all knock-in lines). Some glomeruli were not labeled consistently in all flies, which we define as variable expression (found in <50% of brains examined).

*Orco-T2A-QF2* labels seven 'non-canonical' glomeruli consistently, and one sporadically. These include VM4 and VL2a, which correspond to Ir76b+ and Ir8a+ OSN populations, respectively (*Figure 3A*, outlines). We also found that the *Orco* knock-in sparsely but consistently labels the V glomerulus, which is innervated by Gr21a+/Gr63a+ neurons (*Figure 3A*, box and zoom panel). *Orco-T2A-QF2* also labels one Ir25a+ glomerulus consistently (VL1), three additional Ir8a+ glomeruli consistently (DL2d, DL2v, DP1l), and one variably (DC4). Surprisingly, when we crossed the transgenic *Orco-Gal4* line (*Larsson et al., 2004*) to a stronger reporter (*Shearin et al., 2014*), we found that several of these additional glomeruli were weakly labeled by the transgenic line (*Figure 3—figure supplement 2A*). This suggests that there are OSN populations in which *Orco* is expressed either at low levels or in few cells, which might be why this expression was previously missed. We found this to be the case with the *IrCo* knock-ins, as well (described below).

There has been some inconsistency in the literature as to which glomeruli are innervated by Ir8a-expressing OSNs. For example, *Silbering et al., 2011* note that their *Ir8a-Gal4* line labels approximately 10 glomeruli, 6 of which are identified (DL2, DP1l, VL2a, VL2p, DP1m, DC4). An *Ir8a-Gal4* line generated by *Ai et al., 2013* also labels about 10 glomeruli, only 2 of which are identified (DC4 and DP1m) and which correspond to 2 glomeruli in *Silbering et al., 2011*. Finally, *Min et al., 2013* identify three additional glomeruli innervated by an *Ir8a-Gal4* line (VM1, VM4, and VC5) but not reported in the other two papers. DL2 was later subdivided into two glomeruli (*Prieto-Godino et al., 2017*), bringing the total number of identified Ir8a+ glomeruli to 10. However, we found that *Ir8a-T2A-QF2* consistently labels twice as many glomeruli as previously reported. These additional glomeruli include an Ir25a+ glomerulus (VL1, *Figure 3B*), numerous Orco+ glomeruli (such as VA3 and VA5), and an Orco+/Ir76b+ glomerulus (VC3) (see *Figure 3—source data 1* for a full list of new glomeruli and *Figure 3—source data 2* for outlined examples). Some of these additional glomeruli are weakly labeled by an *Ir8a-Gal4* line (*Figure 3—figure supplement 2B*), but this innervation is only apparent when examined with a strong reporter.

Of the four chemosensory co-receptor genes, the previously reported expression of *Ir76b* is the narrowest, with only four identified glomeruli (VM1, VM4, VC3, VC5) (*Silbering et al., 2011*). The *Ir76b-T2A-QF2* knock-in labels more than three times this number, including several Orco+ glomeruli (such as DC3 and VC4), most Ir8a+ glomeruli (including DP1l), and one additional Ir25a+ glomerulus (VL1) (*Figure 3C*). As with *Orco* and *Ir8a*, some but not all of these glomeruli can be identified by crossing the transgenic *Ir76b-Gal4* line to a strong reporter (*Figure 3—figure supplement 2C*). However, the *Ir76b-Gal4* line labels additional glomeruli not seen in the knock-in (*Figure 3—figure*

**Table 3.** Summary of expression patterns for all knock-in lines (related to *Figures 3–5*).

Summarized here are all of the olfactory sensory neuron (OSN) classes innervating the 58 antennal lobe glomeruli[†]; their corresponding sensilla and tuning receptors; the previously reported (original) co-receptors they express; and whether or not each of the co-receptor knock-in lines labels those glomeruli. Variable indicates that the glomerulus was labeled in <50% of brains examined in the given knock-in line. Sensilla or glomeruli that have been renamed or reclassified have their former nomenclature listed in parentheses. Question marks indicate expression that has been reported but not functionally validated. * See also *Figure 3—source data 1* and *Figure 3—source data 2*.

| Glomerulus[†] | Sensillum | Tuning receptor(s) | Original co-receptor(s) | Orco-T2A-QF2 | Ir8a-T2A-QF2 | Ir76b-T2A-QF2 | Ir25a-T2A-QF2 | References |
|---|---|---|---|---|---|---|---|---|
| D | Ab9A | Or69aA, Or69aB | Orco | Yes | Variable | No | No | *Couto et al., 2005*; *Fishilevich and Vosshall, 2005* |
| DA1 | At1A | Or67d | Orco | Yes | No | Variable | Yes | *Couto et al., 2005*; *Fishilevich and Vosshall, 2005*; *Kurtovic et al., 2007* |
| DA2 | Ab4B | Or56a, Or33a | Orco | Yes | No | No | Yes | *Couto et al., 2005*; *Fishilevich and Vosshall, 2005* |
| DA3 | Ai2B (At2B) | Or23a | Orco | Yes | No | No | Yes | *Couto et al., 2005*; *Fishilevich and Vosshall, 2005*; *Lin and Potter, 2015* |
| DA4l | Ai3C (At3C) | Or43a | Orco | Yes | No | No | Yes | *Couto et al., 2005*; *Fishilevich and Vosshall, 2005*; *Lin and Potter, 2015* |
| DA4m | Ai3B (At3B) | Or2a | Orco | Yes | No | No | Yes | *Couto et al., 2005*; *Lin and Potter, 2015* |
| DC1 | Ai3A (At3A) | Or19a, Or19b | Orco | Yes | No | No | Yes | *Couto et al., 2005*; *Fishilevich and Vosshall, 2005*; *Lin and Potter, 2015* |
| DC2 | Ai1A (Ab6A) | Or13a | Orco | Yes | No | No | Yes | *Couto et al., 2005*; *Fishilevich and Vosshall, 2005*; *Lin and Potter, 2015* |
| DC3 | Ai2A (At2A) | Or83c | Orco | Yes | No | Yes | Variable | *Couto et al., 2005*; *Fishilevich and Vosshall, 2005*; *Lin and Potter, 2015* |
| DL1 | Ab1D | Or10a, Gr10a | Orco | Yes | Yes | Yes | Yes | *Couto et al., 2005*; *Fishilevich and Vosshall, 2005* |
| DL3 | At4B | Or65a, Or65b, Or65c | Orco | Yes | No | No | Yes | *Couto et al., 2005*; *Fishilevich and Vosshall, 2005* |
| DL4 | Ab10B | Or49a, Or85f | Orco | Yes | No | No | Yes | *Couto et al., 2005*; *Fishilevich and Vosshall, 2005* |
| DL5 | Ab4A | Or7a | Orco | Yes | No | No | Variable | *Couto et al., 2005* |
| DM1 | Ab1A | Or42b | Orco | Yes | No | No | Yes | *Couto et al., 2005*; *Fishilevich and Vosshall, 2005* |
| DM2 | Ab3A | Or22a, Or22b | Orco | Yes | No | No | Yes | *Couto et al., 2005*; *Fishilevich and Vosshall, 2005* |
| DM3 | Ab5B | Or47a, Or33b | Orco | Yes | No | No | No | *Couto et al., 2005*; *Fishilevich and Vosshall, 2005* |
| DM4 | Ab2A | Or59b | Orco | Yes | No | No | Yes | *Couto et al., 2005* |
| DM5 | Ab2B | Or85a, Or33b | Orco | Yes | No | No | No | *Couto et al., 2005*; *Fishilevich and Vosshall, 2005* |
| DM6 | Ab10A | Or67a | Orco | Yes | No | No | Yes | *Couto et al., 2005* |
| VA1d | At4C | Or88a | Orco | Yes | No | No | Yes | *Couto et al., 2005*; *Fishilevich and Vosshall, 2005* |
| VA1v | At4A | Or47b | Orco | Yes | No | No | Yes | *Couto et al., 2005*; *Fishilevich and Vosshall, 2005* |
| VA2 | Ab1B | Or92a | Orco | Yes | No | No | Yes | *Couto et al., 2005*; *Fishilevich and Vosshall, 2005* |
| VA3 | Ab9B | Or67b | Orco | Yes | Yes | Yes | Yes | *Couto et al., 2005*; *Fishilevich and Vosshall, 2005* |
| VA4 | Pb3B | Or85d | Orco | Yes | No | No | Yes | *Couto et al., 2005* |
| VA5 | Ai1B (Ab6B) | Or49b | Orco | Yes | Yes | Yes | Yes | *Couto et al., 2005*; *Lin and Potter, 2015* |

*Table 3 continued on next page*

*Table 3 continued*

| Glomerulus[†] | Sensillum | Tuning receptor(s) | Original co-receptor(s) | Orco-T2A-QF2 | Ir8a-T2A-QF2 | Ir76b-T2A-QF2 | Ir25a-T2A-QF2 | References |
|---|---|---|---|---|---|---|---|---|
| VA6 | Ab5A | Or82a | Orco | Yes | Yes | Yes | Yes | *Couto et al., 2005*; *Fishilevich and Vosshall, 2005* |
| VA7l | Pb2B | Or46a | Orco | Yes | No | No | Yes | *Couto et al., 2005*; *Fishilevich and Vosshall, 2005* |
| VA7m | UNK | UNK | Orco | Yes | No | Variable | Yes | *Couto et al., 2005* |
| VC1 | Pb2A | Or33c, Or85e | Orco | Yes | No | No | Yes | *Couto et al., 2005*; *Fishilevich and Vosshall, 2005* |
| VC2 | Pb1B | Or71a | Orco | Yes | No | No | Yes | *Couto et al., 2005*; *Fishilevich and Vosshall, 2005* |
| VC4 | Ab7B | Or67c | Orco | Yes | No | Yes | Yes | *Couto et al., 2005* |
| VM2 | Ab8A | Or43b | Orco | Yes | No | No | No | *Couto et al., 2005* |
| VM3 | Ab8B | Or9a | Orco | Yes | No | No | No | *Couto et al., 2005* |
| VM5d | Ab3B | Or85b?, Or98b? | Orco | Yes | Variable | No | Yes | *Couto et al., 2005* |
| VM5v | Ab7A | Or98a | Orco | Yes | Yes | No | Yes | *Couto et al., 2005*; *Fishilevich and Vosshall, 2005* |
| VM7d | Pb1A | Or42a | Orco | Yes | No | No | Yes | *Couto et al., 2005*; *Endo et al., 2007*; *Fishilevich and Vosshall, 2005* |
| VM7v (1) | Pb3A | Or59c | Orco | Yes | No | No | Yes | *Couto et al., 2005*; *Endo et al., 2007* |
| VC3 | Ac3B | Or35a | Orco, Ir76b | Yes | Yes | Yes | Yes | *Couto et al., 2005*; *Fishilevich and Vosshall, 2005*; *Silbering et al., 2011* |
| V | Ab1C | Gr21a, Gr63a | N/A | Yes | No | No | Yes | *Couto et al., 2005*; *Jones et al., 2007*; *Kwon et al., 2007* |
| DC4 | Sacculus, chamber III | Ir64a | Ir8a | Variable | Yes | No | Yes | *Ai et al., 2013*; *Ai et al., 2010*; *Silbering et al., 2011* |
| DL2d | Ac3A | Ir75b | Ir8a | Yes | Yes | No | Yes | *Prieto-Godino et al., 2017*; *Silbering et al., 2011* |
| DL2v | Ac3A | Ir75c | Ir8a | Yes | Yes | No | Yes | *Prieto-Godino et al., 2017*; *Silbering et al., 2011* |
| DP1l | Ac2 | Ir75a | Ir8a | Yes | Yes | Yes | Yes | *Silbering et al., 2011* |
| DP1m | Sacculus, chamber III | Ir64a | Ir8a | No | Yes | Yes | Yes | *Ai et al., 2013*; *Ai et al., 2010*; *Silbering et al., 2011* |
| VL2a | Ac4 | Ir84a | Ir8a | Yes | Yes | Yes | Yes | *Silbering et al., 2011* |
| VL2p | Ac1 | Ir31a | Ir8a | No | Yes | Yes | Yes | *Silbering et al., 2011* |
| VC5 | Ac2 | Ir41a | Ir8a, Ir25a, Ir76b | No | Yes | Yes | Yes | *Hussain et al., 2016*; *Min et al., 2013*; *Silbering et al., 2011* |
| VM1 | Ac1 | Ir92a | Ir8a, Ir25a, Ir76b | No | Yes | Yes | Yes | *Min et al., 2013*; *Silbering et al., 2011* |
| VM4 | Ac4 | Ir76a | Ir8a, Ir25a, Ir76b | Yes | Yes | Yes | Yes | *Benton et al., 2009*; *Min et al., 2013*; *Silbering et al., 2011* |
| VL1 | Ac1, Ac2, Ac4 | Ir75d | Ir25a | Yes | Yes | Yes | Yes | *Silbering et al., 2011* |
| VM6v (VM6) | Ac1 | Rh50, Amt | Ir25a | No | Yes (weak) | No | Yes | *Chai et al., 2019*; *Li et al., 2016*; *Schlegel et al., 2021*; *Vulpe et al., 2021*, this paper |
| VM6m (new) | Sacculus, chamber III | Rh50, Amt | N/A (this paper) | No | Yes (weak) | No | Yes | *Chai et al., 2019*; *Li et al., 2016*; *Schlegel et al., 2021*; *Vulpe et al., 2021*, this paper |
| VM6l* (new) | Sacculus, chamber III | Rh50, Amt | N/A (this paper) | No | Yes (strong) | No | Yes | *Chai et al., 2019*; *Li et al., 2016*; *Schlegel et al., 2021*; *Vulpe et al., 2021*, this paper |

Table 3 continued

| Glomerulus[†] | Sensillum | Tuning receptor(s) | Original co-receptor(s) | Orco-T2A-QF2 | Ir8a-T2A-QF2 | Ir76b-T2A-QF2 | Ir25a-T2A-QF2 | References |
|---|---|---|---|---|---|---|---|---|
| VP1d | Sacculus, chamber II | Ir40a, Ir93a | Ir25a | No | No | No | Yes | *Enjin et al., 2016*; *Frank et al., 2017*; *Knecht et al., 2017*; *Knecht et al., 2016*; *Marin et al., 2020*; *Silbering et al., 2011* |
| VP1l | Sacculus, chamber I | Ir21a, Ir93a | Ir25a | No | No | No | Yes | *Frank et al., 2017*; *Knecht et al., 2017*; *Knecht et al., 2016*; *Marin et al., 2020*; *Silbering et al., 2011* |
| VP1m | Sacculus, chamber I | Ir68a, Ir93a | Ir25a | No | No | No | Yes | *Frank et al., 2017*; *Knecht et al., 2017*; *Knecht et al., 2016*; *Marin et al., 2020*; *Silbering et al., 2011* |
| VP2 | Arista | Gr28b.d, Ir93a | Ir25a | No | No | No | Yes | *Enjin et al., 2016*; *Frank et al., 2017*; *Marin et al., 2020*; *Miwa et al., 2018*; *Ni et al., 2013* |
| VP3 | Arista | Ir21a, Ir93a | Ir25a | No | No | No | Yes | *Budelli et al., 2019*; *Enjin et al., 2016*; *Frank et al., 2017*; *Silbering et al., 2011* |
| VP4 | Sacculus, chambers I + II | Ir40a, Ir93a | Ir25a | No | No | No | Yes | *Enjin et al., 2016*; *Frank et al., 2017*; *Knecht et al., 2017*; *Knecht et al., 2016*; *Marin et al., 2020*; *Silbering et al., 2011* |
| VP5 | Sacculus, chamber II | Ir68a, Ir93a | Ir25a | No | No | No | Yes | *Frank et al., 2017*; *Knecht et al., 2017*; *Marin et al., 2020* |

*VM6l was initially named VC6 in version 1 of our pre-print (**Task et al., 2020**) but was reclassified using additional data from EM reconstructions in the antennal lobe (AL) and immunohistochemical experiments in the periphery (see **Figure 5**).

[†]The VM6 subdivisions (VM6v, VM6m, VM6l) are separated in this table for clarity but counted together as one glomerulus in accordance with **Schlegel et al., 2021**.

supplement 2C, Orco+ cluster). In total, the *Ir76b-T2A-QF2* knock-in labels 15 glomeruli consistently and two variably (**Figure 3—source data 1** and **Figure 3—source data 2**).

*Ir25a-T2A-QF2* innervation of the AL was the most expanded compared to what has previously been reported. In addition to the novel expression we identified in the palps (**Figure 2H**), we found that the *Ir25a* knock-in innervates many Orco+ glomeruli receiving inputs from the antennae (**Figure 3D**). The extensive, dense innervation of the AL by Ir25a+ processes made identification of individual glomeruli difficult and necessitated further experiments to fully characterize this expression pattern (described in greater detail below). While it was previously reported that the transgenic *Ir25a-Gal4* line labels only a subset of Ir25a+ neurons (compared to anti-Ir25a antibody staining), it was assumed that neurons not captured by the transgenic line would reside in coeloconic sensilla, the arista, or sacculus (the original locations for all IR+ OSNs) (**Abuin et al., 2011**). When we crossed *Ir25a-Gal4* to a strong reporter, we found labeling of a few Orco+ glomeruli (**Figure 3—figure supplement 2D**), but this was a small fraction of those labeled by the knock-in. To further examine *Ir25a* expression and the potential co-expression of multiple co-receptors in greater detail, we employed a combination of approaches, including single-nucleus RNAseq (snRNAseq), immunohistochemistry, and optogenetics.

## Confirmation of co-receptor co-expression

The innervation of the same glomeruli by multiple co-receptor knock-in lines challenges the previous view of segregated chemosensory receptor expression in *D. melanogaster* and suggests two possible explanations: either the same olfactory neurons express multiple co-receptors (co-expression) or different populations of olfactory neurons expressing different receptors converge upon the same glomeruli (co-convergence). These scenarios are not necessarily mutually exclusive. To examine these possibilities in a comprehensive, unbiased way, we analyzed snRNAseq data from adult fly antennae (**McLaughlin et al., 2021**). **Figure 4A** shows the expression levels of the four co-receptor genes in 20 transcriptomic clusters (tSNE plots [**Van der Maaten and Hinton, 2008**], top row), which were mapped to 24 glomerular targets in the brain (AL maps, bottom row). The proportion of cells in each cluster expressing the given co-receptor gene is indicated by the opacity of the glomerular fill color, normalized to maximum expression for that gene (see Materials and methods and **Figure 4—source data 1** for details on expression normalization). The OSN classes to which these clusters map include Orco+ neurons (**Figure 4A**, right column, teal), Ir25a+ neurons (**Figure 4A**, right column, purple), Ir8a+ neurons (**Figure 4A**, right column, pink), and GR+neurons (**Figure 4A**, right column, dark blue).

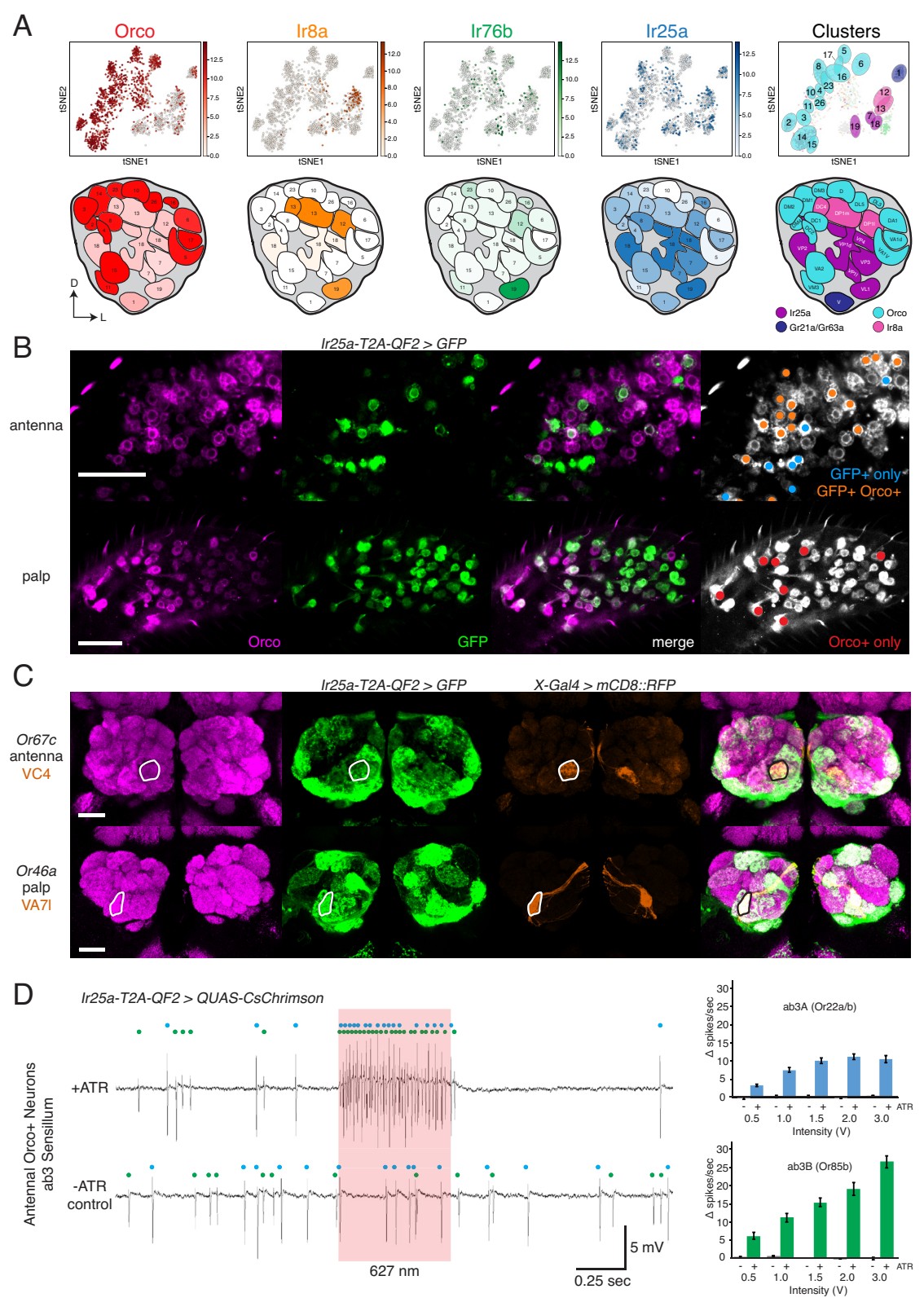

**Figure 4.** Confirmation of co-receptor co-expression. (**A**) snRNAseq of adult fly antennae (**McLaughlin et al., 2021**) confirms expanded expression of olfactory co-receptors. Top: tSNE plots show expression of each co-receptor in 20 decoded olfactory sensory neuron (OSN) clusters. Bottom: clusters were mapped to 24 glomeruli. Opacity of fill in each glomerulus indicates the proportion of cells in that cluster expressing the given co-receptor, normalized to total expression for that co-receptor gene (see **Figure 4—source data 1**). Right column: clusters color-coded according to original

*Figure 4 continued on next page*

*Figure 4 continued*

chemoreceptor gene family. Compass: D = dorsal; L = lateral. (**B**) Anti-Orco antibody staining in antennal cryosections (top) and whole-mount palps (bottom) confirms co-expression of Orco and Ir25a in the periphery (genotype: *Ir25a-T2A-QF2>GFP*). Right panels show cells pseudo-colored gray with specific single- or double-labeled cells indicated by colored cell markers (GFP+ only in blue, GFP+Orco+ in orange, Orco+ only in red). (**C**) Co-labeling experiments with various transgenic *Gal4* lines driving mCD8::RFP (orange) and the *Ir25a-T2A-QF2* knock-in driving GFP (green). *Ir25a-T2A-QF2* labels glomeruli innervated by both antennal (top) and palpal (bottom) OSNs. (**D**) Verification of *Ir25a* expression in antennal ab3 sensilla using optogenetics. Single sensillum recordings (SSR) from ab3 Orco+ neurons in *Ir25a-T2A-QF2>QUAS-CsChrimson* flies. Representative traces from ab3 using 1.5 V of 627 nm LED light (red box) to activate *CsChrimson*. Bottom trace is control animal, which has the same genotype as the experimental animal but was not fed the required all-trans retinal cofactor (-ATR). Spikes from the ab3A and ab3B neurons are indicated by blue and green dots, respectively. Right: quantification of neuronal activity in response to light at various LED intensities (N = 7–12). These optogenetic experiments support *Ir25a* expression in both ab3A neurons (*Or22a/b*, top; corresponding to DM2 glomerulus) and ab3B neurons (*Or85b*, bottom; corresponding to VM5d glomerulus). Scale bars = 25 µm. See also *Figure 4—figure supplement 1*, *Table 3*, *Figure 4—source data 1*, *Figure 4—source data 2*, and *Figure 4—source data 3*.

The online version of this article includes the following source data and figure supplement(s) for figure 4:

**Source data 1.** snRNAseq co-receptor expression in adult olfactory sensory neurons (OSNs).

**Source data 2.** Individual glomerular analyses.

**Source data 3.** Optogenetic validation of *Ir25a* expression.

**Figure supplement 1.** Optogenetic experiments to examine *Ir25a* expression in Orco+ neurons.

They also include example OSNs from all sensillar types (basiconic, intermediate, trichoid, coeloconic) as well as from the arista and sacculus. The snRNAseq analyses confirmed expanded expression of all four co-receptor genes into OSN classes not traditionally assigned to them. For example, *Orco* and *Ir25a* are expressed in cluster 1, which maps to the V glomerulus (Gr21a+/Gr63a+). Similarly, *Ir8a* and *Ir76b* are expressed in cluster 19 (VL1 glomerulus, Ir25a+), and *Ir25a* is expressed in multiple Orco+ clusters (such as 15/VA2, 16/DL3, and 8/DC1).

The snRNAseq analyses confirm transcript co-expression in olfactory neurons in the periphery. To demonstrate protein co-expression in OSNs, we performed anti-Orco antibody staining on *Ir25a-T2A-QF2>GFP* antennae and palps (**Figure 4B**). In the antennae, we found examples of Orco+ GFP+ double-labeled cells, as well as many cells that were either GFP+ or Orco+ (**Figure 4B**, top-right panel). Interestingly, in the palps the vast majority of cells were double labeled. We found a small population of palpal neurons that were only Orco+, and no neurons that were only GFP+ (**Figure 4B**, bottom-right panel). These results are consistent with our anti-Ir25a staining experiments in the palps (**Figure 2I**), which showed that most of the ~120 palpal OSNs express Ir25a protein.

The snRNAseq data from the antennae and peripheral immunohistochemical experiments in the palps helped to identify some of the novel OSN populations expressing *Ir25a*. We extended these analyses with co-labeling experiments in which we combined transgenic *OrX-*, *IrX-*, or *GrX-Gal4* lines labeling individual glomeruli with the *Ir25a* knock-in to verify the glomerular identity of Ir25a+ axonal targets in the AL. Two examples are shown in **Figure 4C** (one antennal and one palpal OSN population), and the full list of OSN classes checked can be found in **Figure 4—source data 2**.

For some OSN classes not included in the snRNAseq dataset for which co-labeling experiments yielded ambiguous results, we employed an optogenetic approach. We used the *Ir25a-T2A-QF2* knock-in to drive expression of *QUAS-CsChrimson*, a red-shifted channelrhodopsin (**Klapoetke et al., 2014**), and performed single sensillum recordings (SSR) from sensilla previously known to house only Orco+ neurons. If these neurons do express *Ir25a*, then stimulation with red light should induce neuronal firing. We recorded from ab3 sensilla, which have two olfactory neurons (A and B; indicated with blue and green dots, respectively, in **Figure 4D**). Ab3A neurons innervate DM2 and ab3B neurons innervate VM5d. Both neurons responded to pulses of 627 nm light at various intensities in a dose-dependent manner, confirming *Ir25a* expression in these neurons. No light-induced responses were found in control flies, which had the same genotype as experimental flies but were not fed all-trans retinal (-ATR), a necessary co-factor for channelrhodopsin function (see Materials and methods). We used similar optogenetic experiments to examine *Ir25a* expression in OSN classes innervating DM4 (ab2A, Or59b+) and DM5 (ab2B, Or85a/Or33b+) (**Figure 4—figure supplement 1A and B**), as well as D (ab9A, Or69aA/aB+) and VA3 (ab9B, Or67b+) (**Figure 4—figure supplement 1C and D**). These experiments indicated that *Ir25a* is expressed in ab2A (DM4) and ab9B (VA3) neurons, but not ab2B (DM5) or ab9A (D) neurons (see also **Figure 4—source data 2** and **Figure 4—source data 3**). Results of these experiments are summarized in **Table 3**.

## Identification of new OSN classes

The co-receptor knock-ins allowed us to analyze the olfactory neuron innervation patterns for all AL glomeruli. Interestingly, the *Ir8a-T2A-QF2* and *Ir25a-T2A-QF2* knock-ins strongly labeled a previously uncharacterized posterior region of the AL. By performing a co-labeling experiment with *Ir41a-Gal4*, which labels the VC5 glomerulus, we narrowed down the anatomical location of this region and ruled out VC5 as the target of these axons (*Figure 5A*). While both knock-ins clearly labeled VC5, they also labeled a region lateral and slightly posterior to it (*Figure 5A*, outline). We performed additional co-labeling experiments with *Ir8a-T2A-QF2* and various *Gal4* lines labeling all known posterior glomeruli to confirm that this AL region did not match the innervation regions for other previously described OSN populations (*Figure 5—figure supplement 1*). We recognized that this novel innervation pattern appeared similar to a portion of the recently identified Rh50+ ammonia-sensing olfactory neurons (*Vulpe et al., 2021*). Co-labeling experiments with *Rh50-Gal4* and *Ir8a-T2A-QF2* confirmed that they indeed partially overlapped (*Figure 5B*). We determined that these Rh50+ olfactory neurons mapped to a portion of the VM6 glomerulus, with the strongly Ir8a+ region innervating the 'horn' of this glomerulus. The difference in innervation patterns between Ir8a+ and Rh50+ neurons in this AL region suggested at least two different subdivisions or OSN populations within this VM6 glomerulus. In fact, in between the main body of VM6 and the Ir8a+ horn there appeared to be a third region (*Figure 5B*, horn outlined in white, other two regions outlined in blue). We designated these subdivisions VM6l, VM6m, and VM6v (for lateral, medial, and ventral). We coordinated the naming of this glomerulus with recent connectomics analyses of the entire fly AL (*Schlegel et al., 2021*). In this connectomics study, dendrites of olfactory projection neurons were found to innervate the entire region described here as VM6l, VM6m, and VM6v. No projection neurons were identified to innervate only a subdomain. As such, the new VM6 nomenclature reflects this unique subdivision of a glomerulus by OSNs but not second-order projection neurons.

We sought to determine the identity of the olfactory neurons that might be innervating these three VM6 subdivisions. Rh50+ neurons can be found in two regions of the antenna: ac1 coeloconic sensilla and the sacculus (*Figure 5C*; *Vulpe et al., 2021*). The shape of the VM6v subdomain most closely matches the glomerulus described as VM6 by previous groups (e.g., *Couto et al., 2005*; *Endo et al., 2007*), which had been suggested to be innervated by coeloconic sensilla (*Chai et al., 2019*; *Li et al., 2016*). In addition, antibody staining had previously shown that Rh50+ ac1 neurons broadly co-express Ir25a but generally not Ir8a (*Vulpe et al., 2021*). This suggested that the other VM6 subdomains might be innervated by the Rh50+ sacculus olfactory neurons. Antibody staining in *Rh50-Gal4>GFP* antennae confirmed co-expression with both Ir25a protein (broad overlap) and Ir8a protein (narrow overlap) in the third chamber of the sacculus (*Figure 5D*; quantified in *Table 4*). Most sacculus neurons appear to be Ir25a+, and in contrast to the *Ir8a* knock-in, the three VM6 subdivisions are all strongly innervated by the *Ir25a* knock-in (*Figure 5A*). Two previously described OSN populations in the third chamber of the sacculus had been characterized to express *Ir8a* along with *Ir64a* and innervate the DP1m and DC4 glomeruli (*Ai et al., 2013*; *Ai et al., 2010*). To demonstrate that the Rh50+ Ir8a+ sacculus neurons represented a distinct olfactory neuron population, we performed immunohistochemistry experiments in *Rh50-Gal4>GFP* antennae with an anti-Ir64a antibody (*Figure 5E*, top), and in *Ir64a-Gal4>GFP* antennae with an anti-Ir8a antibody (*Figure 5E*, bottom). These experiments confirmed a new, distinct population of Ir8a+ Ir64a- cells in the sacculus.

The VM6l olfactory projections are difficult to identify in the hemibrain connectome (*Scheffer et al., 2020*) due to the medial truncation of the AL in that dataset (see *Schlegel et al., 2021* for additional details). Here, we used FlyWire (*Dorkenwald et al., 2020*), a recent segmentation of a full adult fly brain (FAFB) (*Zheng et al., 2018*), to reconstruct the VM6 OSN projections in both left and right ALs. Synapse-based hierarchical clustering (syNBLAST) (*Buhmann et al., 2021*) of the VM6 OSNs demonstrated the anatomical segregation into three distinct subpopulations: VM6l, VM6m, and VM6v (*Figure 5F*). This subdivision was subsequently confirmed in a reanalysis of the VM6 glomerulus in the hemibrain dataset (*Schlegel et al., 2021*). Olfactory neurons innervating VM6l were strongly Ir8a+, while olfactory neurons innervating VM6m and VM6v were weakly and sparsely Ir8a+ (see *Figure 3—source data 2*, page 3). This pattern may be due to *Ir8a* expression in only one or a few cells.

Based on the EM reconstructions, genetic AL analyses, and peripheral staining experiments, we propose a model of the anatomical locations and molecular identities of the olfactory neurons innervating the VM6 subdivisions (*Figure 5F*). All VM6 subdivisions broadly express *Rh50* and *Ir25a*; the

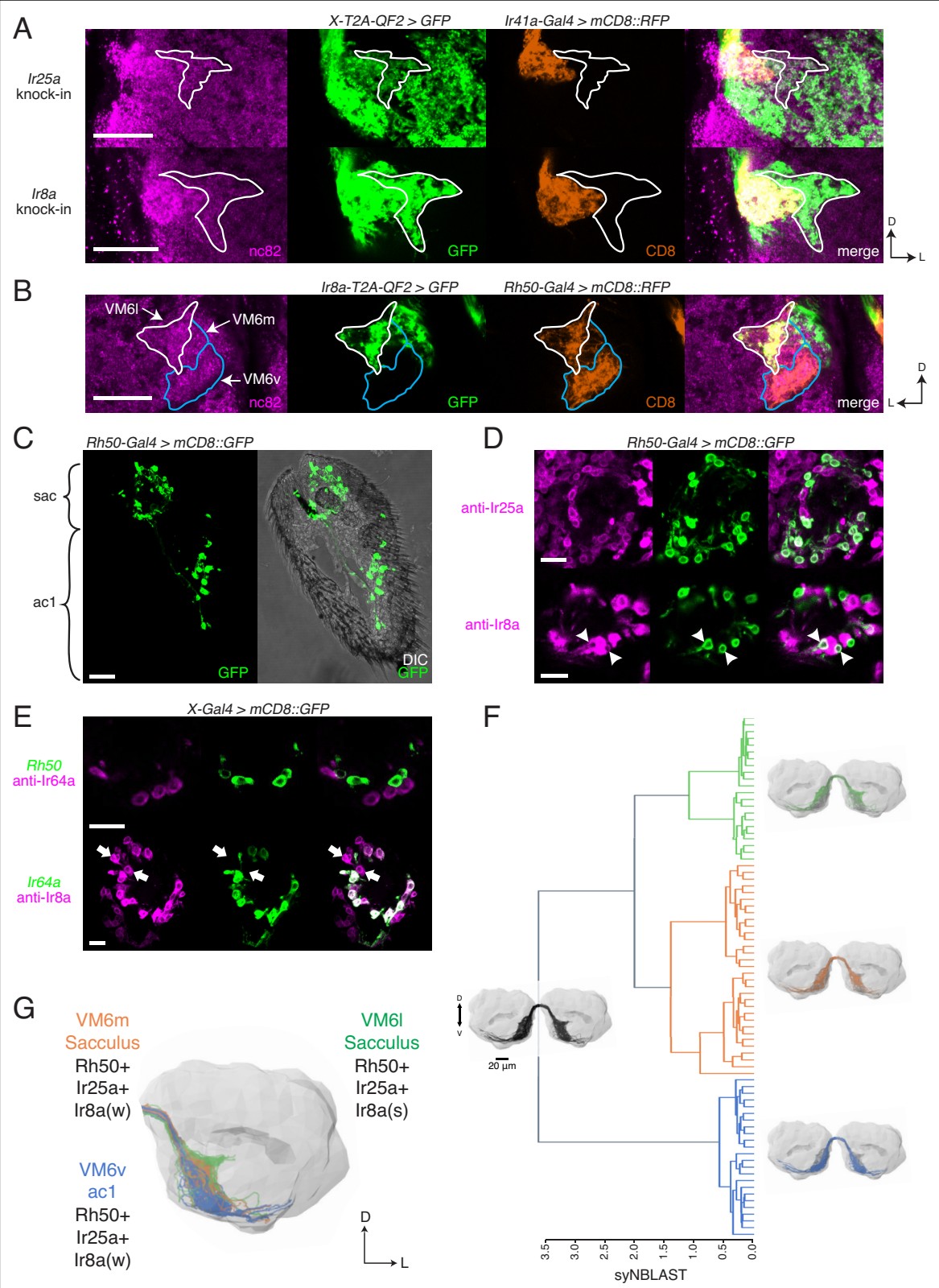

**Figure 5.** Identification of new olfactory sensory neuron (OSN) classes. (**A**) Co-labeling experiments with *Ir41a-Gal4* show that both *Ir25a-T2A-QF2* and *Ir8a-T2A-QF2* label the VC5 glomerulus (orange), and also a previously unidentified antennal lobe (AL) region (outline). (**B**) The new innervation pattern corresponds to the 'horn' (white outline) of the VM6 glomerulus labeled by Rh50+ neurons (orange). One portion of VM6 is strongly Ir8a+ (VM6l), while two other portions show little to no Ir8a expression (VM6m and VM6v, blue outlines). (**C**) *Rh50-Gal4>GFP* labels neurons in the sacculus (sac)

*Figure 5 continued on next page*

*Figure 5 continued*

and antennal coeloconic ac1 sensilla. (**D**) In the sacculus, all Rh50+ neurons appear to be Ir25a+ (top), and a subset are Ir8a+ (bottom, arrowheads). (**E**) Top: Rh50+ neurons in the sacculus do not overlap with Ir64a+ neurons. Bottom: there are two distinct populations of Ir8a+ neurons in the sacculus – those that are Ir64a+ and those that are Ir64a- (arrows). The latter likely correspond to Rh50+ neurons. (**F**) EM reconstructions of VM6 OSNs in a full brain volume (*Dorkenwald et al., 2020*) reveal three distinct subpopulations. (**G**) Model of OSN innervation of the VM6 region. VM6 can be subdivided into three OSN populations based on anatomical location in the periphery and chemoreceptor expression: VM6v (blue) OSNs originate in ac1, strongly (s) express Rh50 and Ir25a, and weakly (w) or infrequently express Ir8a; VM6m (orange) neurons originate in the sacculus and have a similar chemoreceptor expression profile to VM6v; VM6l (green) OSNs originate in the sacculus but strongly express Ir8a in addition to Rh50 and Ir25a. Compass: D = dorsal, L = lateral. Scale bars: 20 μm in (**A–C**) and (**F**), 10 μm in (**D, E**). N = 9–11 for (**C–E**). See also *Figure 5—figure supplement 1* and *Tables 3 and 4*.

The online version of this article includes the following figure supplement(s) for figure 5:

**Figure supplement 1.** The new glomerular region labeled by the *Ir8a* knock-in does not correspond to previously identified posterior glomeruli.

VM6v OSNs are housed in ac1 sensilla and express *Ir8a* either weakly or only in a small subset of neurons; both the VM6m and VM6l OSNs are found in the sacculus and can be distinguished by their levels or extent of *Ir8a* expression, with VM6l neurons being strongly Ir8a+. Because all three VM6 subdivisions share the same downstream projection neurons, this AL region has been classified as a single glomerulus (*Schlegel et al., 2021*). We maintain this convention here, for a total of 58 AL glomeruli. It is possible that this number may need to be re-evaluated in the future, and the three VM6 subdivisions reconsidered as bona fide separate glomeruli (bringing the OSN glomerular total to 60). Such a separation might be warranted if it is found that these OSN populations express different tuning receptors, and those receptors respond to different odorants.

*Table 3* summarizes the chemosensory receptor expression patterns for all four co-receptor knock-in lines across all OSNs, sensillar types, and glomeruli. For clarity, this summary considers the newly identified OSN populations described here separately. We find that *Orco-T2A-QF2* consistently labels 45 total glomeruli out of 58 (7 more than previously reported); *Ir8a-T2A-QF2* consistently labels 18 glomeruli (8 more than previously identified); *Ir76b-T2A-QF2* consistently labels 15 glomeruli (11 more than previously identified); and *Ir25a-T2A-QF2* consistently labels 51 glomeruli (39 more than previously identified).

**Table 4.** Co-expression of Rh50 and Ir8a in the sacculus (related to *Figure 5*).

Antennal cryosections of *Rh50-Gal4>GFP* flies were stained with an anti-Ir8a antibody, and the overlap of Ir8a+ and GFP+ cells was quantified in the sacculus. 22% of Ir8a+ cells expressed Rh50, 35% of Rh50+ cells expressed Ir8a, and 16% of all cells were double labeled. N = 11.

| Genotype | Sample | Ir8a+ cells | GFP+ cells | Double-labeled cells | Total cells |
|---|---|---|---|---|---|
| *Rh50-Gal4>GFP* | 20210226 a1 | 18 | 9 | 2 | 25 |
| *Rh50-Gal4>GFP* | 20210226 a2 | 22 | 15 | 4 | 33 |
| *Rh50-Gal4>GFP* | 20210226 a3 | 41 | 22 | 7 | 56 |
| *Rh50-Gal4>GFP* | 20210226 a4 | 41 | 14 | 5 | 50 |
| *Rh50-Gal4>GFP* | 20210129 a1 | 26 | 20 | 9 | 37 |
| *Rh50-Gal4>GFP* | 20210129 a2 | 32 | 24 | 7 | 49 |
| *Rh50-Gal4>GFP* | 20210129 a3 | 29 | 19 | 7 | 41 |
| *Rh50-Gal4>GFP* | 20210216 a1 | 26 | 21 | 8 | 39 |
| *Rh50-Gal4>GFP* | 20210216 a2 | 30 | 18 | 7 | 41 |
| *Rh50-Gal4>GFP* | 20210216 a3 | 34 | 23 | 8 | 49 |
| *Rh50-Gal4>GFP* | 20210216 a4 | 34 | 23 | 9 | 48 |
| | Total across samples: | 333 | 208 | 73 | 468 |
| | | Proportion of Ir8a+ cells that are GFP+: | Proportion of GFP+ cells that are Ir8a+: | Proportion of all cells that are double labeled: | |
| | | 0.22 | 0.35 | 0.16 | |

## Co-receptor contributions to olfactory neuron physiology

How might the broad, combinatorial co-expression of various chemosensory families affect olfactory neuron function? To begin to address this question, we examined olfactory responses in neuronal populations co-expressing just two of the four chemosensory receptor families (*Orco* and *Ir25a*). We chose to test eight OSN classes previously assigned to the Orco+ domain that we found to have strong or intermediate *Ir25a* expression – two in the antennae and six in the maxillary palps. The two antennal OSN classes are found in the same ab3 sensillum (ab3A, Or22a/b+, DM2 glomerulus; and ab3B, Or85b+, VM5d glomerulus). The six palpal OSN classes represent the entire known olfactory neuron population of the maxillary palps (pb1A, Or42a+, VM7d; pb1B, Or71a+, VC2; pb2A, Or33c/Or85e+, VC1; pb2B, Or46a+, VA7l; pb3A, Or59c+, VM7v; pb3B, Or85d+, VA4). In both the antennae and the palps, we compared the olfactory responses of OSNs to a panel of 13 odorants in three genotypes: *wildtype*, *Ir25a²* mutant, and *Orco²* mutant flies. This panel included odorants typically detected by *ORs*, such as esters and aromatics, and odorants typically detected by *IRs*, such as acids and amines (**Silbering et al., 2011**). In the previously accepted view of olfaction in *D. melanogaster*, Orco+ neurons express only Orco/OrX receptors, and all olfactory responses in the neurons can be attributed to these receptors. Thus, in an *Ir25a²* mutant background, there should be no difference in olfactory responses from *wildtype* if either (a) *Ir25a* is not expressed in these neurons or (b) *Ir25a* is expressed, but is not playing a functional role in these neurons. In an *Orco²* mutant background, there would be no trafficking of Orco/OrX receptors to the dendritic membrane, and no formation of functional ion channels (**Benton et al., 2006**; **Larsson et al., 2004**). Thus, in the traditional view of insect olfaction, *Orco²* mutant neurons should have no odor-evoked activity. However, in the new co-receptor co-expression model of olfaction, if *Ir25a* is contributing to olfactory responses in Orco+ neurons, then mutating this co-receptor might affect the response profiles of these neurons. Similarly, *Orco²* mutant neurons that co-express *Ir25a* might retain some odor-evoked activity.

We first examined olfactory responses in palp basiconic sensilla. In the palps, three types of basiconic sensilla (pb1, pb2, and pb3) contain two neurons each (A and B) (**Figure 6A**), for a total of six OSN classes (**Couto et al., 2005**; **de Bruyne et al., 1999**; **Fishilevich and Vosshall, 2005**; **Goldman et al., 2005**; **Ray et al., 2008**; **Ray et al., 2007**). We found robust responses to several odorants in our panel in both the *wildtype* and *Ir25a²* mutant flies, including odorants like 1-octen-3-ol typically considered as an OR ligand (**Figure 6B**), and IR ligands like pyrrolidine. Neither odor-evoked nor spontaneous activity was detected in the *Orco²* mutant (**Figure 6B**, bottom row; see also **Figure 6—figure supplement 1A**). This was true of all sensilla tested in the palps. The SSR experiments in **Figure 6A–D** were performed at 4–8 DPE. We recently discovered that neurodegeneration of *Orco²* mutant olfactory neurons occurs in the palps by ~6 DPE (**Task and Potter, 2021**), which could potentially confound our interpretation. We repeated the experiments in young (1–3 DPE) flies but similarly detected neither odor-evoked activity nor spontaneous activity in *Orco²* mutant palpal neurons (**Figure 6—figure supplement 1B**). There was also no spontaneous or odor-evoked activity in an *Ir25a²; Orco²* double mutant (**Figure 6—figure supplement 1C**). This suggests one of three possibilities: first, *Orco²* mutant neurons in the palps could already be dysfunctional at this early stage, despite not yet showing cell loss, and *Ir25a*-dependent activity is not sufficient to maintain either baseline or stimulus-induced activity; second, *Ir25a* function may be *Orco*-dependent in these cells, or act downstream of *Orco*, such that loss of *Orco* function affects *Ir25a* function; third, we did not stimulate neurons with an *Ir25a*-dependent odorant. The latter possibility would not, however, explain why there is no spontaneous activity in these cells. Future experiments will be needed to address these possibilities. Given the lack of neuronal activity in the *Orco²* mutant, we focused subsequent analyses in the palps on the two other genotypes: *wildtype* and *Ir25a²*.

The response in the pb1A neuron to 1-octen-3-ol was significantly higher in the *Ir25a²* mutant compared to the *wildtype* (Mann–Whitney *U* test, p=0.0016), as was the response to methyl salicylate (p=0.0177), while the response to ethyl acetate (EA) was higher in *wildtype* (p=0.008) (**Figure 6C**; see **Figure 6—source data 1** for results of all statistical analyses). The differences in responses across all six OSN classes in the palps between *wildtype* and *Ir25a²* mutant flies are summarized in **Figure 6D**. In each neuron class, we found 1–3 odorants whose response profiles differed between the two genotypes. However, the specific stimuli eliciting different responses, and the directionality of those responses, varied. For example, 2,3-butanedione elicited higher responses in the *Ir25a²* mutant in both pb2B and pb3A neurons, but lower responses in the mutant (higher in the *wildtype*) in pb3B.

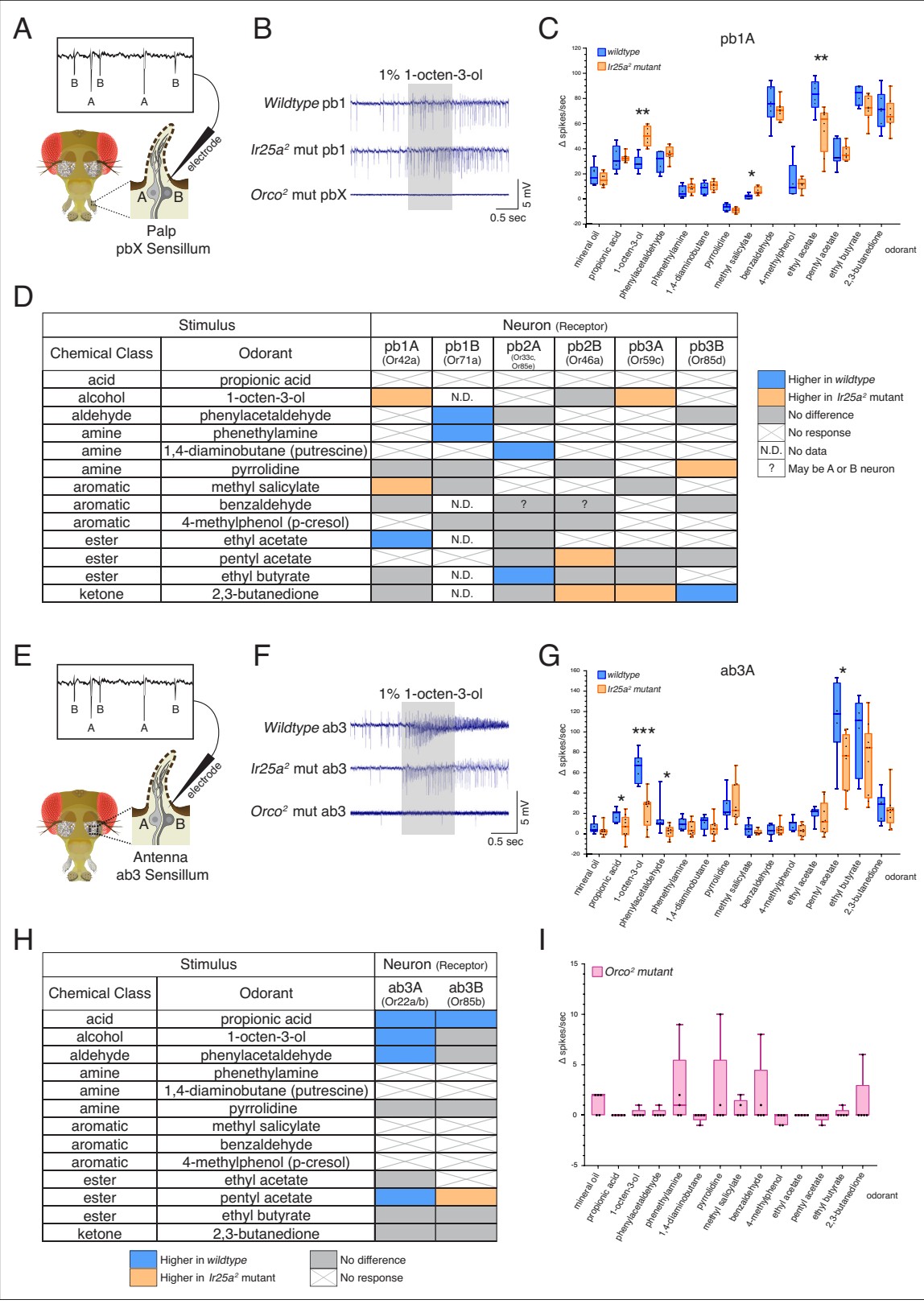

**Figure 6.** Co-receptor contributions to olfactory neuron physiology. (**A–I**) Single sensillum recording (SSR) experiments were performed in three genetic backgrounds: *wildtype*, *Ir25a²* mutant, and *Orco²* mutant flies. A panel of 13 odorants was tested. In all box plots, *p<0.05, **p<0.01, and ***p<0.001. (**A**) Cartoon of a fly head, zooming in on a single sensillum in the palp. Each palpal sensillum (pbX) contains two neurons, A and B. An electrode is inserted into the sensillum, and neuronal activity is recorded in response to odorants. Activity of the A and B neurons can be distinguished based on

*Figure 6 continued*

their spike amplitudes (top). (**B**) Representative traces from recordings in palp basiconic pb1 sensilla in the three genotypes in response to 1% 1-octen-3-ol. Sensilla were identified based on responses to reference odorants (***de Bruyne et al., 1999***; see Materials and methods). The *Orco²* mutant did not exhibit odor-evoked activity nor spontaneous activity, making it difficult to determine the identity of the recorded sensillum. *Orco²* mutant sensilla are thus denoted pbX. (**C**) Quantification of responses to the panel of odorants in *wildtype* (blue; N = 5–9 flies) and *Ir25a²* mutant (orange; N = 6–10 flies) pb1A neurons. Responses were higher in the *Ir25a²* mutant than in the *wildtype* for 1-octen-3-ol and methyl salicylate, and lower in the *Ir25a²* mutant for ethyl acetate. Mann–Whitney $U$ tests indicated these differences were statistically significant: 1-octen-3-ol: $Mdn_{Ir25amut}$ = 50, $Mdn_{wildtype}$ = 28, $U(N_{Ir25amut}$ = 8, $N_{wildtype}$ = 5) = 0, p=0.0016; methyl salicylate: $Mdn_{Ir25amut}$ = 5, $Mdn_{wildtype}$ = 2, $U(N_{Ir25amut}$ = 7, $N_{wildtype}$ = 5) = 3, p=0.0177; ethyl acetate: $Mdn_{Ir25amut}$ = 63.5, $Mdn_{wildtype}$ = 83.5, $U(N_{Ir25amut}$ = 8, $N_{wildtype}$ = 6) = 4, p=0.008. (**D**) Summary of differences in responses across all six neuron classes in the palps between *wildtype* and *Ir25a²* mutant flies. Comparisons were made using Mann–Whitney $U$ tests. Orange indicates higher response in *Ir25a²* mutant, blue indicates higher response in *wildtype*. Gray is no difference between genotypes, X indicates no response to the given stimulus, and N.D. is no data (strong A neuron response obscured B neuron spikes preventing quantification). In the *wildtype*, for one sensillum-odorant combination (pb2 and benzaldehyde), it could not be distinguished if responses arose from the A or B neuron or both (indicated by a question mark). (**E**) Fly head cartoon, zooming in on a single sensillum in the antenna. We recorded from antennal ab3 sensilla, each of which contains two neurons, A and B. As in the palps, responses from these neurons can be distinguished based upon their spike amplitude (top). (**F**) Representative traces from recordings in antennal basiconic ab3 sensilla in the three genotypes in response to 1% 1-octen-3-ol. In *Orco²* mutant ab3 sensilla spontaneous activity was observed, but there was no significant odor-evoked activity. *Wildtype* N = 7 sensilla from five flies; *Ir25a²* mutant N = 10 sensilla from five flies. (**G**) Quantification of responses in *wildtype* (blue; N = 7) and *Ir25a²* mutant (orange; N = 9) ab3A neurons. Responses were significantly higher in *wildtype* compared to *Ir25a²* mutant ab3A neurons for four odorants (Mann–Whitney $U$ results in parentheses; all $N_{wildtype}$ = 7 and $N_{Ir25amut}$ = 9): propionic acid ($Mdn_{wildtype}$ = 21, $Mdn_{Ir25amut}$ = 7, $U$ = 12.5, p=0.0441); 1-octen-3-ol ($Mdn_{wildtype}$ = 67, $Mdn_{Ir25amut}$ = 29, $U$ = 1.5, p=0.0004); phenylacetaldehyde ($Mdn_{wildtype}$ = 10, $Mdn_{Ir25amut}$ = 3, $U$ = 9, p=0.015); and pentyl acetate ($Mdn_{wildtype}$ = 118, $Mdn_{Ir25amut}$ = 77, $U$ = 9, p=0.0164). Difference between *wildtype* and *Ir25a²* mutant to phenylacetaldehyde is significant even with the large *wildtype* outlier removed (p=0.0336). (**H**) Summary of differences in responses in the two neuron classes in ab3 between *wildtype* and *Ir25a²* mutant flies. Comparisons were made using Mann–Whitney $U$ tests. Orange indicates higher response in *Ir25a²* mutant, blue indicates higher response in *wildtype*, gray is no difference between genotypes, and X is no response to the given stimulus. One *Ir25a²* mutant fly was excluded from analyses as it had high responses to the mineral oil control (40–53 Δ spikes/s), not seen in any other animal of any genotype. (**I**) Weak responses in *Orco²* mutant flies to certain stimuli (≤10 Δ spikes/s) were occasionally detected. While there were some statistically significant differences from mineral oil control (pentyl acetate p=0.0109, propionic acid p=0.0434, ethyl acetate p=0.0434, 1,4-diaminobutane p=0.0109, p-cresol p=0.0021), these were not deemed biologically significant due to very small Δ spike values relative to zero. For more details, see Materials and methods. N = 5 flies. See also ***Figure 6—figure supplements 1–3*** and ***Figure 6—source data 1***.

The online version of this article includes the following source data and figure supplement(s) for figure 6:

**Source data 1.** Single sensillum recording (SSR) of *Ir25a* mutant, *Orco* mutant, and *wildtype* flies.

**Figure supplement 1.** Electrophysiological experiments to examine *Ir25a* function in Orco+ neurons.

**Figure supplement 2.** No IrX expression of the top candidates in the maxillary palps.

**Figure supplement 3.** Example traces for odorants eliciting differences between *wildtype* and *Ir25a* mutant sensilla.

Interestingly, when we examined a list of candidate IrX tuning receptors (***Li et al., 2021***) in the palps using in situ, we did not find expression (see ***Figure 6—figure supplement 2*** and Appendix 1—key resources table). This suggests that *Ir25a* may not be functioning as a traditional co-receptor in Orco+ olfactory neurons in the palps (an expanded role for *Ir25a* beyond co-reception has previously been suggested; see ***Budelli et al., 2019***; ***Chen et al., 2015***).

We next examined olfactory responses in antennal basiconic ab3 sensilla in *wildtype*, *Ir25a²* mutant, and *Orco²* mutant flies (***Figure 6E–I***). As in the palps, ab3 sensilla contain two neurons, A and B (***Figure 6E***). In contrast to the palps, *Orco²* mutant ab3 sensilla did occasionally show spontaneous activity (***Figure 6F***, bottom row; see ***Figure 6—figure supplement 1D*** and ***Figure 6—figure supplement 3*** for additional example traces). Although there are two Orco+ neurons in this sensillum, we consistently observed only a single spike amplitude in the *Orco²* mutant. Thus, we cannot determine at this time whether this activity arises from the A or B neuron. We occasionally observed small responses (≤10 Δ spikes/s) in the *Orco²* mutant; however, across all flies tested, these responses were not significantly different from the mineral oil control (***Figure 6I***; statistical analyses can be found in ***Figure 6—source data 1***). For these reasons, *Orco²* mutant flies were excluded from the analyses in ***Figure 6G and H***.

As in the palps, we found significant differences in the responses of both ab3A and ab3B neurons to some odorants between the two genotypes. A comparison of all ab3A responses between the *wildtype* and *Ir25a²* mutant genotypes is shown in ***Figure 6G***, and results from both the A and B neurons are summarized in ***Figure 6H*** (Mann–Whitney $U$, as in ***Figure 6A–C***; see ***Figure 6—source data 1*** for all analyses). In the ab3A neuron, the *wildtype* showed higher responses to propionic acid (p=0.0441),

1-octen-3-ol (p=0.0004), phenylacetaldehyde (p=0.015), and pentyl acetate (p=0.0164). Interestingly, two of these four odorants are typically associated with *IRs* (propionic acid and phenylacetaldehyde). In the ab3B neuron, only two odorants elicited significantly different responses between the *wildtype* and *Ir25a²* mutant: propionic acid (response higher in *wildtype*, as with ab3A; p=0.0388), and pentyl acetate (response higher in mutant, in contrast to ab3A; p=0.0385). While responses to propionic acid are small in both ab3 neurons, they are abolished in the *Ir25a²* mutant background (Kruskal–Wallis with uncorrected Dunn's comparing odorant responses to mineral oil control; ab3A p=0.3957; ab3B p=0.5184), suggesting that propionic acid detection in ab3 may be *Ir25a*-dependent.

## Co-receptor co-expression in other insect olfactory organs

To determine if co-receptor co-expression might exist in other insects besides *D. melanogaster*, we used RNA in situ hybridization to examine expression of *Orco* and *Ir25a* orthologues in the fly *D. sechellia* and in the mosquito *A. coluzzii* (*Figure 7*). *D. melanogaster* and *D. sechellia* diverged approximately 5 million years ago (*Hahn et al., 2007*), while the *Drosophila* and *Anopheles* lineages diverged nearly 260 million years ago (*Gaunt and Miles, 2002*; *Figure 7A*). Because co-receptor sequences are highly conserved, we could use our *D. mel. Orco* and *Ir25a* in situ probes (*Figure 7B*) to examine the expression of these genes in the maxillary palps of *D. sechellia*. We found widespread co-expression of *Orco* and *Ir25a* (63% of all cells were double labeled), consistent with our findings in *D. melanogaster* (*Figure 7C*). For *A. coluzzii* mosquitoes, we designed *Anopheles*-specific *Orco* and *Ir25a* probes, and examined co-receptor co-expression in antennae (*Figure 7D*) and maxillary palps (*Figure 7E*). We observed broad co-expression of *AcOrco* and *AcIr25a* in the maxillary palp capitate peg sensilla (47% of all cells were double labeled), and narrower co-expression in the antennae (25% double labeled). Co-expression results for all tissues examined are summarized in *Figure 7F*, and cell counts can be found in *Table 5*. These results suggest that *Orco* and *Ir25a* co-receptor co-expression extends to other Drosophilid species as well as mosquitoes (see also *Ye et al., 2021*; *Younger et al., 2020*).

## The co-receptor co-expression map of olfaction in *D. melanogaster*

Co-receptor co-expression of insect chemosensory receptors suggests that multiple receptors may influence the response properties of an olfactory neuron, as we have shown in ab3 and palpal sensilla. To aid future investigations of co-receptor co-expression signaling, we synthesized our results (*Table 3*) into a comprehensive new map of the AL. *Figure 8* summarizes the expression patterns of all the co-receptor knock-in lines and presents a new model for chemosensory receptor expression in *D. melanogaster*. In *Figure 8A*, the expression pattern of each knock-in line is presented separately (see also *Figure 3—source data 1*). The new AL map is updated with the recent reclassification of VP1 into three glomeruli (*Marin et al., 2020*) and indicates the new VM6 subdivisions. In *Figure 8A*, the original glomerular innervation pattern for each co-receptor is shown in green, with new innervation revealed by the *T2A-QF2* knock-in lines color coded by intensity: strongly labeled glomeruli are in orange, intermediate glomeruli in yellow, and weakly labeled glomeruli are in pink. Glomeruli labeled in <50% of brains examined are designated variable (gray), and glomeruli not labeled by the given knock-in are in white. The new VM6v, VM6m, and VM6l subdivisions are labeled with gray stripes.

In the previous model of olfaction in *Drosophila*, the Orco/OR domain primarily occupied the anterior AL, while the IR domains innervated more posterior glomeruli. While the former is still, for the most part, accurate (*Figure 8A*, *Orco*), the latter is not: both *Ir8a-T2A-QF2* and *Ir76b-T2A-QF2* label several more anterior glomeruli (such as VA3 or VA6), and *Ir25a-T2A-QF2* labels the majority of glomeruli throughout the anterior to posterior axis (*Figure 8A*, *Ir25a*). The expansion of the Ir25a+ domain is the most dramatic of the four co-receptors: previously, Ir25a+ glomeruli accounted for 21% of the AL (12/58 glomeruli) (*Enjin et al., 2016*; *Frank et al., 2017*; *Marin et al., 2020*; *Silbering et al., 2011*); the *Ir25a-T2A-QF2* knock-in consistently labels 88% of the AL (51/58 glomeruli, excluding variable). This represents a greater than fourfold expansion. Similarly, the number of Ir76b+ glomeruli increased more than threefold, from 7% of the AL (4/58 glomeruli) (*Silbering et al., 2011*) to 26% (15/58, excluding variable). The Ir8a+ domain has nearly doubled, from 17% of the AL originally (10/58 glomeruli) (*Silbering et al., 2011*) to 31% (18/58 glomeruli, excluding variable). The most modest increase in reported expression is in the Orco+ domain: from 66% of the AL (38/58 glomeruli) (*Couto et al., 2005*; *Fishilevich and Vosshall, 2005*) to 78% (45/58, excluding variable).

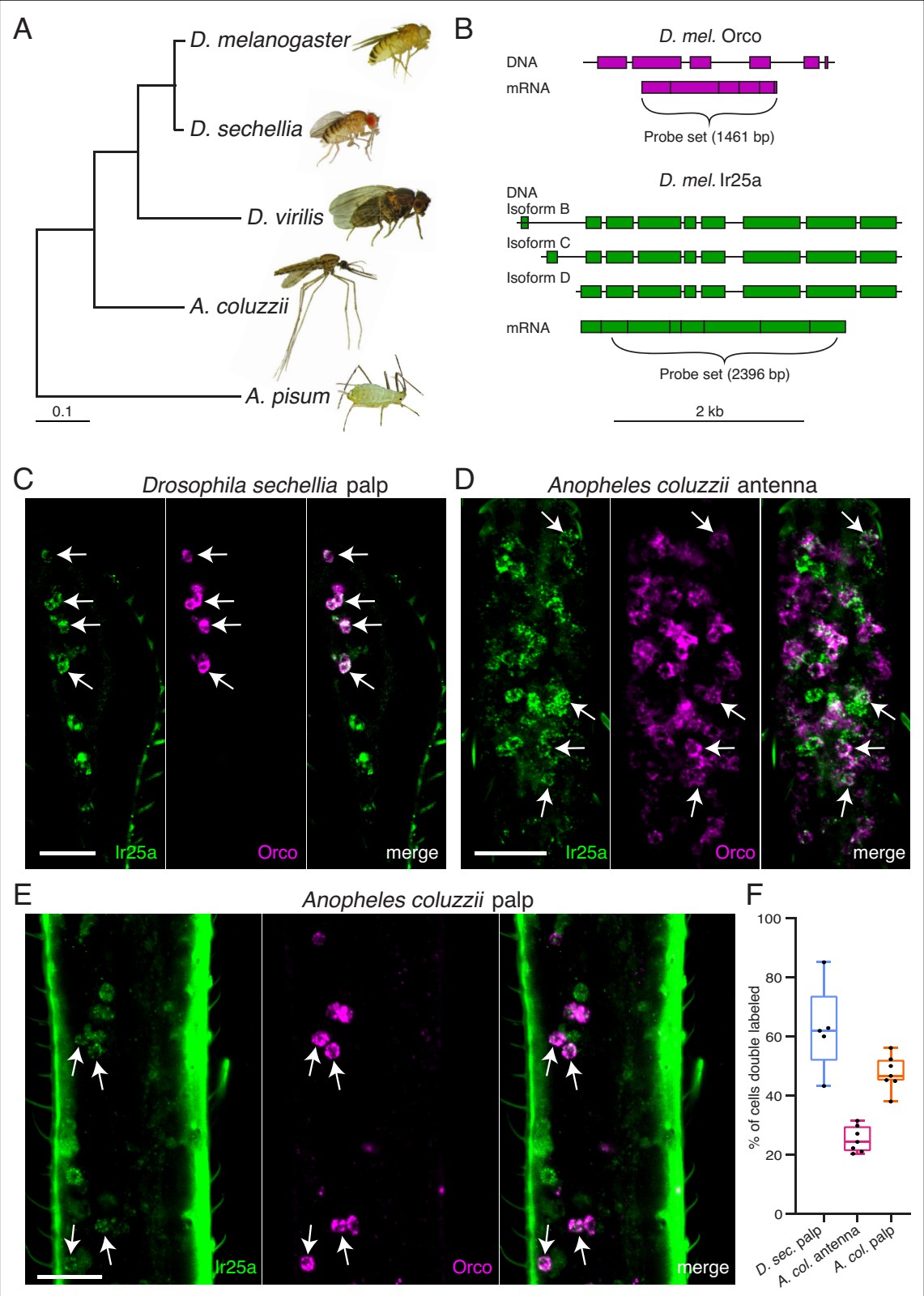

**Figure 7.** *Orco* and *Ir25a* are co-expressed in *Drosophila sechellia* and *Anopheles coluzzii* olfactory organs. (**A**) Phylogenetic tree based on the *Orco* sequences from the five insects shown (*D.* = *Drosophila*, *A. coluzzii* = *Anopheles coluzzii*, *A. pisum* = *Acyrthosiphon pisum*). Evolutionary history was inferred using the Maximum Likelihood method and Tamura–Nei model (*Tamura and Nei, 1993*). Pea aphid image **A** was reproduced from PLoS Biology Issue Image (2010). (**B**) The *Drosophila melanogaster Orco* in situ probe set, which covers the entire *Orco* coding sequence (top, magenta), was

*Figure 7 continued on next page*

*Figure 7 continued*

used to examine *Orco* expression in the maxillary palps of other *Drosophila* fly species. We designed a new probe set covering the most conserved portion of *D. mel. Ir25a* (bottom) as determined by analyzing the *Ir25a* sequences from multiple fly species and comparing them to the various *Drosophila melanogaster Ir25a* isoforms (three of which are illustrated in green). (**C**) Many olfactory sensory neurons (OSNs) in the *Drosophila sechellia* maxillary palps co-express both *Orco* and *Ir25a*, as revealed by in situ experiments (four example cells indicated with arrows). N = 5. (**D**) In situs in *Anopheles coluzzii* antennae reveal a small proportion of cells expressing both co-receptors (arrows). N = 7. (**E**) In situs in *Anopheles coluzzii* maxillary palps show many cells express both *Orco* and *Ir25a* (four examples indicated with arrows). N = 7. (**F**) Summary of co-expression analyses in (**C–E**). For each olfactory organ examined, we divided the number of Orco+ Ir25a+ double-labeled cells by the total number of cells labeled by either probe. We found that 63% of *D. sec.* palpal OSNs express both *Orco* and *Ir25a* (blue), 25% of *A. col.* antennal OSNs express both co-receptors (pink), and 47% of *A. col.* palpal OSNs are double labeled (orange). (**C–E**) are maximum intensity projections of partial z-stacks. See also ***Table 5***, and Appendix 1—key resources table.

The expression overlap in the AL of the four co-receptor families is summarized in the Venn diagram shown in ***Figure 8B*** (excluding the variably labeled glomeruli from ***Figure 8A***). The table at the right lists the names of the glomeruli that correspond to the sections of the Venn diagram. This analysis reveals nine glomeruli labeled by all four knock-in lines; furthermore, it shows that the Ir8a+ and Ir76b+ domains do not have glomeruli unique to them. Most of the AL is innervated by Orco+ Ir25a+ neurons (25 glomeruli that are only Orco+ Ir25a+, plus an additional 13 that have Orco, Ir25a, and one or both other co-receptors). The Orco+ and Ir25a+ domains reveal glomeruli unique to them (six glomeruli that are only Orco+, seven glomeruli that are only Ir25a+). Expression analyses also reveal that Ir8a does not co-express with Orco alone or Ir76b alone.

A unified AL map organized by chemosensory gene families (ORs, IRs, and GRs) is shown in ***Figure 8C*** (right panel), and the left two panels extend this information into the periphery. Here, we include the GR+innervation of the V glomerulus. However, a knock-in line for either *Gr21a* or *Gr63a* does not currently exist; thus, it is possible these receptors (as well as other poorly characterized antennal GRs) might also be more broadly expressed than previous transgenic lines indicate (***Fujii et al., 2015***; ***Menuz et al., 2014***). All four OR and IR co-receptors are expressed in the antenna, while olfactory neurons in the palps express *Orco* and *Ir25a* (***Figure 8C***, left panel). In the antennae, there are many different classes of OSNs expressing various combinations of chemosensory receptors and co-receptors: there are Orco+ only neurons (***Figure 8C***, middle panel, #2), such as those innervating the VM2 and VM3 glomeruli (teal); IrCo+ only neurons (purple), which include neurons expressing one, two, or all three *IR* co-receptors (such as VP2, VM6v, or DP1m, respectively) (***Figure 8C***, middle panel, #1); and neurons expressing both *Orco* and *IrCo(s)* (teal and purple stripe) (***Figure 8C***, middle panel, #3 and 4).

The expression data suggest that different subpopulations of olfactory neurons might be targeting a shared glomerulus. Our data indicate that both Orco+ and Ir25a+ neurons innervate the GR+ V glomerulus (dark blue; see also ***Figure 8A***). Based on the sparse innervation of the V glomerulus by the *Orco-T2A-QF2* knock-in (***Figure 3A***) and the lower expression levels in the snRNAseq data (***Figure 4A***), we hypothesize that *Orco* may be expressed in only a subset of Gr21a/Gr63a+ neurons. This contrasts with the *Ir25a-T2A-QF2* knock-in, which appears to label most Gr21a/Gr63a+ neurons. Thus, two subpopulations of neurons may be co-converging upon the same V glomerulus: neurons that express *Gr21a/Gr63a* and *Ir25a* (dark blue and purple stripes), and neurons that express *Gr21a/Gr63a*, *Ir25a*, and *Orco* (dark blue, purple, and teal stripes) (***Figure 8C***, middle panel, #5). Such co-convergence has recently been shown in the olfactory system of *Aedes aegypti* mosquitoes (***Younger et al., 2020***). Similarly, the sparse *Orco-T2A-QF2* knock-in innervation of the DP1l glomerulus suggests that there are OSN populations expressing mostly *IRs*, but with a subset of neurons that additionally express *Orco* (***Figure 8C***, middle panel, #3). The converse may also be possible (***Figure 8C***, middle panel, #4): OSN populations that have some neurons expressing only *Orco*, and a subset expressing both *Orco* and *IrCo(s)* co-converging onto the same glomerulus. There is some evidence for this in the palps, based on our anti-Orco and anti-Ir25a antibody staining (***Figure 2C and I***, ***Figure 4B***, ***Table 2***). The snRNAseq data suggest that this may also be the case in the antennae (see ***Figure 4—source data 1***).

**Table 5.** Co-expression of *Orco* and *Ir25a* in non-*melanogaster* insect olfactory organs (related to *Figure 7*).
Whole-mount palps from *Drosophila sechellia* flies, and whole-mount antennae and palps from *Anopheles coluzzii* mosquitoes, were examined using fluorescence in situ hybridization with probe sets against *Orco* and *Ir25a*. Co-expression between Orco and Ir25a co-receptors was observed in both insects, with *D. sec.* palps having the highest degree of co-expression (63% of cells double labeled) and *A. col.* antennae having the lowest (25% of cells double labeled). N = 5 for *D. sec.* and 7 for *A. col.*

| Species | Sample | Orco+ cells | Ir25a+ cells | Double-labeled cells | Total cells |
|---|---|---|---|---|---|
| *Drosophila sechellia* | Palp 1 | 70 | 44 | 44 | 70 |
| *Drosophila sechellia* | Palp 2 | 64 | 48 | 42 | 70 |
| *Drosophila sechellia* | Palp 3 | 63 | 39 | 39 | 63 |
| *Drosophila sechellia* | Palp 4 | 78 | 38 | 35 | 81 |
| *Drosophila sechellia* | Palp 5 | 86 | 75 | 74 | 87 |
| | Total across samples: 361 | | 244 | 234 | 371 |
| | | Proportion of Orco+ cells that are Ir25a+: | Proportion of Ir25a+ cells that are Orco+: | Proportion of all cells that are double labeled: | |
| | | 0.65 | 0.96 | 0.63 | |
| *Anopheles coluzzii* | Antenna 1 | 52 | 17 | 12 | 57 |
| *Anopheles coluzzii* | Antenna 2 | 47 | 24 | 17 | 54 |
| *Anopheles coluzzii* | Antenna 3 | 57 | 20 | 13 | 64 |
| *Anopheles coluzzii* | Antenna 4 | 50 | 27 | 14 | 63 |
| *Anopheles coluzzii* | Antenna 5 | 62 | 30 | 18 | 74 |
| *Anopheles coluzzii* | Antenna 6 | 53 | 21 | 17 | 57 |
| *Anopheles coluzzii* | Antenna 7 | 49 | 26 | 16 | 59 |
| | Total across samples: 370 | | 165 | 107 | 428 |
| | | Proportion of Orco+ cells that are Ir25a+: | Proportion of Ir25a+ cells that are Orco+: | Proportion of all cells that are double labeled: | |
| | | 0.29 | 0.65 | 0.25 | |
| *Anopheles coluzzii* | Palp 1 | 34 | 30 | 23 | 41 |
| *Anopheles coluzzii* | Palp 2 | 30 | 36 | 22 | 44 |
| *Anopheles coluzzii* | Palp 3 | 26 | 35 | 19 | 42 |
| *Anopheles coluzzii* | Palp 4 | 35 | 34 | 19 | 50 |
| *Anopheles coluzzii* | Palp 5 | 32 | 39 | 22 | 49 |
| *Anopheles coluzzii* | Palp 6 | 31 | 35 | 21 | 45 |
| *Anopheles coluzzii* | Palp 7 | 30 | 37 | 23 | 44 |
| | Total across samples: 218 | | 246 | 149 | 315 |
| | | Proportion of Orco+ cells that are Ir25a+: | Proportion of Ir25a+ cells that are Orco+: | Proportion of all cells that are double labeled: | |
| | | 0.68 | 0.61 | 0.47 | |

## Discussion

Here, we present evidence of widespread chemosensory co-receptor co-expression in the olfactory system of *D. melanogaster*, contrasting a previously accepted view of segregated, mutually exclusive olfactory domains. By generating targeted knock-ins of the four main chemosensory co-receptor genes (*Orco, Ir8a, Ir76b, Ir25a*), we demonstrate that all four co-receptors have broader olfactory neuron expression than previously appreciated. The *Ir25a* co-receptor was previously thought to be expressed only in a small subset of olfactory neurons (coeloconic, sacculus, and arista neurons), but we present evidence that it is expressed in subsets of all sensilla classes and in OSNs that innervate the majority of the fly's AL glomeruli. We further find that the *Ir25a* co-receptor may be involved in

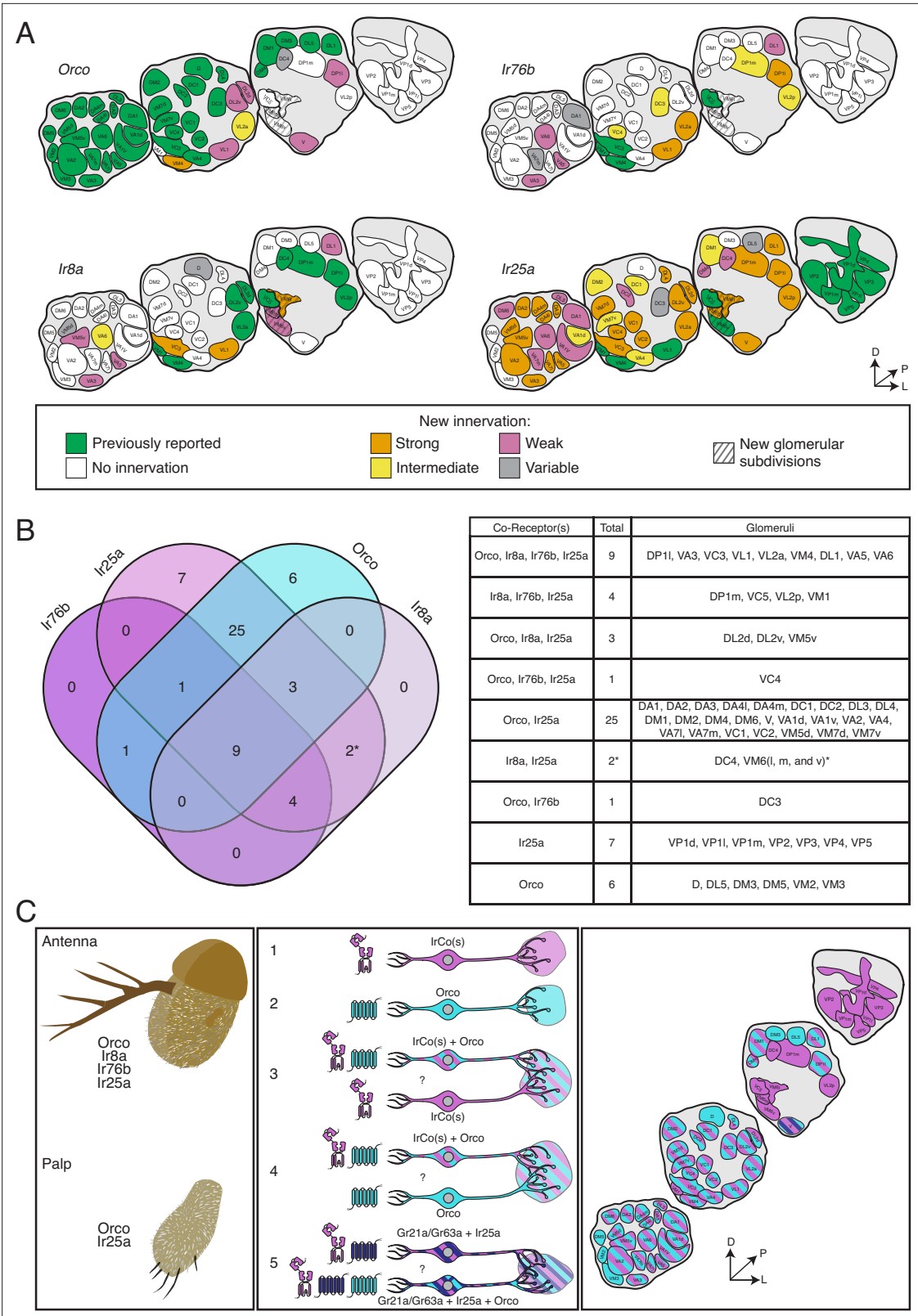

**Figure 8.** The co-receptor co-expression map of olfaction in *Drosophila melanogaster*. (**A**) Summary of antennal lobe (AL) expression for all co-receptor knock-in lines (from all brains examined in *Figures 3–5*; *Orco* N = 8, *Ir8a* N = 15, *Ir76b* N = 11, *Ir25a* N = 15). The previously reported innervation pattern for each co-receptor is shown in green; new innervation reported here is color-coded according to strength of glomerular labeling, from strong (orange), to intermediate (yellow), to weak (pink). Glomeruli labeled in <50% of brains examined for a given knock-in line are designated variable (gray); glomeruli

*Figure 8 continued on next page*

*Figure 8 continued*

not labeled are white. The novel VM6 glomerular subdivisions reported here are indicated by gray stripes. (**B**) Overlap of chemosensory modalities in the AL. In the Venn diagram (left), *IR* co-receptors are color-coded in shades of purple, while *Orco* is in teal, as in *Figure 1*. Numbers indicate how many glomeruli are found in the given intersection of co-receptors out of 58 total glomeruli. Variably labeled glomeruli were excluded from these analyses. The table lists the names of the glomeruli in each section of the Venn diagram. The new glomerular subdivisions are indicated with an asterisk. (**C**) New view of olfaction in *Drosophila*. Left: in the periphery, all four co-receptors are expressed in the antenna (top), while palpal neurons express *Orco* and *Ir25a* (bottom). Middle: many different classes of olfactory sensory neurons (OSNs) express various combinations of chemosensory receptors and co-receptors. While some neurons express only *IrCos* (purple, #1) or *Orco* (teal, #2), many neurons co-express these chemoreceptors (indicated with striped fill, #3 and 4). Within the latter group, there may be OSN populations in which *IRs* are the dominant receptors, and *OR* expression is sparse (#3), and other populations where *ORs* are the primary receptors and *IR* expression is infrequent (#4). GR+ neurons (dark blue) also express *Ir25a* (#5, dark blue and purple striped fill), and some of these neurons additionally express *Orco* (#5, dark blue, purple, and teal striped fill). Question marks indicate potential instances of co-convergence of different subtypes of OSNs onto the same glomeruli. Right: a comprehensive map of the antennal lobe shows that most glomeruli are innervated by OSNs that co-express multiple chemoreceptors. Compass in (**A**) and (**C**): D = dorsal, L = lateral, P = posterior. See also *Table 3* and *Figure 3—source data 1* and *Figure 3—source data 2*.

modulating the activity of some Orco+ OSNs, both in the antennae and maxillary palps. We present a new AL map that will aid future inquiries into the role that specific chemoreceptor co-expression plays in distinct OSN populations.

Based on the co-receptor innervation patterns in the ALs, we identified a glomerulus, VM6, that is uniquely partitioned by different OSNs (*Figure 5*; also see *Schlegel et al., 2021*). The co-receptor expression patterns allowed us to pinpoint the likely origin of the innervating OSNs. Since the VM6 glomerulus was labeled by both the *Ir25a-T2A-QF2* and *Ir8a-T2A-QF2* knock-in lines, the cell bodies of these neurons had to reside in the antenna; furthermore, since we did not find *Ir8a-T2A-QF2* labeling of the arista, these neurons were likely to be either in coeloconic sensilla or in the sacculus. Indeed, we determined the VM6 glomerulus to be innervated by the newly discovered Rh50+ Amt+ olfactory neurons that reside in the sacculus and ac1 sensilla (*Vulpe et al., 2021*). Based on our results, Rh50+ Amt+ sacculus neurons are further subdivided into those that strongly express Ir8a, which innervate the VM6l region, and those that weakly or infrequently express Ir8a, which innervate VM6m and VM6v. The functional consequences of this unusual subdivision by olfactory neurons for a glomerulus, and how this relates to the fly's olfactory perception of ammonia or other odorants, remain to be determined. These results also highlight the value of exploring chemosensory receptor expression patterns using knock-in lines even in the era of connectomics as the VM6 glomerulus and its subdivisions were not easily identifiable in prior electron microscopy reconstructions of the entire AL (*Bates et al., 2020*; *Scheffer et al., 2020*; *Schlegel et al., 2021*).

A model for OR/IR segregation was initially supported by developmental evidence. Two pro-neural genes specify the development of sensory structures on the antennae: *amos* and *atonal*. *Amos* mutants lack basiconic and trichoid sensilla, while *atonal* mutants do not develop coeloconic sensilla, the arista, or the sacculus (*Goulding et al., 2000*; *Gupta and Rodrigues, 1997*; *Jhaveri et al., 2000a*; *Jhaveri et al., 2000b*). It was observed that *amos* mutants lose *Orco* expression while retaining *Ir25a* expression (*Benton et al., 2009*). Our results generally do not conflict with this view. In the traditional segregated model of the fly olfactory system, it was presumed that *atonal* mutant antennae would show the reverse pattern: loss of *IrCo* expression but not *Orco* expression. However, our co-receptor knock-in expression results suggest that *atonal* mutants should have significant *IrCo* expression, particularly of *Ir25a*. This was indeed found to be the case in RNAseq analyses performed on *atonal* mutant antennae, which showed that both *Ir25a* and *Ir76b* expression, but not *Ir8a* expression, remained (*Menuz et al., 2014*). Based upon the strength of the corresponding glomerular innervations, it does appear that the previously reported Ir25a+ neurons have stronger or more consistent *Ir25a* expression, while the new Ir25a+ olfactory neurons in the antennae reported here (e.g., *OR*-expressing OSNs) are often weakly or stochastically labeled. This might also explain why *Ir25a* expression was initially overlooked in these Orco+ neural populations. The developmental pattern is different in the maxillary palps, where it is *atonal* and not *amos*, which is required for the development of the basiconic sensilla (*Gupta and Rodrigues, 1997*). Interestingly, the *Ir25a* knock-in expression corresponds well with the *atonal* developmental program across the olfactory appendages: the strongest expression of *Ir25a* is in coeloconic sensilla, but outside of these sensilla the strongest and most consistent *Ir25a* expression is in palp basiconic sensilla.

Chemosensory co-receptor co-expression may help clarify previously confounding observations regarding *D. melanogaster* odor coding. For example, olfactory neurons in the sacculus that express the *Ir64a* tuning receptor along with the *Ir8a* co-receptor project to two different glomeruli: DC4 and DP1m (*Ai et al., 2013*; *Ai et al., 2010*). These two glomeruli exhibit different olfactory response profiles, with DC4 being narrowly tuned to acids or protons, while DP1m is more broadly tuned. The molecular mechanism for these differences was previously unclear. However, the co-receptor knock-in data presented here reveals that the Ir64a+ OSN subpopulations express different combinations of co-receptors: in addition to the *Ir8a* co-receptor, neurons innervating DC4 express *Ir25a* (and occasionally *Orco*) (see *Figure 8*). In contrast, neurons innervating DP1m express *Ir8a*, *Ir25a*, and *Ir76b*. Thus, perhaps it is *Ir76b* expression in DP1m-targeting Ir64a+ neurons that makes them olfactory generalists. This idea is supported by experiments in which *Ir64a* was misexpressed in neurons targeting the VM4 glomerulus, conferring a DP1m-like, rather than a DC4-like, response profile to VM4 (*Ai et al., 2010*). We show here that VM4-targeting neurons express *Ir8a*, in addition to *Ir25a* and *Ir76b* (as well as *Orco*): thus, molecularly, the VM4 neuron profile is more similar to DP1m (co-expressing all *IrCos*) than DC4 (co-expressing two *IrCos*), and the key distinguishing component appears to be *Ir76b*. It would be interesting to repeat such misexpression experiments in an *Ir76b⁻* mutant background to test this hypothesis.

While we demonstrate here that multiple chemosensory co-receptors can be co-expressed in the same olfactory neurons, it remains to be determined if this also applies to tuning (odor-binding) receptors. Previous studies suggest that OrX tuning receptors are generally limited to a single class of olfactory neurons (*Couto et al., 2005*; *Fishilevich and Vosshall, 2005*). However, many IrX tuning receptors remain to be fully characterized and could be co-expressed in multiple olfactory neurons. For example, recordings from ab1, ab3, and ab6 sensilla indicate responses to the typical IR odors 1,4-diaminobutane, ammonia, and butyric acid, respectively (*de Bruyne et al., 2001*), suggesting that tuning IrXs may be involved. We show that *Ir25a* plays a functional role in Orco+ neurons in the antennae and palps; this suggests that these Orco+ neurons could also express as yet unidentified ligand binding IrXs. The recent release of the whole fly single-cell atlas, which includes RNAseq data from maxillary palps, allowed us to identify six IRs that might be expressed in palpal OSNs (*Ir40a*, *Ir51a*, *Ir60a*, *Ir62a*, *Ir76a*, *Ir93a*) (*Li et al., 2021*). However, in situ analyses for these six IRs in the maxillary palps did not detect a signal (see *Figure 6—figure supplement 2* and Appendix 1—key resources table). This suggests that *Ir25a* in the palps may be playing a role independent of its role as a co-receptor, as discussed further below, or that a tuning IrX was missed by the RNAseq analyses. In antennal ab3 sensilla, we did find one odorant (propionic acid) that elicited a small response in *wild-type* neurons and no response in *Ir25a²* mutant neurons. It is possible that other antennal Orco+ OSNs might utilize IR chemoreceptors for signaling. For example, the ac3B neuron, which expresses Or35a/Orco and all IR co-receptors, has recently been suggested to utilize an unidentified IrX to mediate responses to phenethylamine (*Vulpe and Menuz, 2021*). The chemoreceptor expression patterns revealed in this work will help the search for olfactory neurons that may utilize multiple chemosensory families for odor detection.

The widespread expression of *Ir25a* in the fly olfactory system raises the possibility that it might have roles in addition to its function as an IrX co-receptor. For example, *Ir25a* has been found to play a developmental role in forming the unique structure of Cold Cells in the arista (*Budelli et al., 2019*). Evolutionary studies also suggest that *Ir25a* is the most ancient of all the insect chemosensory receptors (*Croset et al., 2010*), and the currently broad expression might reflect its previous ubiquitous role in chemosensory signaling.

Co-expression of chemosensory co-receptors might function to increase the signaling capabilities of an olfactory neuron. For example, the signaling of an Orco+ olfactory neuron may be guided primarily by the tuning OrX, and the sensitivity range extended to include odors detectable by an IrX. Co-expression might also allow synergism, such that weak activation of a co-expressed receptor could increase neuronal activity to levels sufficient to drive behavior. This might be useful in tuning behavioral response to complex odors, such that certain combinations of odors lead to stronger olfactory neuron responses. Alternatively, a co-expressed receptor inhibited by odorants might be able to attenuate a neuron's response to odor mixtures. The observed broad *Ir25a* co-expression might allow an Orco-positive olfactory neuron to be primed to express a functional IrX/Ir25a receptor complex. As suggested above, this could be an evolutionary advantage if the co-expressed IrX receptor improved

olfactory responses to a complex but crucial biologically relevant odor, such as host-seeking cues as observed in the *A. aegypti* mosquito olfactory system (*Younger et al., 2020*). Co-expression of chemosensory receptors could thereby be a mechanism to increase the functional flexibility of a numerically limited olfactory system.

*Ir25a* expression might further modulate chemosensory neuron activity levels driven by Orco/OrX signaling by altering membrane resistance. This might explain the modest activity changes we observed in *Ir25a* mutant Orco-expressing neurons (*Figure 6*). In this manner, altering Ir25a expression levels could be a neuronal mechanism to adjust Orco/OrX activity. Alternatively, *Ir25a* may contribute to olfactory signal transduction or amplification as has recently been shown for a pickpocket ion channel (*Ng et al., 2019*). Experiments addressing potentially expanded roles for *Ir25a* in olfactory neurons will be aided by the new chemosensory co-receptor map presented here.

*D. melanogaster* often serves as a model for many other insect olfactory systems, and information gleaned from vinegar flies is frequently extrapolated to other insects (e.g., *DeGennaro et al., 2013*; *Fandino et al., 2019*; *Riabinina et al., 2016*; *Trible et al., 2017*; *Yan et al., 2017*). Indeed, prompted by our findings of *Orco* and *Ir25a* co-expression in *D. melanogaster*, we extended our observations to two additional insect species. Using in situ hybridization, we found that olfactory neurons in the palps of *D. sechellia* flies, and in the antennae and palps of *A. coluzzii* mosquitoes, also co-express *Orco* and *Ir25a* co-receptors. The work presented here raises the possibility that many insects may also exhibit co-expression of chemosensory co-receptors. Recent work in *A. aegypti* mosquitoes suggests this is indeed the case: *A. aegypti* mosquito olfactory neurons can co-express Orco/IrCo/Gr receptors (*Younger et al., 2020*). Furthermore, *A. coluzzii* mosquitoes have recently been shown to co-express *Orco* and *Ir76b* co-receptors in their olfactory organs (*Ye et al., 2021*). This suggests that co-expression of chemosensory co-receptors may be an important feature of insect olfactory neurons.

## Materials and methods
### Key resources table
See Appendices 1 and 2.

### Resource availability
#### Lead contact
Further information and requests for resources and reagents should be directed to and will be fulfilled by the lead contact, Christopher J. Potter (cpotter@jhmi.edu).

### Materials availability
Fly lines generated in this study have been deposited to the Bloomington Drosophila Stock Center.

### Data and code availability
The snRNAseq dataset analyzed in this paper is published in *McLaughlin et al., 2021*. Sequencing reads and preprocessed sequencing data are available in the NCBI Gene Expression Omnibus (GSE162121). Python code for generating figures is publicly available on GitHub (https://github.com/colleen-mclaughlin/ORN_seq/). VM6 reconstructions using FlyWire can be viewed at https://flywire.ai/#links/Task2021a/all. Raw data used to generate each figure is available at Johns Hopkins Research Data Repository (https://doi.org/10.7281/T1/9VJGPI ).

### Experimental model and subject details
#### Fly husbandry
Fly stocks were maintained at 20–25°C on standard cornmeal-agar food. Male and female flies used for experiments were 3–11 days old, unless otherwise noted.

#### Fly stocks
Fly lines used in this paper can be found in the Appendix 1—key resources table.

While performing co-labeling experiments, we discovered that several *OrX-Gal4* lines label multiple glomeruli, and thus do not accurately represent single OSN classes. These lines were excluded from

analyses and should be used with caution: *Or33c-Gal4* (BDSC# 9966), *Or42a-Gal4* (BDSC# 9970), *Or43b-Gal4* (BDSC# 23894), *Or59b-Gal4* (BDSC# 23897), *Or65a-Gal4* (BDSC# 9994), *Or85a-Gal4* (BDSC# 23133), and *Or85b-Gal4* (BDSC# 23911). We also found that the following *Or35a* lines label the newly identified VM6l glomerulus in addition to VC3: *Or35a-Gal4* (BDSC# 9967), *Or35a-Gal4* (BDSC# 9968), *Or35a-mCD8.GFP* (BDSC# 52624), as well as an *Or35a-Gal4* line from the Carlson lab (*Yao et al., 2005*).

D. *sechellia* flies (strain: Cousin Island, Seychelles; SKU: 14021-0248.25) were obtained from the National *Drosophila* Species Stock Center (Cornell College of Agriculture and Life Sciences) and reared according to our *D. melanogaster* protocol.

## Mosquito husbandry

*A. gambiae* mosquitoes were reared as previously described (*Riabinina et al., 2016*). The *wildtype* N'Gousso strain was a gift from the Insect Transformation Facility in Rockville, Maryland.

## Generation of *QUAS-CsChrimson*

The sequence of CsChrimson.Venus was PCR amplified from the genomic DNA of *UAS-CsChrimson. mVenus* flies (*Klapoetke et al., 2014*) and cloned into the *10XQUAS* vector (Addgene #163629). A fly line was established through random *P*-element insertion. Cloning was confirmed with Sanger sequencing (Genewiz) before being sent for injection (Rainbow Transgenic Flies, Inc). Primers used for PCR amplification and In-Fusion cloning:

IVS-FOR:
TGGGTTGGACTCAGGGAATAGATCTAAAAGGTAGGTTCAACCACT
EcoRI-SV40-REV:
GCTTACGTCAGAATTCAGATCGATCCAGACATGATAAGA

## Generation of HACK knock-in lines

The HACK knock-in approach requires two components: a donor construct and Cas9 (*Lin and Potter, 2016a*). The donor includes *gRNAs* specific to the target gene, as well as the template for HDR-mediated insertion of *T2A-QF2* into the genome (*Figure 2A*, middle row). This template includes ~1 kb homology arms directly up- and downstream of the gene's stop codon flanking a cassette containing *T2A-QF2* and a *3XP3-mCherry* fluorescent eye marker (see *Figure 2—figure supplement 1D and E* and *Figure 2—figure supplement 3A and B*). Outside of these homology arms, the construct has two *RNA polymerase III U6* promoters driving independent expression of two *gRNAs* specific to the region around the target gene's stop codon (*Port et al., 2014*). Two *gRNAs* were used to increase the probability of successfully inducing double-stranded breaks in the target (*Port et al., 2014*). The knock-in construct replaces the target gene's stop codon (*Figure 2A*, bottom row) and introduces a transcriptional stop at the end of *QF2*.

The donor construct can be supplied in one of two ways (*Figure 2—figure supplement 1*). The first is to inject the HACK construct directly into embryos expressing Cas9 in their germline (direct injection method) (*Figure 2—figure supplement 1A and B*). The second approach is to establish transgenic donor lines through random *P*-element insertion or *ΦC31* integration (*Bischof et al., 2007*; *Gloor et al., 1991*; *Groth et al., 2004*) of the construct into the genome, followed by genetic crosses with germline Cas9 flies for the generation of the knock-in (cross method) (*Figure 2—figure supplement 1C*). Only one (direct injection method) or two (cross method) generations of crosses are required for the creation of a knock-in line (*Figure 2—figure supplement 1B and C*). The HACK 3XP3-mCherry selection marker is bright but shows positional effects (*Figure 2—figure supplement 3A*). Potential knock-in flies can be screened at the adult stage (*Figure 2—figure supplement 3A*), or at the larval or pupal stages (*Figure 2—figure supplement 1D*). We generated *T2A-QF2* knock-in lines for all four co-receptor genes using the direct injection method. Additionally, we tested the feasibility of the cross approach with two genes: *Orco* and *Ir25a*. Knock-ins were confirmed by PCR genotyping and sequencing (*Figure 2—figure supplement 3B–D*), and by crosses to a *QUAS-GFP* reporter to check for expression in the brain (*QUAS-mCD8::GFP* was used only to establish the *Orco-T2A-QF2* knock-in line, after which the reporter was removed via genetic crosses; for all AL analyses, we used the

*10XQUAS-6XGFP* reporter line). We found no difference in expression pattern in the brain between these two approaches (*Figure 2—figure supplement 1G*). After establishing a knock-in line, the 3XP3-mCherry marker can be removed via *Cre* recombination (*Siegal and Hartl, 1996*). This can be useful as 3XP3-mCherry is expressed broadly throughout the fly nervous system and can interfere with red fluorescent reporters (*Figure 2—figure supplement 1E*). We produced two unmarked knock-in lines (for *Orco* and *Ir25a*) and confirmed no difference in brain GFP expression between marked and unmarked lines (*Figure 2—figure supplement 1F*).

Both approaches produced knock-ins at high rates (*Table 1*). Efficiency was calculated as the number of potentially HACKed knock-in flies (mCherry+), divided by the total number of flies from the given cross ($G_1$ or $F_2$ progeny; see *Figure 2—figure supplement 1A–C*). We further calculated the percentage of founders producing knock-in lines as this gives an indication of effort (how many initial crosses need to be set up to produce a knock-in). The aggregate efficiency rates for a given target locus ranged from 8% for Ir8a to 33% for Orco (*Table 1*); however, for individual crosses, efficiency rates were as high as 100% (see *Table 1—source data 1*), meaning that all progeny were potential mCherry+ knock-ins. For the two genes for which we created knock-in lines via both direct injection and genetic cross (*Orco* and *Ir25a*), we found efficiency rates comparable between approaches (*Orco*: 33% for direct injection, 28% for cross; *Ir25a*: 23% for direct injection, 24% for cross). For the direct injection approach, we tested 51 independent knock-in lines across the four target genes and found 100% to be correctly targeted events (*Table 1*). However, for the genetic cross approach, of the 32 independent knock-in lines tested for the two target genes, 6 (~19%) had the HACK mCherry eye marker but did not have *QF2*-driven GFP expression in the brain.

Information on plasmid construction can be found in the 'Method details' section. All *D. melanogaster* embryo injections were performed by Rainbow Transgenic Flies, Inc (Camarillo, CA). For HACKing via genetic cross, *Orco-T2A-QF2* and *Ir25a-T2A-QF2* constructs were injected into $w^{1118}$ flies for *P*-element insertion, and donor lines were established on the second or third chromosomes by crossing to double balancers (see Appendix 1—key resources table). Donor lines were then crossed to *Vas-Cas9* (BDSC# 51323). Knock-in lines were established from *cis*-chromosomal HACK (donor line on same chromosome as target gene) (*Lin and Potter, 2016a*). For HACKing via direct injection, knock-in constructs were injected into the following lines: *Vas-Cas9* (BDSC# 51324) for *Ir8a; Act5C-Cas9* (BDSC# 54590) for *Orco*, *Ir76b*, and *Ir25a*. The following lines were used to verify knock-in expression: *QUAS-mCD8::GFP* (BDSC# 30003), *10XQUAS-6XGFP* (BDSC# 52264). Knock-in lines were confirmed by PCR genotyping (Phusion, NEB) and Sanger sequencing (Genewiz). Unmarked *Orco-T2A-QF2* and *Ir25a-T2A-QF2* knock-in lines were generated by crossing mCherry+ knock-in flies to the Crey+ 1B fly line (see Appendix 1—key resources table; *Siegal and Hartl, 1996*).

To investigate the effect of *T2A-QF2* knock-in on gene function, we performed SSR on homozygous flies for each co-receptor knock-in (*Figure 2—figure supplement 2*), comparing their responses to *wildtype* flies to panels of Orco- and Ir-dependent odorants (*Abuin et al., 2011*; *de Bruyne et al., 2001*; *Lin and Potter, 2015*). In ab2 basiconic sensilla, *Orco-T2A-QF2* knock-in flies had slightly lower baseline activity as compared to *wildtype* (*Figure 2—figure supplement 2A*); however, there were no significant differences in odor-evoked activity between these two genotypes across all stimuli tested (*Figure 2—figure supplement 2B*). In ac2 coeloconic sensilla, responses of *Ir8a-T2A-QF2* knock-in flies to hexanol and cadaverine were slightly lower than *wildtype* (*Figure 2—figure supplement 2D*); however, these are not typically considered Ir8a-dependent odorants. Responses of the *Ir8a-T2A-QF2* knock-in to Ir8a-dependent odorants (*Abuin et al., 2011*) were similar to *wildtype* controls (example trace in *Figure 2—figure supplement 2C*, quantification in *Figure 2—figure supplement 2D*). Responses of *Ir76b-T2A-QF2* knock-in ac2 neurons to phenethylamine and acetic acid differed slightly from *wildtype* controls (*Figure 2—figure supplement 2E and F*). The reasons for this are unclear. The largest difference in responses between a knock-in and *wildtype* were for *Ir25a-T2A-QF2* (*Figure 2—figure supplement 2G and H*); the knock-in has significantly reduced or abolished responses to Ir25a-dependent odorants, recapitulating an *Ir25a* mutant phenotype (*Abuin et al., 2011*; *Silbering et al., 2011*; see also *Figure 2—source data 1*).

## Method details

### Plasmid construction

The construction of *QF2^X-HACK* knock-in plasmids requires three steps of cloning, as previously described (***Lin and Potter, 2016a***). All knock-in constructs were created using the *pHACK-QF2* plasmid (Addgene #80274) as the backbone. The backbone was digested with the following enzymes for cloning: *MluI* for the 5′ homology arms; *SpeI* for the 3′ homology arms; and *BbsI* for the gRNAs. All cloning was performed using In-Fusion Cloning (Clontech #639645). The homology arms were PCR-amplified from genomic DNA extracted from *wildtype* flies using the DNeasy Blood and Tissue Kit (QIAGEN #69506), while the gRNAs were PCR-amplified using the *pHACK-QF2* backbone as a template, with the primers themselves containing the gRNA target sequences. All homology arms were approximately 1 kb (Orco: 5HA = 1012 bp, 3HA = 1027 bp; Ir8a: 5HA = 1027 bp, 3HA = 1079 bp; Ir76b: 5HA = 997 bp, 3HA = 956 bp; Ir25a: 5HA = 1119 bp, 3HA = 990 bp). gRNAs were selected by analyzing the region around the stop codon of each gene using an online tool (https://flycrispr.org/ ***Gratz et al., 2014***). When possible, gRNAs were chosen to minimize potential off-target cleavage sites (zero predicted for *Orco*, *Ir8a*, and *Ir76b*; one predicted for *Ir25a*, discussed below). They were selected such that one gRNA targeted upstream of the stop codon, within the last exon of the gene; the second gRNA targeted downstream of the stop codon, within the 3′UTR; and the two gRNAs were <100 bp apart. In order to prevent the gRNAs from targeting the homology arms, three synonymous nucleotide substitutions were made in each homology arm. The final knock-in lines did not always have all three substitutions (see ***Figure 2—figure supplement 3D***), possibly due to PCR or HDR error. Note that due to the way the primers are designed, each targeted gene loses a small portion of its native 3′UTR (Orco = 72 bp, Ir8a = 31 bp, Ir76b = 27 bp, Ir25a = 24 bp). Cloning was confirmed with Sanger sequencing (Genewiz) before being sent for injection (Rainbow Transgenic Flies, Inc). Below are the gRNAs used for each gene, with the PAM sequence in parentheses.

> Orco:
> CTGCACCAGCACCATAAAGT (AGG)
> GCACAGTGCGGAGGGGGCAA (GGG)
> Ir8a:
> GTTTGTTTGTTCGGCCATGT (TGG)
> GGTGCCTCTGACTCCCACAG (TGG)
> Ir76b:
> GCAGTGATGCGAACTTCATA (TGG)
> GTATTGAAAGAGGGCCGCCG (AGG)
> Ir25a:
> GCCGGATACTGATTAAAGCG (CGG)
> ATTATGGTAAAATGAGCACT (CGG)

### Primers

*Italics* = In-Fusion Cloning 15 bp overhang; **bold** = gRNA; lowercase = adding back restriction site; underline = synonymous substitution to prevent Cas9 targeting of donor construct.

### PCR primers for cloning

> Orco_gRNA_FOR:
> *TCCGGGTGAACTTC***GCACAGTGCGGAGGGGGCAA**GTTTTAGAGCTAGAAATAGCAAGTTA
> Orco_gRNA_REV:
> *TTCTAGCTCTAAAAC***ACTTTATGGTGCTGGTGCAG**CGACGTTAAATTGAAAATAGGTC
> Orco_5HA_FOR:
> *CCCTTACGTAACGCG*tCAGCTTGTTTGACTTACTTGATTAC
> Orco_5HA_REV:
> *CGCGGCCCTCACGCG*tCTTGAGCTGT<u>A</u>C<u>A</u>AG<u>T</u>ACCATAAAGT
> Orco_3HA_FOR:
> *GTTATAGATCACTAG*tCTC<u>A</u>GT<u>A</u>CT<u>A</u>TGCAACCAGCAATA
> Orco_3HA_REV:

*AATTCAGATCACTAG*tGTTTTATGAAAGCTGCAAGAAATAA

Ir8a_gRNA_FOR:
*TCCGGGTGAACTTCG***TTTGTTTGTTCGGCCATGT**GTTTTAGAGCTAGAAATAGCAAGTTA

Ir8a_gRNA_REV:
*TTCTAGCTCTAAAAC***CTGTGGGAGTCAGAGGCAC**CGACGTTAAATTGAAAATAGGTC

Ir8a_5HA_FOR:
*CCCTTACGTAACGCG*tCTATTGGCTATTCGTCGTACTCATGC

Ir8a_5HA_REV:
*CGCGGCCCTCACGCG*tCTCCATGTAGCCACT<u>A</u>TG<u>T</u>GAGTCAGA<u>T</u>

Ir8a_3HA_FOR:
*GTTATAGATCACTAG*tGTTTCTTGTCGCACCTAATTAACAAGTG

Ir8a_3HA_REV:
*AATTCAGATCACTAG*tCATACTTAAGCTCCTTGAGGTCCAGC

Ir76b_gRNA_FOR:
*TCCGGGTGAACTTCG***CAGTGATGCGAACTTCATA**GTTTTAGAGCTAGAAATAGCAAGTTA

Ir76b_gRNA_REV:
*TTCTAGCTCTAAAAC***CGGCGGCCCTCTTTCAATAC**GACGTTAAATTGAAAATAGGTC

Ir76b_5HA_FOR:
*CCCTTACGTAACGCG*tACCAATGAATCCTTGTCCATGCTAAA

Ir76b_5HA_REV:
*CGCGGCCCTCACGCG*tCTCGGTGTAGCTGT<u>C</u>TTGAA<u>GGA</u><u>A</u>

Ir76b_3HA_FOR:
*GTTATAGATCACTAG*tGCCTAATTGGAATACCTTCTACATAATGGA

Ir76b_3HA_REV:
*AATTCAGATCACTAG*tGGCAAGGCACAAAATAAAACGAAG

Ir25a_gRNA_FOR:
*TCCGGGTGAACTTC***GCCGGATACTGATTAAAGCG**GTTTTAGAGCTAGAAATAGCAAGTTA

Ir25a_gRNA_REV:
*TTCTAGCTCTAAAAC***AGTGCTCATTTTACCATAAT**CGACGTTAAATTGAAAATAGGTC

Ir25a_5HA_FOR:
*CCCTTACGTAACGCG*tTGCATGACTTCATTGACACCTCAAG

Ir25a_5HA_REV:
*CGCGGCCCTCACGCG*tGAAACGAGGCTTAAACGT<u>A</u>GC<u>T</u>GGATA<u>T</u>T

Ir25a_3HA_FOR:
*GTTATAGATCACTAG*tAATATTATGGT<u>T</u>AA<u>G</u>TGAGC<u>T</u>CTCGG

Ir25a_3HA_REV:
*AATTCAGATCACTAG*tCAAAGCTAAGTTCATCGTCATAGAGAC

Genotyping and sequencing primers (PCR fragment size):

Orco_Seq_FOR (~2 kb):
GATGTTCTGCTCTTGGCTGATATTC

Ir8a_Seq_FOR (~1.9 kb):
CATCGACTTCATCATCAGGCTTTCG

Ir76b_Seq_FOR (~1.9 kb):
CAACGATATCCTCACGAAGAACAAGC

Ir25a_Seq_FOR (~1.9 kb):
CGAAAGGATACAAAGGATACTGCAT

HACK_Seq_REV (same for all):
TGTATTCCGTCGCATTTCTCTC

## Checking for off-target effects

One of the gRNAs for *QF2*[*Ir25a-HACK*] had one predicted potential off-target cut site in the genome, in the *tetraspanin 42ej* (*Tsp42Ej*) gene. We sequenced this locus in the *Ir25a-T2A-QF2* knock-in line and compared the sequence to our *wildtype* lab stock. We found no evidence of indels in the knock-in line. Primers used:

Tsp42Ej_FOR:
GAGAAGTCGTTTCCCATAACACCCT
Tsp42Ej_REV:
GAGGAGCAGTTTTCGGAGTCGCCTTC

## HACK marker screening

Adult flies were anesthetized on a $CO_2$ pad and screened in one of two ways: either with a Nightsea Stereo Microscope Fluorescence Adapter with the green SFA-GR LED light source (Nightsea LLC, Lexington, MA) and viewed with a Zeiss Stemi SV6 stereo microscope; or illuminated with an X-Cite 120Q excitation light source and viewed with a Zeiss SteREO Discovery V8 microscope equipped with a ds-Red filter.

## Whole-animal imaging

Whole adults were anesthetized on ice before imaging. Whole larvae, pupae, or freshly dissected adult heads were affixed to slides with clear nail polish before imaging. All animals were imaged on an Olympus SZX7 microscope equipped with GFP and RFP filters. Animals were illuminated with an X-Cite Series 120Q light source. Images were acquired using a QImaging QIClick Cooled digital CCD camera and Q-Capture Pro 7 software. Multiple images were taken at different focal planes and then merged in Photoshop (CS6). Gain was adjusted in Fiji. Images appear in the following figures/panels: *Figure 2B, D, F and H*; *Figure 2—figure supplement 1D*; *Figure 2—figure supplement 3A*; and *Figure 2—figure supplement 4*. For *Figure 7*, *D. melanogaster*, *D. sechellia*, *D. virilis*, and *A. coluzzii* animals were immobilized with clear nail polish and imaged on a Zeiss SteREO Discovery V8 microscope. Images were acquired with a smartphone camera attached to the microscope ocular and processed in Photoshop (CS6) to remove background.

## Immunohistochemistry

All flies were used at 3–11 days old. Apart from the cryosection protocols and portions of the antennal whole-mount protocol, all immunostaining steps were done on a nutator. All steps involving or following the addition of fluorescently conjugated secondary antibodies were done in the dark.

Brain and VNC staining was performed as in *Xie et al., 2018*. The tissue was dissected in PBS and then fixed for 20 min at room temperature (RT) in 4% paraformaldehyde in PBT (1× PBS with 0.3% Triton X-100). After fixation, the tissue was quickly rinsed three times with PBT, then put through three longer washes in PBT at RT (at least 15 min each). The tissue was blocked in PBT + 5% normal goat serum (NGS) at RT for at least 30 min, then transferred to block + primary antibody solution, and incubated at 4°C in primary antibodies for 1–2 days. The tissue was then washed three times with PBT at RT (at least 15 min per wash) and incubated in a secondary antibody solution in block at 4°C for 1 day. The tissue was washed three final times with PBT at RT for 15 min each, and then mounted in SlowFade Gold (Thermo Fisher Scientific #S36936). For experiments in which the 10XQUAS-6XGFP or 20XUAS-6XGFP reporters were used, the endogenous, unstained GFP signal was visualized, and no secondary green antibodies were used. Primary antibodies used: mouse anti-nc82 (DSHB, 1:25) and rat anti-mCD8 (Thermo Fisher #14-0081-82, 1:100 or 1:200). Secondary antibodies used: Cy3 goat anti-rat (Jackson ImmunoResearch #112-165-167, 1:200), Cy3 goat anti-mouse (Jackson ImmunoResearch #115-165-166, 1:200), Alexa 647 goat anti-rat (Jackson ImmunoResearch #112-605-167, 1:200), and Alexa 647 goat anti-mouse (Jackson ImmunoResearch #115-605-166, 1:200).

Whole-mount staining of maxillary palps was performed according to the brain staining protocol above, with the exception of a shorter fixation step (15 min). Primary antibodies used: rabbit anti-Ir25a (gift from Richard Benton, University of Lausanne, 1:100), rabbit anti-Orco (gift from Leslie Vosshall, Rockefeller University, 1:100), and rat anti-elav (DSHB, 1:100). Secondary antibodies used: Cy3 goat anti-rabbit (Jackson ImmunoResearch #111-165-144, 1:200), Alexa 647 goat anti-rabbit (Jackson ImmunoResearch #111-605-144, 1:200), and Alexa 647 goat anti-rat (Jackson ImmunoResearch #112-605-167, 1:200). For whole-mount staining of *Orco²* mutant palps, 3-day-old flies were used to check for Orco expression before neurons degenerate (*Task and Potter, 2021*).

The protocol for whole-mount staining of antennae was adapted from *Karim et al., 2014*; *Saina and Benton, 2013*; *Younger et al., 2020*. Fly heads were dissected into CCD buffer (50 units chitinase, 1000 units chymotrypsin [25 mg of 40 units/mg], 10 mL HEPES larval buffer [119 mM NaCl,

48 mM KCl, 2 mM CaCl$_2$, 2 mM MgCl$_2$, 25 mM HEPES], 100 µL DMSO) on ice, then warmed on a 37°C heat block for 10 min. Heads were incubated in CCD buffer at 37°C while rotating for 1 hr 20 min. Antennae were subsequently dissected off heads into fixative solution (4% PFA in PBT). All subsequent steps were done without rotation to prevent antennae from sticking to the walls or lids of the tubes. The antennae were fixed at RT for 40 min, then washed with PBT three times at RT, at least 15 min each time, and blocked in PBT plus 5% NGS for at least 1 hr at RT. The antennae were incubated in primary antibodies in blocking solution at 4°C for 4 days, washed three times for 15 min each at RT, and incubated in a secondary antibody solution at 4°C for 3 days. The antennae were then washed three times for 15 min each time at RT and mounted in SlowFade Gold. Primary antibody: rabbit anti-Orco (gift from Leslie Vosshall, 1:100). Secondary antibody: Cy3 goat anti-rabbit (Jackson ImmunoResearch #111-165-144, 1:200). The endogenous GFP signal was visualized.

The cryosection protocol was adapted from *Spletter et al., 2007*. Fly heads were dissected and lined up in cryomolds (Tissue-Tek #4565), covered with OCT compound (Tissue-Tek #4583), and frozen at –80°C. The samples were sectioned at ~12 µm on a Microm HM 500 cryostat (Microm International GmbH, Walldorf, Germany) and collected on SuperFrost Plus slides (Fisher #12-550-15). Slides were stored at –80°C until further processing. The slides were fixed at RT for 15 min in 4% paraformaldehyde in PBT (1× PBS with 0.3% Triton X-100), washed three times in PBT at RT (15 min each), blocked at RT for at least 30 min in PBT + 2.5% NGS + 2.5% normal donkey serum (NDS), then incubated overnight at 4°C in primary antibodies in fresh block solution in a special humidified chamber. On the next day, the slides were washed three times (15 min each) with PBT at RT and then incubated in secondary antibodies in block at 4°C overnight in the same humidified chamber covered in foil. Finally, the slides were washed three times (15 min each) with PBT at RT. DAPI (1:10,000) was included in the first wash as a nuclear counterstain. After washes, the slides were mounted in SlowFade Gold (Thermo Fisher Scientific #S36936). Primary antibody: guinea pig anti-Ir8a (gift from Richard Benton, University of Lausanne, 1:1000). Secondary antibody: Cy3 donkey anti-guinea pig (Jackson ImmunoResearch #706-165-148, 1:200).

For sacculus staining, 7- to 10-day-old flies were placed in an alignment collar. Their heads were encased in OCT (Tissue-Plus Fisher) in a silicone mold, frozen on dry ice, and snapped off. The head blocks were stored in centrifuge tubes at –80°C. A Leica cryostat was used to collect 20 µm sections of antennae. Immunohistochemical staining was carried out by fixing tissue in 4% paraformaldehyde for 10 min, followed by three 5 min washes in 1× PBS. The tissue was washed in 1× PBS containing 0.2% Triton-X (PBST) for 30 min to permeabilize the cuticle. Lastly, the tissue was washed in PBST containing 1% bovine serum albumin (BSA) to block nonspecific antibody binding. Primary antibody solution was made in PBST + 1% BSA, 200 µL was pipetted onto each slide under bridged coverslips, and slides were placed at 4°C overnight to incubate. The following day, the primary antibody was removed, and the slides were washed three times for 10 min each in PBST. Secondary antibody solution was made in PBST + 1% BSA, 200 µL was pipetted onto each slide under bridged coverslips, and left at RT in a dark box to incubate for 2 hr. After the 2 hr incubation, the slides were washed three times for 5 min each in PBST. After the last wash, the slides were allowed to dry in the dark staining box for ~30 min before being mounted in Vectashield, coverslipped, and stored at 4°C. Primary antibodies: rabbit anti-Ir25a (gift from Richard Benton, University of Lausanne, 1:100), guinea pig anti-Ir8a (gift from Richard Benton, University of Lausanne, 1:100), and rabbit anti-Ir64a (gift from Greg Suh, NYU/KAIST, 1:100). Secondary antibodies: Jackson Immuno Cy3 conjugated AffiniPure 568 goat anti-rabbit (111-165-144, 1:500) and Alexa Fluor 568 goat anti-guinea pig (A11075, 1:500).

## In situ HCR

Cryosectioning for antennal in situs was performed as described above. The HCR protocol was adapted from Molecular Instruments HCR v3.0 protocol for fresh frozen or fixed frozen tissue sections (*Choi et al., 2018*). Slides were fixed in ice-cold 4% PFA in PBT for 15 min at 4°C, dehydrated in an ethanol series (50% EtOH, 70% EtOH, 100% EtOH, 100% EtOH, 5 min each step), and air dried for 5 min at RT. The slides were then incubated in proteinase K solution in a humidified chamber for 10 min at 37°C, rinsed twice with PBS and dried, then pre-hybridized for 10 min at 37°C in a humidified chamber. The slides were then incubated in probe solution (0.4 pmol Ir76b probe) overnight in the 37°C humidified chamber. On day 2, the slides were washed with a probe wash buffer/SSCT series (75% buffer/25% SSCT, 50% buffer/50% SSCT, 25% buffer/75% SSCT, 100% SSCT, 15 min each) at 37°C, then washed

for 5 min at RT with SSCT and dried. The slides were pre-amplified for 30 min at RT in the humidified chamber while hairpins were snap cooled (6 pmol concentration). The slides were incubated in fresh amplification buffer with hairpins overnight in a dark humidified chamber at RT. On day 3, the slides were rinsed in SSCT at RT (2 × 30 min, 1 × 10 min with 1:10,000 DAPI, 1 × 5 min) and mounted in SlowFade Diamond (Thermo Fisher S36972). For the overnight steps, the slides were covered with HybriSlips (Electron Microscopy Sciences 70329-62) to prevent solution evaporation.

The whole-mount palp in situ protocol for all *Drosophila* species tested in this paper was adapted from a combination of *Prieto-Godino et al., 2020*; *Saina and Benton, 2013*; *Younger et al., 2020* and the *D. melanogaster* whole-mount embryo protocol from Molecular Instruments (*Choi et al., 2016*). All steps after dissection were performed while rotating unless otherwise noted. Fly mouthparts (palps and proboscises) were dissected into CCD buffer (same as for whole-mount IHC on antennae above), incubated for 20–65 min in CCD at 37°C (5 min on heat block, 15 min to 1 hr rotating), then pre-fixed in 4% PFA in PBT for 20 min at RT. Tissue was washed with 0.1% PBS-Tween on ice (4 × 5 min), incubated for 1 hr at RT in 80% methanol/20% DMSO, and washed for 10 min in PBS-Tween at RT. The tissue was incubated in Proteinase K solution (1:1000) in PBS-Tween at RT for 30 min, then washed in PBS-Tween at RT (2 × 10 min) and post-fixed in 4% PFA in PBS-Tween at RT for 20 min. After post-fixation, the tissue was washed in PBS-Tween at RT (3 × 15 min), then pre-hybridized in a pre-heated probe hybridization buffer at 37°C for 30 min. The tissue was incubated in probe solution (2–30 pmol in hybridization buffer) at 37°C for 2–3 nights. After probe incubation, the tissue was washed in a pre-heated probe wash buffer at 37°C (5 × 10 min), then washed in SSCT (1× SSC plus 1% Tween) at RT (2 × 5 min). The tissue was pre-amplified with RT-equilibrated amplification buffer at RT for 10 min, then incubated in hairpin mixture (6–30 pmol snap-cooled hairpins in amplification buffer) in the dark at RT for 1–2 nights. After hairpin incubation, the tissue was washed at RT with SSCT (2 × 5 min, 2 × 30 min, 1 × 5 min), then mounted in SlowFade Diamond (Thermo Fisher S36972). Sequences for all in situ probes can be found in Appendix 2. In addition to *D. sechellia* (Cousin Island, Seychelles genome line, SKU: 14021-0248.25; *Figure 7*), we also tested the same in situ probes for six other species of *Drosophila* (apart from *D. virilis*, all species were ordered from the National *Drosophila* Species Stock Center): *Drosophila simulans* (genome line w[501], SKU: 14021-0251.195), *Drosophila erecta* (wild-type genome line, SKU: 14021-0224.01), *Drosophila ananassae* (wildtype line, Queensland, Australia, SKU: 14024-0371.11), *Drosophila pseudoobscura* (genome line, Anderson, Mesa Verde, CO, SKU: 14011-0121.94), *Drosophila mojavensis* (wildtype line, Catalina Island, CA [2002], SKU: 15081-1352.22), and *Drosophila virilis* (wildtype, Carolina Biological Supply, item# 172890). While we could detect a clean signal for the *D. mel. Orco* probe in these other fly species, the high background and poor signal-to-noise ratio for *D. mel. Ir25a* prevented co-localization analyses for all species but *D. sechellia*.

*Anopheles* in situs were performed essentially as described above, with the following modifications: olfactory appendages were dissected from 5- to 6-day-old female mosquitoes and incubated in CCD buffer for either 20 min (antennae) or 1.5 hr (maxillary palps) at 37°C while rotating. The tissue was then pre-fixed for 24 hr at 4°C. After the PBS-Tween and methanol/DMSO washes, the tissue was incubated overnight at –20°C in absolute methanol. The next day, the tissue was rehydrated in a graded methanol/PBS-Tween series. Subsequent steps follow the *Drosophila* whole-mount palp protocol. Probe concentration was 8 pmol in hybridization buffer (two night incubation), and hairpin concentration was 18 pmol in amplification buffer (one night incubation). The tissue was rinsed three times in SlowFade Diamond before being mounted.

For two-color in situs, we used the Molecular Instruments B2 amplifier conjugated to Alexa 647 for *Orco* and the B4 amplifier conjugated to Alexa 488 for *Ir25a*. For single-color in situs (*Orco*, *Ir76b* or tuning *IrXs*), we used the B2 amplifier conjugated to Alexa 647.

## Confocal imaging and analysis

Brains, VNCs, antennae, maxillary palps, and antennal cryosections were imaged on a Zeiss LSM 700 confocal microscope equipped with Fluar 10×/0.50 air M27, LCI Plan-Neofluar 25×/0.8 water Korr DIC M27, Plan-Apochromat 40×/1.3 Oil DIC M27, and C-Apochromat 63×/1.2 water Korr M27 objectives. Images were acquired at either 512 × 512 or 1024 × 1024-pixel resolution with 0.43, 0.58, 2.37, or 6.54 μm z-steps. For illustration purposes, confocal images were processed in Fiji/ImageJ to collapse Z-stacks into a single image using maximum intensity projection. Where noted, single slices or partial z projections were used as opposed to full stacks. For co-labeling experiments, Fiji was used to convert

red Look-Up Tables (LUTs) to orange for a colorblind-friendly palette. Similarly, in *Figure 2—figure supplement 1E*, Fiji was used to convert magenta LUT to blue for clarity. For *Figure 4B*, Fiji was used to convert the two-channel maximum intensity projection to a gray LUT, and the cell-counting plug-in was used in separate channels to identify single- and double-labeled cells. Fiji was also used to adjust the gain in separate channels in all figures/images; no other image processing was performed on the confocal data. For *Figure 8A*, glomeruli were assigned to the categories strong, intermediate, and weak by visual inspection of the strength of their innervation compared to the previously reported glomeruli for each respective knock-in line. Strong glomeruli generally have similar brightness/intensity of GFP signal as most of the originally reported glomeruli for the given knock-in line.

For sacculus staining (*Figure 5C–E*), slides were imaged on a Nikon A1R confocal microscope in the UConn Advanced Light Microscopy Facility with a 40× oil immersion objective at 1024 × 1024-pixel resolution. Stacks of images (0.5 μm z-step size) were gathered and analyzed with ImageJ/Fiji software. Image processing was performed as described above.

Magnification used:

> 10×: *Figure 1D* (brain), *Figure 2—figure supplement 1E*
> 25×: *Figure 2—figure supplement 1F and G*, *Figure 3—figure supplement 1A and B*
> 40×: *Figures 5C–E and 7D and E*
> 63×: *Figure 2C, E, G, and I*, *Figure 3A–D*, *Figure 4B and C*, *Figure 5A and B*, *Figure 2—figure supplement 3E–H*, *Figure 3—figure supplement 2A–E*, *Figure 5—figure supplement 1*, *Figure 6—figure supplement 2*, *Figure 7C*

Note regarding *Ir8a* knock-in expression: in the ALs, we found that the sparse Ir8a+ expression in olfactory neurons targeting VM6m and VM6v could potentially be sexually dimorphic. Male brains generally had stronger and more frequent Ir8a+ innervation in these two glomeruli compared to female brains, as shown in *Figure 3—source data 2* (see *Figure 3—source data 1* for a summary of AL analyses). However, we did not find corresponding evidence for sexual dimorphism in Ir8a+ expression in the periphery. The reason for this discrepancy is currently unclear, and future work will be needed to determine whether there are functional male/female differences in Ir8a+ neurons.

## Basiconic single sensillum recordings

Flies were immobilized and visualized as previously described (*Lin and Potter, 2015*). Basiconic sensilla were identified either using fluorescent-guided SSR (for ab3 sensilla) or using the strength of the A and B neuron responses to the reference odorants 1% ethyl acetate (EA) (Sigma #270989) and 1% pentyl acetate (PA) (Sigma #109584) (for ab2 and pb1-3 sensilla) (*de Bruyne et al., 1999*; *Lin and Potter, 2015*). For example, pb1A has strong responses to both odorants, while pb3A does not respond to EA. Similarly, ab2 sensilla were distinguished from ab3 based on the A neuron responses: ab2A responds strongly to EA and weakly to PA, while ab3A neurons have the reverse response (weak EA and strong PA response). The glass recording electrode was filled with Beadle–Ephrussi Ringer's solution (7.5 g of NaCl + 0.35 g of KCl + 0.279 g of CaCl$_2$-2H$_2$O in 1 L H$_2$O). Extracellular activity was recorded by inserting the glass electrode into the shaft or base of the sensillum of 3- to 10-day-old flies (unless otherwise specified in the young *Orco$^2$* mutant experiments). A tungsten reference electrode was inserted into the fly eye. Signals were amplified 100× (USB-IDAC System; Syntech, Hilversum, the Netherlands), input into a computer via a 16-bit analog-digital converter, and analyzed offline with AUTOSPIKE software (USB-IDAC System; Syntech). The low cutoff filter setting was 50 Hz, and the high cutoff was 5 kHz. Stimuli consisted of 1000 ms air pulses passed over odorant sources. The Δ spikes/s was calculated by counting the spikes in a 1000 ms window from ~500 ms after odorant stimuli were triggered, subtracting the spikes in a 1000 ms window prior to each stimulation. For ab3 recordings from *wildtype*, *Orco$^2$* mutant, and *Ir25a$^2$* mutant flies, spikes were counted in a 500 ms window from the start of the response and multiplied by 2. Then, the spikes in the 1000 ms window prior to stimulation were subtracted from this to calculate the Δ spikes/s. Stimuli used: mineral oil (Sigma CAS# 8042-47-5), EA (Sigma CAS# 141-78-6), PA (Sigma CAS# 628-63-7), benzaldehyde (Sigma CAS# 100-52-7), ethyl butyrate (Sigma CAS# 105-54-4), hexanol (Sigma CAS# 111-27-3), e2-hexenal (Sigma CAS# 6728-26-3), geranyl acetate (Sigma CAS# 105-87-3), 2-heptanone (Sigma CAS# 110-43-0), 1-octen-3-ol (Sigma CAS#3391-86-4), 2,3-butanedione (Sigma CAS#431-03-8), phenylacetaldehyde (Sigma CAS# 122-78-1), phenethylamine (Sigma CAS# 64-04-0), propionic

acid (Sigma CAS# 79-09-4), 1,4-diaminobutane (Sigma CAS# 110-60-1), pyrrolidine (Sigma CAS# 123-75-1), p-cresol (Sigma CAS# 106-44-5), and methyl salicylate (Sigma CAS# 119-36-8). Odorants were dissolved in mineral oil at a concentration of 1%, and 20 µL of solution was pipetted onto filter paper in a glass Pasteur pipette. Stimuli were delivered by placing the tip of the Pasteur pipette through a hole in a plastic pipette (Denville Scientific Inc, 10 mL pipette) that carried a purified continuous air stream (8.3 mL/s) directed at the antenna or maxillary palp. A solenoid valve (Syntech) diverted delivery of a 1000 ms pulse of charcoal-filtered air (5 mL/s) to the Pasteur pipette containing the odorant dissolved on filter paper. Fresh odorant pipettes were used for no more than five odorant presentations. *Ir25a²* and *Orco²* mutant fly lines were outcrossed into the *w¹¹¹⁸* wildtype genetic background for at least five generations. Full genotypes for ab3 fgSSR were *Pin/CyO;Or22a-Gal4,15XUAS-IVS-mcd8GFP/TM6B* (*wildtype*), *Ir25a²;Or22a-Gal4,15XUAS-IVS-mcd8GFP/TM6B* (*Ir25a² mutant*), and *Or22a-Gal4/10XUAS-IVS-mcd8GFP (attp40);Orco²* (*Orco² mutant*). These stocks were made from the following Bloomington Stocks (outcrossed to the Potter lab *w¹¹¹⁸* genetic background): BDSC# 9951, 9952, 23130, 32186, 32193, and 41737.

## Coeloconic single sensillum recordings

Coeloconic SSR was performed similarly as for basiconic sensilla. Three- to five-day-old female flies were wedged in the tip of a 200 µL pipette, with the antennae and half the head exposed. A tapered glass electrode was used to stabilize the antenna against a coverslip. A BX51WI microscope (Olympus) was used to visualize the prep, which was kept under a 2000 mL/min humidified and purified air stream. A borosilicate glass electrode was filled with sensillum recording solution (***Kaissling and Thorson, 1980***) and inserted into the eye as a reference electrode. An aluminosilicate glass electrode was filled with the same recording solution and inserted into individual sensilla. Different classes of coeloconic sensilla were identified by their known location on the antenna and confirmed with their responses to a small panel of diagnostic odorants: in *wildtype* flies, ac2 sensilla were identified by their strong responses to 1,4-diaminobutane and 2,3-butanedione. The absence of a strong response to ammonia was used to rule out ac1 sensilla, the absence of a hexanol response was used to rule out ac3 sensilla, and the absence of a phenethylamine response was used to rule out ac4 sensilla. In *Ir25a* mutant flies in which amine responses were largely abolished, ac2 and ac4 sensilla were distinguished based on anatomical location, as well as the strong response of ac2 to 2,3-butanedione and the moderate response to propanal (both absent in ac4). ac1 and ac3 sensilla were excluded similarly in the mutant and wildtype flies. No more than four sensilla per fly were recorded. Each sensillum was tested with multiple odorants, with a rest time of at least 10 s between applications. The odorants used were acetic acid (Fisher, 1%, CAS# 64-19-7), ammonium hydroxide (Fisher, 0.1%, CAS# 7664-41-7), cadaverine (Sigma-Aldrich, 1%, CAS# 462-94-2), hexanol (ACROS Organics, 0.001%, CAS# 111-27-3), 2,3-butanedione (ACROS Organics, 1%, CAS# 431-03-8), phenethylamine (ACROS Organics, 1%, CAS# 64-04-0), propanal (ACROS Organics, 1%, CAS# 123-38-6), and 1,4-diaminobutane (ACROS Organics, 1%, CAS# 110-60-1). Odorants were diluted in water or paraffin oil. Odorant cartridges were made by placing a 13 mm antibiotic assay disc (Whatman) into a Pasteur pipette, pipetting 50 µL odorant onto the disc, and closing the end with a 1 mL plastic pipette tip. Each odorant cartridge was used a maximum of four times. The tip of the cartridge was inserted into a hole in the main airflow tube, and odorants were applied at 500 mL/min for 500 ms. Delivery was controlled via LabChart Pro v8 software (ADInstruments), which directed the opening and closing of a Lee valve (02-21-08i) linked to a ValveBank 4 controller (AutoMate Scientific). Extracellular action potentials were collected with an EXT-02F amplifier (NPI) with a custom 10X head stage. Data were acquired and AC filtered (300–1700 Hz) at 10 kHz with a PowerLab 4/35 digitizer and LabChart Pro v8 software. Spikes were summed in coeloconic recordings due to their similar sizes, and they were counted over a 500 ms window, starting at 100 ms after stimulus onset.

## Optogenetics

*Ir25a-T2A-QF2* was crossed to *QUAS-CsChrimson* #11C and double balanced to establish a stable stock (*Ir25a-T2A-QF2/CyO; QUAS-CsChrimson* #11C/*TM6B*). Newly eclosed flies (age <1 day old) were transferred to fly vials containing 0.4 mM all trans-retinal in fly food (Sigma-Aldrich #R2500, dissolved in pure DMSO with stock concentration of 0.4 M). Vials with flies were kept in the dark for at least 4 days before experiments. 627 nm LED light source (1-up LED Lighting Kit, Part#

ALK-1UP-EH-KIT) powered by an Arduino Uno (https://docs.arduino.cc/hardware/uno-rev3/) was used to activate CsChrimson. By setting the voltage to 2 V and the distance of the light source to 20 cm between the LED and the fly antenna, the light intensity was equivalent to 1.13 W/m². The antenna was stimulated for 500 ms followed by 5 s of recovery period for the total recording length of 20 s (three stimulations). The identity of ab2 and ab3 sensilla were first verified with 1% EA (Sigma #270989) and 1% PA (Sigma #109584) before optogenetic experiments. Identification of ab9 sensilla was assisted by fluorescence-guided SSR (fgSSR) (*Lin and Potter, 2015*) using *Or67b-Gal4* (BDSC #9995) recombined with *15XUAS-IVS-mCD8::GFP* (BDSC #32193). The Δ spikes/s was calculated as for other basiconic SSR. For all optogenetic experiments, the control flies were of the same genotype as experimental flies but had not been fed all-trans retinal.

## Single-nucleus RNA-sequencing analyses

Dataset analyzed in this paper was published in *McLaughlin et al., 2021*. The expression levels for the *Ir co-receptors* across all OSNs were lower than for *Orco*, even for their corresponding 'canonical' glomeruli. To account for these differences and facilitate comparisons, we performed within-gene normalization in *Figure 4—source data 1* and used the normalized values to generate the AL maps in *Figure 4A*. The normalization was performed as follows: first, we determined the fraction of cells within each cluster expressing the given co-receptor (read counts per million, CPM threshold ≥3). The cluster with the highest fraction value was taken as the maximum. Then, the fraction for each cluster was divided by this maximum value. The normalized value shows the relative strength of expression within each cluster for the given co-receptor gene.

## EM neuron reconstruction

VM6 OSNs (*Figure 5F and G*) were reconstructed in the FAFB EM volume (*Zheng et al., 2018*) using FlyWire (https://flywire.ai/; *Dorkenwald et al., 2020*). Initial candidates were selected based on either being upstream of the VM6 (previously called VC5 in *Bates et al., 2020*) projection neurons or based on co-fasciculation with already identified VM6 OSNs. Analyses were performed in Python using the open-source packages *navis* (https://github.com/schlegelp/navis) and *fafbseq* (https://github.com/flyconnectome/fafbseg-py). OSNs were clustered using FAFB synapse predictions (*Buhmann et al., 2021*) for a synapse-based NBLAST ('syNBLAST,' implemented in *navis*). The reconstructions can be viewed at https://flywire.ai/#links/Task2021a/all.

## Phylogenetic analysis

*D. melanogaster*, *D. sechellia*, and *D. virilis Orco* sequences were compared using the FlyBase (https://flybase.org/) BLAST tool (reference sequences examined: XM_002038370.1, XM_032721743.1, XM_002056720.3). The *A. coluzzii Orco* sequence was downloaded from VectorBase (https://vectorbase.org/) (sequence reference UniProtKB/TrEMBL;Acc:A0A182LER8). The pea aphid *Orco* sequence was acquired from the European Nucleotide Archive (https://www.ebi.ac.uk/ena/browser/view/AQS60741) (sequence reference ENA|AQS60741|AQS60741.1). Sequences were aligned using MUSCLE in MEGA11 software. This alignment was used to generate the phylogenetic tree shown in *Figure 7A*. The tree with the highest log likelihood (–6037.19) was used. Initial trees for the heuristic search were obtained automatically by applying Neighbor-Join and BioNJ algorithms to a matrix of pairwise distances estimated using the Tamura–Nei model, and then selecting the topology with superior log likelihood value. The tree in *Figure 7A* is drawn to sale, with branch lengths measured in the number of substitutions per site (scale bar = 0.1). Codon positions included were 1st + 2nd + 3rd + Noncoding. There were a total of 1488 positions in the final dataset. Evolutionary analyses were conducted in MEGA11 (*Tamura et al., 2021*). The tree is rooted to the pea aphid (*Acyrthosiphon pisum*) outgroup, thought to represent one of the most evolutionarily ancient examples of functional Orco/Or complexes (*Missbach et al., 2014*; *Smadja et al., 2009*; *Soffan et al., 2018*).

## Quantification and statistical analysis
### Cell counting
To quantify knock-in co-expression with the corresponding antibodies (*Figure 2*), the 3D reconstruction software Amira (FEI, OR) was used to manually mark individual cell bodies throughout the z-stack in each channel (antibody in far red channel, knock-in in green channel), and the cell markers between

channels were compared. We also used Amira for the *D. sechellia* palp cell counts (*Figure 7C*). Cells were first marked in the far red (Orco) channel, then subsequently in the green (Ir25a) channel.

For sacculus cell counts (*Figure 5C–E*), cells were counted within ImageJ/Fiji using the cell counter tool. Counts were done manually by going through each stack within an image and using different colored markers for each cell type.

For *Anopheles* cell counts (*Figure 7D and E*), cells were counted in the Zeiss software (Zen Black) with the help of a manual cell counter.

All cell count data were gathered in Excel and analyzed for percent colocalization.

## Statistics

Statistical analyses on SSR data were done in GraphPad Prism (version 8), except for optogenetic experiments, which were analyzed in Microsoft Excel. Box plots were made using GraphPad Prism; bar graphs were made in Excel. For all analyses, significance level $\alpha$ = 0.05. The following analyses were performed on all SSR data (excluding optogenetics): within genotype, Kruskal–Wallis test with uncorrected Dunn's to determine which odorant responses were significantly different from mineral oil or paraffin oil control; between genotype, Mann–Whitney $U$ to compare responses of two genotypes to the same odorant (e.g., *wildtype* vs. $Ir25a^2$ mutant, or *wildtype* vs. *Orco-T2A-QF2* knock-in). Summary tables in *Figure 6* are filled in based on the following criteria: no response means neither *wildtype* nor $Ir25a^2$ mutant odor-evoked activity for given odorant was significantly different from its respective mineral oil control, nor was the difference between the genotypes statistically significant; no difference means that either *wildtype* or mutant or both had a significantly different odor-evoked response to the stimulus compared to mineral oil control, but the difference between the two genotypes was not statistically significant; higher response (in either *wildtype* or mutant) means that there was a statistically significant difference between genotypes for the given odorant. This could mean that (a) one genotype did not have a response, while the other did; (b) both genotypes had a response, and one was higher; (c) responses are different from each other, but not from their respective mineral oil controls; or (d) neural activity was inhibited by the odorant in one genotype compared to mineral oil control, and either not inhibited in the other genotype or inhibited to a lesser degree. Nonparametric tests were chosen due to small sample sizes and/or data that were not normally distributed.

In *Figure 6I*, stimulus responses that were statistically significantly different from mineral oil control were those whose Δ spike values were zero due to the fact that the mineral oil control Δ spike value was nonzero (median = 1.2, range = 0–2). Because of this, we did not deem these differences as biologically relevant. Nevertheless, p-values are reported in the figure legend of *Figure 6*, and detailed information can be found in *Figure 6—source data 1*.

## Acknowledgements

We thank E Marr and K Robinson for splinkerette genetic mapping, Y-T Chang for cloning of *pHACK*[Ir8a] components, O Riabinina for initial *QUAS-CsChrimson* characterization, S Maguire for preliminary SSR experiments, P Mohapatra for RNAseq insights, D Baktash for help with figures, J Konopka for advice on statistical and phylogenetic analyses, S Shankar for discussion of AL mapping, and R Mann for providing lab resources. We would like to thank EC Marin and M Costa for discussions regarding posterior AL glomeruli. We are also grateful to the Seung and Murthy labs for access to the flywire. ai reconstruction community. Many thanks to the following labs for sharing antibody and fly stock reagents: Leslie B Vosshall (Rockefeller), Richard Benton (UNIL), Paul Garrity (Brandeis), Marco Gallio (Northwestern), Greg Suh (NYU/KAIST), Andrew Gordus (JHU), and Thomas R Clandinin (Stanford). We are grateful for in situ advice from Richard Benton, Steeve Cruchet, and Margo Herre. We thank the Center for Sensory Biology Imaging Facility (NIH P30DC005211) for use of the LSM700 confocal microscope, and the Johns Hopkins School of Public Health Malaria Research Institute for use of the Olympus SZX7 microscope equipped with QImaging QIClick Cooled digital CCD camera. We thank S Maguire and J Konopka for comments on the manuscript and members of the Potter and Menuz labs for discussion. We thank the Vosshall lab for sharing their *Aedes* findings before publication. This work was supported by a Shelanski award to C-CL; a Wellcome Trust Collaborative Award (203261/Z/16/Z) and an NIH BRAIN Initiative grant (1RF1MH120679-01) to GSXEJ; grants from the National Institutes of Health to HL (R00 AG062746); grants from the National Institutes of Health to KM (NIGMS R35GM133209; NIDCD 1R21DC017868); grants from the Department of Defense to CJP

(W81XWH-17-PRMRP) and from the National Institutes of Health to CJP (NIAID R01Al137078; NIDCD R01DC013070). Portions of this work appear in Chapter 3 of DT's doctoral dissertation.

## Additional information

### Funding

| Funder | Grant reference number | Author |
|---|---|---|
| National Institute of Allergy and Infectious Diseases | R01Al137078 | Christopher J Potter |
| National Institute on Deafness and Other Communication Disorders | R01DC013070 | Christopher J Potter |
| U.S. Department of Defense | W81XWH-17-PRMRP | Christopher J Potter |
| Shelanski Research Innovation Award in Pathology | 2018 | Chun-Chieh Lin |
| Wellcome Trust | 203261/Z/16/Z | Gregory SXE Jefferis |
| National Institutes of Health | BRAIN Initiative | Gregory SXE Jefferis |
| National Institutes of Health | R00 AG062746 | Hongjie Li |
| National Institute of General Medical Sciences | R35GM133209 | Karen Menuz |
| National Institute on Deafness and Other Communication Disorders | R21DC017868 | Karen Menuz |

The funders had no role in study design, data collection and interpretation, or the decision to submit the work for publication.

### Author contributions

Darya Task, Conceptualization, Formal analysis, Investigation, Methodology, Visualization, Writing – original draft, Writing – review and editing; Chun-Chieh Lin, Maria Brbic, Philipp Schlegel, Hongjie Li, Conceptualization, Formal analysis, Investigation, Methodology, Visualization, Writing – review and editing; Alina Vulpe, Conceptualization, Formal analysis, Investigation, Methodology, Writing – review and editing; Ali Afify, Sydney Ballou, Joshua Raji, Formal analysis, Investigation, Methodology, Writing – review and editing; Gregory SXE Jefferis, Supervision, Writing – review and editing; Karen Menuz, Conceptualization, Methodology, Supervision, Writing – review and editing; Christopher J Potter, Conceptualization, Funding acquisition, Methodology, Supervision, Visualization, Writing – original draft, Writing – review and editing

### Author ORCIDs

Darya Task http://orcid.org/0000-0003-0166-3626
Philipp Schlegel http://orcid.org/0000-0002-5633-1314
Gregory SXE Jefferis https://orcid.org/0000-0002-0587-9355
Christopher J Potter http://orcid.org/0000-0002-5223-8112

### Decision letter and Author response

Decision letter https://doi.org/10.7554/eLife.72599.sa1
Author response https://doi.org/10.7554/eLife.72599.sa2

## Additional files

### Supplementary files
• Transparent reporting form

### Data availability
All data generated or analyzed during this study are included in the manuscript and supporting files; Source Data files have been provided for Table 1, Figure 2, Figure 3, Figure 4, Figure 6, and Figure 7. Raw data used to generate each figure is available at Johns Hopkins Research Data Repository (https://doi.org/10.7281/T1/9VJGPI).

The following dataset was generated:

| Author(s) | Year | Dataset title | Dataset URL | Database and Identifier |
| --- | --- | --- | --- | --- |
| Task D, Lin CC, Vulpe A, Afify A, Ballou S, Brbić M, Schlegel P, Raji J, Jefferis GSXE, Li H, Menuz K, Potter CJ | 2023 | Data associated with the publication: Chemoreceptor co-expression in *Drosophila melanogaster* olfactory neurons. | https://doi.org/10.7281/T1/9VJGPI | Johns Hopkins Research Data Repository, 10.7281/T1/9VJGPI |

The following previously published datasets were used:

| Author(s) | Year | Dataset title | Dataset URL | Database and Identifier |
| --- | --- | --- | --- | --- |
| McLaughlin CN, Brbić M, Xie Q, Li T, Horns F, Kolluru SS, Kebschull JM, Vacek D, Xie A, Li J, Jones RC, Leskovec J, Quake SR, Luo L Li H | 2021 | Single-cell transcriptomes of developing and adult olfactory receptor neurons in *Drosophila* | https://www.ncbi.nlm.nih.gov/geo/query/acc.cgi?acc=GSE162121 | NCBI Gene Expression Omnibus, GSE162121 |
| Schlegal P, Jefferis GSXE | 2021 | EM Reconstructions of VM6 OSNs | https://flywire.ai/#links/Task2021a/all | FAFB EM volume, FAFB |

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

# Appendix 1

## Appendix 1—key resources table

| Reagent type (species) or resource | Designation | Source or reference | Identifiers | Additional information |
|---|---|---|---|---|
| Antibody | Anti-nc82 (mouse monoclonal) | DSHB | Cat# nc82; RRID:AB_2314866 | IHC (1:25) |
| Antibody | Anti-cd8 (rat monoclonal) | Thermo Fisher Scientific | Cat# 14-0081-82; RRID:AB_467087 | IHC (1:100 or 1:200) |
| Antibody | Anti-elav (rat monoclonal) | DSHB | Cat# Rat-Elav-7E8A10; RRID:AB_528218 | IHC (1:100) |
| Antibody | Anti-Orco (rabbit polyclonal) | Gift from Leslie Vosshall *Larsson et al., 2004* | | IHC (1:100) |
| Antibody | Anti-Ir25a (rabbit polyclonal) | Gift from *Benton et al., 2009* | RRID:AB_2567027 | IHC (1:100) |
| Antibody | Anti-Ir8a (guinea pig polyclonal) | Gift from Richard Benton *Abuin et al., 2011* | RRID:AB_2566833 | IHC (1:100 for whole-mount, 1:1000 for cryosections) |
| Antibody | Anti-Ir64a (rabbit polyclonal) | Gift from Greg Suh *Ai et al., 2010* | RRID:AB_2566854 | IHC (1:100) |
| Antibody | Anti-guinea pig Alexa 568 (goat polyclonal) | Thermo Fisher Scientific | Cat# A11075; RRID:AB_141954 | IHC (1:500) |
| Antibody | Anti-rabbit Cy3 conjugated AffiniPure 568 (goat polyclonal) | Jackson ImmunoResearch | Cat# 111-165-144; RRID:AB_2338006 | IHC (1:500) |
| Antibody | Anti-mouse Cy3 (goat polyclonal) | Jackson ImmunoResearch | Cat# 115-165-166; RRID:AB_2338692 | IHC (1:200) |
| Antibody | Anti-mouse Alexa 647 (goat polyclonal) | Jackson ImmunoResearch | Cat# 115-605-166; RRID:AB_2338914 | IHC (1:200) |
| Antibody | Anti-guinea pig Cy3 (donkey polyclonal) | Jackson ImmunoResearch | Cat# 706-165-148; RRID:AB_2340460 | IHC (1:200) |
| Antibody | Anti-rat Cy3 (goat polyclonal) | Jackson ImmunoResearch | Cat# 112-165-167; RRID:AB_2338251 | IHC (1:200) |
| Antibody | Anti-rat Alexa 647 (goat polyclonal) | Jackson ImmunoResearch | Cat# 112-605-167; RRID:AB_2338404 | IHC (1:200) |
| Antibody | Anti-rabbit Cy3 (goat polyclonal) | Jackson ImmunoResearch | Cat# 111-165-144; RRID:AB_2338006 | IHC (1:200) |
| Antibody | Anti-rabbit Alexa 647 (goat polyclonal) | Jackson ImmunoResearch | Cat# 111-605-144; RRID:AB_2338078 | IHC (1:200) |
| Recombinant DNA reagent | pHACK-QF2 (plasmid) | Addgene *Lin and Potter, 2016a* | Plasmid# 80274; RRID:Addgene_80274 | QF2 HACK backbone Contains QF2-hsp70, but no gRNAs |
| Recombinant DNA reagent | p10XQUAS-CsChrimson-SV40 (plasmid) | This paper | Plasmid# 163629; RRID:Addgene_163629 | For red-shifted optogenetic activation of neurons under control of the Q-system; see 'Fly stocks' |
| Recombinant DNA reagent | pHACK-QF2$^{Orco}$ (plasmid) | This paper | | HACK construct targeting the *Orco* gene; see 'Plasmid construction' |
| Recombinant DNA reagent | pHACK-QF2$^{Ir8a}$ (plasmid) | This paper | | HACK construct targeting the *Ir8a* gene; see 'Plasmid construction' |
| Recombinant DNA reagent | pHACK-QF2$^{Ir76b}$ (plasmid) | This paper | | HACK construct targeting the *Ir76b* gene; see 'Plasmid construction' |
| Recombinant DNA reagent | pHACK-QF2$^{Ir25a}$ (plasmid) | This paper | | HACK construct targeting the *Ir25a* gene; see 'Plasmid construction' |
| Genetic reagent (*Drosophila melanogaster*) | Orco-T2A-QF2 knock-in | This paper | BDSC 92400, 92401, 92402 | See 'Generation of HACK knock-in lines' |
| Genetic reagent (*D. melanogaster*) | Ir8a-T2A-QF2 knock-in | This paper | BDSC 92398, 92399 | See 'Generation of HACK knock-in lines' |
| Genetic reagent (*D. melanogaster*) | Ir76b-T2A-QF2 knock-in | This paper | BDSC 92396, 92397 | See 'Generation of HACK knock-in lines' |

*Appendix 1 Continued on next page*

*Appendix 1 Continued*

| Reagent type (species) or resource | Designation | Source or reference | Identifiers | Additional information |
|---|---|---|---|---|
| Genetic reagent (*D. melanogaster*) | *Ir25a-T2A-QF2 knock-in* | This paper | BDSC 92392, 92393, 92394, 92395 | See 'Generation of HACK knock-in lines' |
| Genetic reagent (*D. melanogaster*) | *10XQUAS-CsChrimson (y[1] w[*]; Pin[1]/ CyO; P(w[+mC] = 10XQUAS-CsChrimson. mVenus)11c)* | This paper | BDSC 91996, FlyBase FBst0091996 | See 'Fly stocks' |
| Genetic reagent (*D. melanogaster*) | *Ir21a-T2A-Gal4 knock-in* | Gift from Paul Garrity **Marin et al., 2020** | | |
| Genetic reagent (*D. melanogaster*) | *Ir68a-T2A-Gal4 knock-in* | Gift from Paul Garrity **Marin et al., 2020** | | |
| Genetic reagent (*D. melanogaster*) | *Or7a-Gal4 knock-in (y[1] w[*] TI(GAL4)Or7a[KI-GAL4.w-])* | Potter lab | BDSC 91991, FlyBase FBti0214362 | |
| Genetic reagent (*D. melanogaster*) | *Ir64a-Gal4* | Gift from Greg Suh lab **Ai et al., 2010** | | |
| Genetic reagent (*D. melanogaster*) | *Rh50-Gal4* | Menuz lab **Vulpe et al., 2021** | | |
| Genetic reagent (*D. melanogaster*) | *Amt-Gal4* | Menuz lab **Menuz et al., 2014** | | |
| Genetic reagent (*D. melanogaster*) | *Repo-Gal80* | **Awasaki et al., 2011** | FlyBase FBtp0067904 | |
| Genetic reagent (*D. melanogaster*) | *wCS (Cantonized w$^{1118}$)* | **Koh et al., 2014** | FlyBase FBrf0226011 | |
| Genetic reagent (*D. melanogaster*) | *Or35a-Gal4* | **Yao et al., 2005** | | |
| Genetic reagent (*D. melanogaster*) | *Crey +1B (y[1] w[67c23] P(y[+mDint2] = Crey)1b; sna[Sco]/CyO)* | Bloomington Drosophila Stock Center | BDSC 766, FlyBase FBti0012692 | |
| Genetic reagent (*D. melanogaster*) | *Gr21a-GAL4 (w[*]; P(w[ + mC] = Gr21a-GAL4.9.323)2/CyO; Dr[1]/TM3, Sb[+])* | Bloomington Drosophila Stock Center | BDSC 57600, FlyBase FBti0162643 | |
| Genetic reagent (*D. melanogaster*) | *Gr28b.d-GAL4 (w[*]; P(w[+mC] = Gr28b.d-GAL4)B27; Dr[1]/TM3, Sb[1])* | Bloomington Drosophila Stock Center | BDSC 57620, FlyBase FBst0057620 | |
| Genetic reagent (*D. melanogaster*) | *Ir25a-GAL4 (w[*]; P(w[ + mC] = Ir25a-GAL4.A)236.1; TM2/ TM6B, Tb[1])* | Bloomington Drosophila Stock Center | BDSC 41728, FlyBase FBti0148895 | |
| Genetic reagent (*D. melanogaster*) | *Ir40a-GAL4 (w[*]; P(w[ + mC] = Ir40a-GAL4.3011)214.1; TM2/ TM6B, Tb[1])* | Bloomington Drosophila Stock Center | BDSC 41727, FlyBase FBst0041727 | |
| Genetic reagent (*D. melanogaster*) | *Ir41a-GAL4 (w[*]; P(y[+ t7.7] w[+mC] = Ir41a-GAL4.2474)attP40; TM2/ TM6B, Tb[1])* | Bloomington Drosophila Stock Center | BDSC 41749, FlyBase FBst0041749 | |
| Genetic reagent (*D. melanogaster*) | *Ir64a-GAL4 (w[*]; P(w[+mC] = Ir64a-GAL4.A)183.8; TM2/ TM6B, Tb[1])* | Bloomington Drosophila Stock Center | BDSC 41732, FlyBase FBti0148898 | |
| Genetic reagent (*D. melanogaster*) | *Ir76a-GAL4 (w[*]; P(w[+ mC] = Ir76a-GAL4. PB)292.3B; TM2/TM6B, Tb[1])* | Bloomington Drosophila Stock Center | BDSC 41735, FlyBase FBst0041735 | |
| Genetic reagent (*D. melanogaster*) | *Ir76b-GAL4 (w[*]; P(w[ + mC] = Ir76b-GAL4.916)226.8; TM2/ TM6B, Tb[+])* | Bloomington Drosophila Stock Center | BDSC 41730, FlyBase FBti0153291 | |

*Appendix 1 Continued on next page*

*Appendix 1 Continued*

| Reagent type (species) or resource | Designation | Source or reference | Identifiers | Additional information |
|---|---|---|---|---|
| Genetic reagent (*D. melanogaster*) | Ir8a-GAL4 (w[*]; P(w[ + mC] = Ir8a-GAL4.A)204.8; TM2/ TM6B, Tb[1]) | Bloomington Drosophila Stock Center | BDSC 41731, FlyBase FBti0148897 | |
| Genetic reagent (*D. melanogaster*) | Or10a-Gal4 (w[*]; l(2)*[*]/ CyO; P(w[ + mC] = Or10a-GAL4.C)134t1.3) | Bloomington Drosophila Stock Center | BDSC 23885, FlyBase FBti0102042 | |
| Genetic reagent (*D. melanogaster*) | Or13a-Gal4 (w[*]; P(w[ + mC] = Or13a-GAL4.C)229t56.2/TM3, Sb[1]) | Bloomington Drosophila Stock Center | BDSC 23886, FlyBase FBti0102056 | |
| Genetic reagent (*D. melanogaster*) | Or22a-Gal4 (w[*]; P(w[ + mC] = Or22a-GAL4.7.717)14.2) | Bloomington Drosophila Stock Center | BDSC 9951, FlyBase FBti0101805 | |
| Genetic reagent (*D. melanogaster*) | Or22a-Gal4 (w[*]; P(w[ + mC] = Or22a-GAL4.7.717)14.21) | Bloomington Drosophila Stock Center | BDSC 9952, FlyBase FBti0101805 | |
| Genetic reagent (*D. melanogaster*) | Or33c-Gal4 (w[*]; P(w[ + mC] = Or33c-GAL4.F)78.3) | Bloomington Drosophila Stock Center | BDSC 9966, FlyBase FBti0101843 | |
| Genetic reagent (*D. melanogaster*) | Or35a-cd8GFP (w[*]; P(w[ + mC] = Or35a-Mmus\Cd8a.GFP)3/ TM3, Sb[1]) | Bloomington Drosophila Stock Center | BDSC 52624, FlyBase FBti0156834 | |
| Genetic reagent (*D. melanogaster*) | Or35a-Gal4 (w[*]; P(w[ + mC] = Or35a-GAL4.F)109.2A) | Bloomington Drosophila Stock Center | BDSC 9967, FlyBase FBti0101810 | |
| Genetic reagent (*D. melanogaster*) | Or35a-Gal4 (w[*]; P(w[ + mC] = Or35a-GAL4.F)109.3) | Bloomington Drosophila Stock Center | BDSC 9968, FlyBase FBti0101844 | |
| Genetic reagent (*D. melanogaster*) | Or42a-Gal4 (w[*]; P(w[ + mC] = Or42a-GAL4.F)48.3B) | Bloomington Drosophila Stock Center | BDSC 9970, FlyBase FBti0101811 | |
| Genetic reagent (*D. melanogaster*) | Or42b-Gal4 (w[*]; P(w[ + mC] = Or42b-GAL4.F)64.3) | Bloomington Drosophila Stock Center | BDSC 9971, FlyBase FBti0101812 | |
| Genetic reagent (*D. melanogaster*) | Or43b-Gal4 (w[*]; P(w[ + mC] = Or43b-GAL4.C)110t8.1) | Bloomington Drosophila Stock Center | BDSC 23894, FlyBase FBti0102047 | |
| Genetic reagent (*D. melanogaster*) | Or46a-Gal4 (w[1,118]; P(w[ + mC] = Or46a-GAL4.G)32.1.y) | Bloomington Drosophila Stock Center | BDSC 23291, FlyBase FBti0076800 | |
| Genetic reagent (*D. melanogaster*) | Or47a-Gal4 (w[*]; P(w[ + mC] = Or47a-GAL4.8.239)15.4A) | Bloomington Drosophila Stock Center | BDSC 9982, FlyBase FBti0101851 | |
| Genetic reagent (*D. melanogaster*) | Or49b-Gal4 (w[*]; P(w[ + mC] = Or49b-GAL4.F)80.1) | Bloomington Drosophila Stock Center | BDSC 9986, FlyBase FBti0101853 | |
| Genetic reagent (*D. melanogaster*) | Or59b-Gal4 (w[*]; P(w[ + mC] = Or59b-GAL4.C)114t2.2) | Bloomington Drosophila Stock Center | BDSC 23897, FlyBase FBti0102060 | |
| Genetic reagent (*D. melanogaster*) | Or59c-Gal4 (w[*]; P(w[ + mC] = Or59c-GAL4.C)129t1.1) | Bloomington Drosophila Stock Center | BDSC 23899, FlyBase FBti0102061 | |
| Genetic reagent (*D. melanogaster*) | Or65a-Gal4 (w[*]; Bl[1]/ SM1; P(w[ + mC] = Or65a-GAL4.F)72.1) | Bloomington Drosophila Stock Center | BDSC 9994, FlyBase FBti0101857 | |
| Genetic reagent (*D. melanogaster*) | Or67a-Gal4 (w[*]; P(w[ + mC] = Or67a-GAL4.C)137t3.3) | Bloomington Drosophila Stock Center | BDSC 23904, FlyBase FBti0102049 | |

*Appendix 1 Continued on next page*

*Appendix 1 Continued*

| Reagent type (species) or resource | Designation | Source or reference | Identifiers | Additional information |
|---|---|---|---|---|
| Genetic reagent (*D. melanogaster*) | Or67b-Gal4 (w[*]; P(w[ + mC] = Or67b-GAL4.F)68.3/TM6B, Tb[1]) | Bloomington Drosophila Stock Center | BDSC 9995, FlyBase FBti0101858 | |
| Genetic reagent (*D. melanogaster*) | Or67c-Gal4 (w[*]; P(w[ + mC] = Or67c-GAL4.C)116t3.2/CyO) | Bloomington Drosophila Stock Center | BDSC 23905, FlyBase FBti0102050 | |
| Genetic reagent (*D. melanogaster*) | Or71a-Gal4 (w[*]; P(w[ + mC] = Or71a-GAL4.F)30.4) | Bloomington Drosophila Stock Center | BDSC 23122, FlyBase FBti0101860 | |
| Genetic reagent (*D. melanogaster*) | Or83c-Gal4 (w[*]; P(w[ + mC] = Or83c-GAL4.F)73.3B) | Bloomington Drosophila Stock Center | BDSC 23131, FlyBase FBti0101829 | |
| Genetic reagent (*D. melanogaster*) | Or85a-Gal4 (w[*]; P(w[ + mC] = Or85a-GAL4.F)67.2) | Bloomington Drosophila Stock Center | BDSC 23133, FlyBase FBti0101830 | |
| Genetic reagent (*D. melanogaster*) | Or85b-Gal4 (w[*]; P(w[ + mC] = Or85b-GAL4.C)179t5.1) | Bloomington Drosophila Stock Center | BDSC 23911, FlyBase FBti0102053 | |
| Genetic reagent (*D. melanogaster*) | Or85d-Gal4 (w[*]; P(w[ + mC] = Or85d-GAL4.C)143t2.1) | Bloomington Drosophila Stock Center | BDSC 24148, FlyBase FBti0102066 | |
| Genetic reagent (*D. melanogaster*) | Or92a-Gal4 (w[*]; P(w[ + mC] = Or92a-GAL4.F)62.1) | Bloomington Drosophila Stock Center | BDSC 23139, FlyBase FBti0101867 | |
| Genetic reagent (*D. melanogaster*) | Or98a-Gal4 (w[*]; P(w[ + mC] = Or98a-GAL4.F)115.1) | Bloomington Drosophila Stock Center | BDSC 23141, FlyBase FBti0101868 | |
| Genetic reagent (*D. melanogaster*) | Orco-Gal4 (w[*]; P(w[ + mC] = Orco-GAL4.W)11.17; TM2/TM6B, Tb[1]) | Bloomington Drosophila Stock Center | BDSC 26818, FlyBase FBti0101150 | |
| Genetic reagent (*D. melanogaster*) | QUAS reporter (y[1] w[1,118]; P(w[ + mC] = QUAS-mCD8::GFP.P)5B/TM6B, Tb[1]) | Bloomington Drosophila Stock Center | BDSC 30003, FlyBase FBti0129937 | |
| Genetic reagent (*D. melanogaster*) | QUAS reporter (y[1] w[*]; PBac(y[ + mDint2] w[ + mC] = 10XQUAS-6XGFP)VK00018/CyO, P(Wee-P.ph0)Bacc[Wee-P20]) | Bloomington Drosophila Stock Center | BDSC 52264, FlyBase FBti0162759 | |
| Genetic reagent (*D. melanogaster*) | UAS reporter (w[*]; P(y[ + t7.7] w[ + mC] = 10XUAS-IVS-mCD8::GFP)attP40) | Bloomington Drosophila Stock Center | BDSC 32186, FlyBase FBti0131963 | |
| Genetic reagent (*D. melanogaster*) | UAS reporter (w[*]; P(y[ + t7.7] w[ + mC] = 15XUAS-IVS-mCD8::GFP)attP2) | Bloomington Drosophila Stock Center | BDSC 32193, FlyBase FBti0131935 | |
| Genetic reagent (*D. melanogaster*) | UAS reporter (w[*]; P(y[ + t7.7] w[ + mC] = 10XUAS-IVS-mCD8::RFP)attP2) | Bloomington Drosophila Stock Center | BDSC 32218, FlyBase FBti0131950 | |
| Genetic reagent (*D. melanogaster*) | UAS reporter (w[*]; P(y[ + t7.7] w[ + mC] = 10XUAS-IVS-mCD8::RFP)attP40) | Bloomington Drosophila Stock Center | BDSC 32219, FlyBase FBti0131967 | |
| Genetic reagent (*D. melanogaster*) | UAS reporter (y[1] w[*]; PBac(y[ + mDint2] w[ + mC] = 20XUAS-6XGFP)VK00018/CyO, P(Wee-P.ph0)Bacc[Wee-P20]) | Bloomington Drosophila Stock Center | BDSC 52261, FlyBase FBti0162758 | |

*Appendix 1 Continued on next page*

*Appendix 1 Continued*

| Reagent type (species) or resource | Designation | Source or reference | Identifiers | Additional information |
|---|---|---|---|---|
| Genetic reagent (*D. melanogaster*) | *UAS* reporter, sacculus experiments (*UAS-mCD8::GFP (2nd)*) | **Lee and Luo, 1999** | FlyBase FBti0012685 | |
| Genetic reagent (*D. melanogaster*) | *UAS* reporter, sacculus experiments (*UAS-mCD8::GFP (3rd)*) | **Lee and Luo, 1999** | FlyBase FBti0012686 | |
| Genetic reagent (*D. melanogaster*) | *Orco²* mutant (*w[*]; TI(w[ + m*] = TI)Orco[2]*) | Bloomington Drosophila Stock Center | BDSC 23130, FlyBase FBti0168777 | |
| Genetic reagent (*D. melanogaster*) | *Ir25a²* mutant (*w[*]; TI(w[ + m*] = TI)Ir25a[2]/CyO*) | Bloomington Drosophila Stock Center | BDSC 41737, FlyBase FBti0168524 | |
| Genetic reagent (*D. melanogaster*) | *Ir8a¹* mutant (*w[*] TI(w[ + mW.hs] = TI)Ir8a[1]; Bl[1] L[2]/CyO*) | Bloomington Drosophila Stock Center | BDSC 41744, FlyBase FBst0041744 | |
| Genetic reagent (*D. melanogaster*) | *Ir76b¹* mutant (*w[*]; Ir76b[1]*) | Bloomington Drosophila Stock Center | BDSC 51309, FlyBase FBst0051309 | |
| Genetic reagent (*D. melanogaster*) | Germline *Cas9* (*y[1] M(RFP[3xP3.PB] GFP[E.3xP3] = vas-Cas9) ZH-2A w[1,118]/FM7c*) | Bloomington Drosophila Stock Center | BDSC 51323, FlyBase FBti0154823 | |
| Genetic reagent (*D. melanogaster*) | Germline *Cas9* (*w[1,118]; PBac(y[ + mDint2] = vas-Cas9) VK00027*) | Bloomington Drosophila Stock Center | BDSC 51324, FlyBase FBti0154822 | |
| Genetic reagent (*D. melanogaster*) | Germline *Cas9* (*y[1] M(w[ + mC] = Act5C-Cas9.P)ZH-2A w[*]*) | Bloomington Drosophila Stock Center | BDSC 54590, FlyBase FBti0159182 | |
| Genetic reagent (*D. melanogaster*) | Double balancer (*19ADrok/FM7c; Pin/CyO*) | Potter lab stock | Derived from BDSC 6666, FBba0000009, FBal0013831, FBba0000025 | |
| Genetic reagent (*D. melanogaster*) | Double balancer (*y,w; Pin/CyO; Dh/TM6B*) | Potter lab stock | Derived from FBal0013831, FBba0000025, FBti0004009, FBba0000057, FBal0016730 | |
| Genetic reagent (*D. melanogaster*) | Double balancer (*y,w; S/CyO; Pr/TM6B*) | Potter lab stock | Derived from FBal0015108, FBba0000025, FBal0013944, FBba0000057, FBal0016730 | |
| Genetic reagent (*D. melanogaster*) | Single balancer (*y,w; +/+; Pr/TM6B*) | Potter lab stock | Derived from FBal0013944, FBba0000057, FBal0016730 | |
| Genetic reagent (*D. melanogaster*) | Single balancer (*y,w; S/CyO; +/+*) | Potter lab stock | Derived from FBal0015108, FBba0000025 | |
| Genetic reagent (*D. melanogaster*) | Wildtype (*w1118 IsoD1*) | Gift from Thomas R. Clandinin | Derived from FBal0018186 | |
| Genetic reagent (*Drosophila sechellia*) | *Wildtype*, genome Cousin Island, Seychelles | National *Drosophila* Species Stock Center | SKU:14021-0248.25 | |
| Genetic reagent (*Anopheles coluzzii*) | *Wildtype*, N'Gousso strain | Gift from Insect Transformation Facility (Rockville, MD) **Riabinina et al., 2016** | | |
| Chemical compound, drug | Mineral oil | Sigma-Aldrich | CAS# 8042-47-5, Cat# 330779-1L | |

*Appendix 1 Continued on next page*

*Appendix 1 Continued*

| Reagent type (species) or resource | Designation | Source or reference | Identifiers | Additional information |
|---|---|---|---|---|
| Chemical compound, drug | Ethyl acetate | Sigma-Aldrich | CAS# 141-78-6, Cat# 650528-1L, Cat# 270989-100ML | |
| Chemical compound, drug | Pentyl acetate | Sigma-Aldrich | CAS# 628-63-7, Cat# 109584-250ML | |
| Chemical compound, drug | Benzaldehyde | Sigma-Aldrich | CAS# 100-52-7, Cat# 418099-100ML, Cat# B1334-100G | |
| Chemical compound, drug | Ethyl butyrate | Sigma-Aldrich | CAS# 105-54-4, Cat# E15701-500M, Cat# E15701-25ML | |
| Chemical compound, drug | Hexanol | Sigma-Aldrich | CAS# 111-27-3, Cat# H13303-100ML | |
| Chemical compound, drug | E2-hexenal | Sigma-Aldrich | CAS# 6728-26-3, Cat# W256005-1KG-K | |
| Chemical compound, drug | Geranyl acetate | Sigma-Aldrich | CAS# 105-87-3, Cat# 173495-25G, Cat# 45896-1ML-F | |
| Chemical compound, drug | 2-Heptanone | Sigma-Aldrich | CAS# 110-43-0, Cat# 537683-100ML | |
| Chemical compound, drug | 1-Octen-3-ol | Sigma-Aldrich | CAS# 3391-86-4, Cat# O5284-25G, Cat# W280518-SAMPLE-K | |
| Chemical compound, drug | 2,3-Butanedione | Sigma-Aldrich | CAS# 431-03-8, Cat# 11038-1ML-F | |
| Chemical compound, drug | Phenylacetaldehyde | Sigma-Aldrich | CAS# 122-78-1, Cat# 107395-100ML | |
| Chemical compound, drug | Phenethylamine | Sigma-Aldrich | CAS# 64-04-0, Cat# 241008-50ML | |
| Chemical compound, drug | Propionic acid | Sigma-Aldrich | CAS# 79-09-4, Cat# 402907-100ML | |
| Chemical compound, drug | 1,4-Diaminobutane | Sigma-Aldrich | CAS# 110-60-1, Cat# D13208-100G | |
| Chemical compound, drug | Pyrrolidine | Sigma-Aldrich | CAS# 123-75-1, Cat# P73803-100ML, Cat# P73803-5ML | |
| Chemical compound, drug | P-cresol | Sigma-Aldrich | CAS# 106-44-5, Cat# 42429-5G-F, Cat# C85751-5G | |
| Chemical compound, drug | Methyl salicylate | Sigma-Aldrich | CAS# 119-36-8, Cat# M6752-250ML | |
| Chemical compound, drug | Paraffin Oil | ACROS Organics | CAS# 8012-95-1, Cat# 171400010 | |
| Chemical compound, drug | Water | SIGMA Life Science | CAS# 7732-18-5, Cat# W3500 | |
| Chemical compound, drug | Acetic acid | Fisher Scientific | CAS# 64-19-7, Cat# A38S | |
| Chemical compound, drug | Ammonium hydroxide | Fisher Scientific | CAS# 7664-41-7, Cat# A669S | |
| Chemical compound, drug | Cadaverine | Sigma-Aldrich | CAS# 462-94-2, Cat# 33211-10ML-F | |
| Chemical compound, drug | Hexanol | ACROS Organics | CAS# 111-27-3, Cat# AC43386 | |
| Chemical compound, drug | 2,3-Butanedione | ACROS Organics | CAS# 431-03-8, Cat# AC10765 | |
| Chemical compound, drug | Phenethylamine | ACROS Organics | CAS# 64-04-0, Cat# AC156491000 | |

*Appendix 1 Continued on next page*

*Appendix 1 Continued*

| Reagent type (species) or resource | Designation | Source or reference | Identifiers | Additional information |
|---|---|---|---|---|
| Chemical compound, drug | Propanal | ACROS Organics | CAS# 123-38-6, Cat# AC220511000 | |
| Chemical compound, drug | 1,4-Diaminobutane | ACROS Organics | CAS# 110-60-1, Cat# AC11212-250 | |
| Chemical compound, drug | All trans-retinal | Sigma-Aldrich | CAS# 116-31-4, Cat# R2500 | |
| Commercial assay or kit | HCR v3.0 | Molecular Instruments | | |
| Commercial assay or kit | In-Fusion Cloning | Clontech Labs | Cat# 639645 | |
| Commercial assay or kit | DNeasy Blood and Tissue Kit | QIAGEN | Cat# 69506 | |
| Software, algorithm | Fiji (ImageJ) | *Schindelin et al., 2012* | https://imagej.net/Fiji | |
| Software, algorithm | GraphPad Prism 8.0 | GraphPad Software | http://www.graphpad.com/ | |
| Software, algorithm | AutoSpike | Syntech | http://www.ockenfels-syntech.com/products/signal-acquisition-systems-2/ | |
| Software, algorithm | LabChart Pro v8 | ADInstruments | https://www.adinstruments.com/support/downloads/windows/labchart | |
| Software, algorithm | Adobe Illustrator CS6 | Adobe, Inc | https://www.adobe.com/products/illustrator.html | |
| Software, algorithm | Adobe Photoshop CS6 | Adobe, Inc | https://www.adobe.com/products/photoshop.html | |
| Software, algorithm | MacVector 16.0 | MacVector, Inc | https://macvector.com/ | |
| Software, algorithm | MEGA11 | *Tamura et al., 2021* | https://www.megasoftware.net/ | |
| Software, algorithm | Zen Black | Carl Zeiss Microscopy | https://www.zeiss.com/microscopy/us/products/microscope-software/zen.html | |
| Software, algorithm | Venn Diagram web tool | VIB/UGent Bioinformatics & Evolutionary Genomics | http://bioinformatics.psb.ugent.be/webtools/Venn/ | |
| Software, algorithm | Amira | Thermo Fisher Scientific | https://www.thermofisher.com/us/en/home/industrial/electron-microscopy/electron-microscopy-instruments-workflow-solutions/3d-visualization-analysis-software/amira-life-sciences-biomedical.html | |
| Software, algorithm | Python packages | *Schlegel et al., 2021* | https://github.com/schlegelp/navis, https://github.com/flyconnectome/fafbseg-py | |
| Software, algorithm | Q-Capture Pro 7 | Teledyne QImaging | https://www.qimaging.com/home | |

*Appendix 1 Continued on next page*

*Appendix 1 Continued*

| Reagent type (species) or resource | Designation | Source or reference | Identifiers | Additional information |
|---|---|---|---|---|
| Other | DAPI stain | Invitrogen | D1306 | IHC (1:10,000) |
| Sequence-based reagent | Orco_gRNA_FOR | This paper, Integrated DNA Technologies | PCR primers | TCCGGGTGAACT TCGCACAGTGCG GAGGGGGCAAG TTTTAGAGCT AGAAATAGCAAGTTA |
| Sequence-based reagent | Orco_gRNA_REV | This paper, Integrated DNA Technologies | PCR primers | TTCTAGCTCTAAAAC ACTTTATGGTGCT GGTGCAGCGACGTTA AATTGAAAATAGGTC |
| Sequence-based reagent | Orco_5HA_FOR | This paper, Integrated DNA Technologies | PCR primers | CCCTTACGTAACGCGTCAGCTT GTTTGACTTACTTGATTAC |
| Sequence-based reagent | Orco_5HA_REV | This paper, Integrated DNA Technologies | PCR primers | CGCGGCCCTCACGCGTCTTGA GCTGTACAAGTACCATAAAGT |
| Sequence-based reagent | Orco_3HA_FOR | This paper, Integrated DNA Technologies | PCR primers | GTTATAGATCACTAGTCTCAG TACTATGCAACCAGCAATA |
| Sequence-based reagent | Orco_3HA_REV | This paper, Integrated DNA Technologies | PCR primers | AATTCAGATCACTAGTGTTTT ATGAAAGCTGCAAGAAATAA |
| Sequence-based reagent | Ir8a_gRNA_FOR | This paper, Integrated DNA Technologies | PCR primers | TCCGGGTGAACTTCGTTTGT TTGTTCGGCCATGTGTTTTAG AGCTAGAAATAGCAAGTTA |
| Sequence-based reagent | Ir8a_gRNA_REV | This paper, Integrated DNA Technologies | PCR primers | TTCTAGCTCTAAAACCTGTGG GAGTCAGAGGCACCGACGT TAAATTGAAAATAGGTC |
| Sequence-based reagent | Ir8a_5HA_FOR | This paper, Integrated DNA Technologies | PCR primers | CCCTTACGTAACGCGtCTATT GGCTATTCGTCGTACTCATGC |
| Sequence-based reagent | Ir8a_5HA_REV | This paper, Integrated DNA Technologies | PCR primers | CGCGGCCCTCACGCGtCTCCA TGTAGCCACTATGTGAGTCAGAT |
| Sequence-based reagent | Ir8a_3HA_FOR | This paper, Integrated DNA Technologies | PCR primers | GTTATAGATCACTAGtGTTTCTT GTCGCACCTAATTAACAAGTG |
| Sequence-based reagent | Ir8a_3HA_REV | This paper, Integrated DNA Technologies | PCR primers | AATTCAGATCACTAGtCATAC TTAAGCTCCTTGAGGTCCAGC |
| Sequence-based reagent | Ir76b_gRNA_FOR | This paper, Integrated DNA Technologies | PCR primers | TCCGGGTGAACTTCGCAGT GATGCGAACTTCATAGTTTT AGAGCTAGAAATAGCAAGTTA |
| Sequence-based reagent | Ir76b_gRNA_REV | This paper, Integrated DNA Technologies | PCR primers | TTCTAGCTCTAAAACCGGCG GCCCTCTTTCAATACGACG TTAAATTGAAAATAGGTC |
| Sequence-based reagent | Ir76b_5HA_FOR | This paper, Integrated DNA Technologies | PCR primers | CCCTTACGTAACGCGtACCAATG AATCCTTGTCCATGCTAAA |
| Sequence-based reagent | Ir76b_5HA_REV | This paper, Integrated DNA Technologies | PCR primers | CGCGGCCCTCACGCGtCTC GGTGTAGCTGTCTTGAAGGAA |
| Sequence-based reagent | Ir76b_3HA_FOR | This paper, Integrated DNA Technologies | PCR primers | GTTATAGATC ACTAGtGCCTA ATTGGAATACC TTCTACATAATGGA |
| Sequence-based reagent | Ir76b_3HA_REV | This paper, Integrated DNA Technologies | PCR primers | AATTCAGATCACTAGtGGCAA GGCACAAAATAAAACGAAG |
| Sequence-based reagent | Ir25a_gRNA_FOR | This paper, Integrated DNA Technologies | PCR primers | TCCGGGTGAACTTCGCCGGA TACTGATTAAAGCGGTTTTA GAGCTAGAAATAGCAAGTTA |
| Sequence-based reagent | Ir25a_gRNA_REV | This paper, Integrated DNA Technologies | PCR primers | TTCTAGCTCTAAAACAGTG CTCATTTTACCATAATCGA CGTTAAATTGAAAATAGGTC |
| Sequence-based reagent | Ir25a_5HA_FOR | This paper, Integrated DNA Technologies | PCR primers | CCCTTACGTAACGCGtTGC ATGACTTCATTGACACCTCAAG |
| Sequence-based reagent | Ir25a_5HA_REV | This paper, Integrated DNA Technologies | PCR primers | CGCGGCCCTCACGCGtGAA ACGAGGCTTAAACGTAGCTGGATATT |

*Appendix 1 Continued on next page*

*Appendix 1 Continued*

| Reagent type (species) or resource | Designation | Source or reference | Identifiers | Additional information |
|---|---|---|---|---|
| Sequence-based reagent | Ir25a_3HA_FOR | This paper, Integrated DNA Technologies | PCR primers | GTTATAGATCACTAGtAATAT TATGGTTAAGTGAGCTCTCGG |
| Sequence-based reagent | Ir25a_3HA_REV | This paper, Integrated DNA Technologies | PCR primers | AATTCAGATCACTAGtCAAA GCTAAGTTCATCGTCATAGAGAC |
| Sequence-based reagent | Orco_Seq_FOR | This paper, Integrated DNA Technologies | PCR primers | GATGTTCTGCTCTTGGCTGATATTC |
| Sequence-based reagent | Ir8a_Seq_FOR | This paper, Integrated DNA Technologies | PCR primers | CATCGACTTCATCATCAGGCTTTCG |
| Sequence-based reagent | Ir76b_Seq_FOR | This paper, Integrated DNA Technologies | PCR primers | CAACGATATCCTCACGAAGAACAAGC |
| Sequence-based reagent | Ir25a_Seq_FOR | This paper, Integrated DNA Technologies | PCR primers | CGAAAGGATACAAAGGATACTGCAT |
| Sequence-based reagent | HACK_Seq_REV | This paper, Integrated DNA Technologies | PCR primers | TGTATTCCGTCGCATTTCTCTC |
| Sequence-based reagent | IVS-FOR | This paper, Integrated DNA Technologies | PCR primers | TGGGTTGGACTCAGGGAATA GATCTAAAAGGTAGGTTCAACCACT |
| Sequence-based reagent | EcoRI-SV40-REV | This paper, Integrated DNA Technologies | PCR primers | GCTTACGTCAGAATTCAGA TCGATCCAGACATGATAAGA |
| Sequence-based reagent | Tsp42Ej_FOR | This paper, Integrated DNA Technologies | PCR primers | GAGAAGTCGTTTCCCATAACACCCT |
| Sequence-based reagent | Tsp42Ej_REV | This paper, Integrated DNA Technologies | PCR primers | GAGGAGCAGTTTTCGGAGTCGCCTTC |
| Sequence-based reagent | *D. melanogaster* Orco | This paper, Molecular Instruments | RNA in situ HCR probe set | See Appendix 2 'In situ probe sequences' |
| Sequence-based reagent | *D. melanogaster* Ir25a | This paper, Molecular Instruments | RNA in situ HCR probe set | See Appendix 2 'In situ probe sequences' |
| Sequence-based reagent | *D. melanogaster* Ir40a | This paper, Molecular Instruments | RNA in situ HCR probe set | See Appendix 2 'In situ probe sequences' |
| Sequence-based reagent | *D. melanogaster* Ir51a | This paper, Molecular Instruments | RNA in situ HCR probe set | See Appendix 2 'In situ probe sequences' |
| Sequence-based reagent | *D. melanogaster* Ir60a | This paper, Molecular Instruments | RNA in situ HCR probe set | See Appendix 2 'In situ probe sequences' |
| Sequence-based reagent | *D. melanogaster* Ir62a | This paper, Molecular Instruments | RNA in situ HCR probe set | See Appendix 2 'In situ probe sequences' |
| Sequence-based reagent | *D. melanogaster* Ir76a | This paper, Molecular Instruments | RNA in situ HCR probe set | See Appendix 2 'In situ probe sequences' |
| Sequence-based reagent | *D. melanogaster* Ir76b | This paper, Molecular Instruments | RNA in situ HCR probe set | See Appendix 2 'In situ probe sequences' |
| Sequence-based reagent | *D. melanogaster* Ir93a | This paper, Molecular Instruments | RNA in situ HCR probe set | See Appendix 2 'In situ probe sequences' |
| Sequence-based reagent | *Anopheles coluzzii* Orco | This paper, Molecular Instruments | RNA in situ HCR probe set | See Appendix 2 'In situ probe sequences' |
| Sequence-based reagent | *A. coluzzii* Ir25a | This paper, Molecular Instruments | RNA in situ HCR probe set | See Appendix 2 'In situ probe sequences' |

## Appendix 2

### In situ probe sequences

The following coding sequences (no introns or UTRs) from FlyBase were used by Molecular Instruments, Inc (Los Angeles, California) to produce custom probe sets:

### Antenna

#### *Ir76b* (1911 bp; Probe set ID PRH997)

AUGGCCACUGGCAUCGAGCUGCUGGUGGCCGCCGCCCUCUGUGUCGCCUGUCCGCC
GCUGAACGAUUCCCCGCCGACGAACCUAAUCCAAAUGGGCGAGAAUGGCACUCUUU
CCCCGGUCACCGAGCUGCCCAUGGAUGUGGACGCAUCGGAAGCUGGAUUCGAUGCG
GAUGCCCCCGUGGAGACGCUGGAGACCAUUAACAGGAAGAAGCCGAAGCUGCGGGA
GAUGCUCGAUUGGAUCGGCGGCAAGCACCUGCGCAUCGCUACCCUGGAGGACUUUC
CGCUCAGCUACACCGAGGUCCUGGAGAACGGCACCCGUGUGGGGCACGGAGUCUCC
UUUCAGAUCAUCGACUUCCUCAAGAAGAAGUUCAACUUCACCUAUGAAGUGGUCGU
GCCCCAAGAUAACAUCAUCGGCUCGCCUAGCGACUUUGAUCGCAGUCUCAUCGAGA
UGGUAAACAGCAGUACGGUGGACUUGGCGGCGGCCUUCAUACCCUCGCUCUCUGAC
CAGCGCAGCUUCGUCUACUACUCCACAACGACGCUGGACGAGGGCGAAUGGAUAAU
GGUGAUGCAGCGUCCCCGCGAGUCGGCUAGUGGGUCCGGACUGCUUGCGCCCUUCG
AGUUCUGGGUGUGGAUCCUGAUCCUCGUCUCGCUGCUGGCCGUGGGGCCGAUCAUC
UACGCGCUGAUCAUCCUGCGAAAUCGGCUGACCGGCGACGGCCAGCAGACGCCCUA
CUCCCUGGGCCACUGCGCUUGGUUCGUCUACGGAGCGCUGAUGAAGCAGGGCAGCA
CCCUGUCGCCCAUUGCAGACUCGACGCGGCUGCUCUUUGCCACCUGGUGGAUUUUC
AUCACGAUACUGACGUCCUUCUACACGGCCAACUUGACCGCCUUCCUGACCCUUUC
CAAGUUCACGCUGCCGUACAACACGGUCAACGAUAUCCUCACGAAGAACAAGCACU
UUGUGUCCAUGCGGGGCGGUGGAGUGGAGUACGCCAUUCGAACGACCAAUGAAUCC
UUGUCCAUGCUAAACCGAAUGAUCCAGAACAACUACGCCGUAUUCUCGGACGAGAC
CAACGACACCUACAAUCUGCAGAACUACGUGGAAAAGAAUGGCUAUGUUUUUGUGA
GGGAUCGGCCGGCGAUAAACAUAAUGUUGUACAGGGACUACCUGUACCGCAAAACC
GUGAGCUUUAGCGACGAGAAGGUCCACUGUCCGUUUGCCAUGGCCAAGGAGCCGUU
CCUGAAGAAGAAGAGGACCUUUGCCUAUCCCAUCGGAUCGAAUUUGAGCCAAUUAU
UUGACCCGGAGCUGCUACACCUGGUGGAAUCUGGAAUCGUGAAGCACCUGUCUAAG
AGAAAUCUGCCCAGUGCCGAGAUCUGUCCGCAGGAUCUCGGCGGAACGGAGCGGCA
GCUGAGGAACGGCGACCUAAUGAUGACCUACUACAUCAUGCUUGCCGGUUUCGCCA
CCGCACUGGCCGUCUUCAGCACGGAGCUAAUGUUCCGGUACGUCAAUAGUCGCCAG
GAGGCGAAUAAGUGGGCGCGCCACGGAAUCGGACGAACGCCCAACGGCCAGUCGGU
GGCUCCAUCCCGGUGGCUCCGUGGCUGGAGGCGAUUGAACAGUGGACAUGGGCAGC
UCCUGGGCGCCUCCACCCACGGCCAAAAUGUCACUCCUCCGCCGCCGUACCAGAGC
AUCUUCAACGGCGGCAGUCACGGAGAUCCACUGAAUCGCUGGCGACGUCCCCUCGC
AAACGGAAACGCCCUUGGCAAUGGUGUCCUCCUGGGCGGCGAUUCUGAAGGUGGUG
UACGGCGCCUGAUCAACGGACGCGACUACAUGGUAUUCCGCAAUCCAAAUGGUCAA
AGCCAGCUCGUGCCGGUUAGAUCGCCCUCGGCGGCCCUCUUUCAAUACAGCUACAC
UGAGUAG

#### PALP

#### *Ir40a* (2203 bp; Probe set ID PRH239)

AUGCAUAAGUUUCUGGCAUUGGGCCUGCUGCCCUACCUUUUGGGAUUGCUAAACAG
CACAAGGCUGACUUUUAUUGGUAACGAUGAGUCAGACACUGCAAUAGCGCUCACCC
AAAUUGUAAGAGGCUUGCAACAAUCUUCUCUUGCCAUAUUGGCGCUACCAAGCCUC
GCUCUAUCUGAUGGAGUUUGCCAGAAAGAGCGCAACGUUUAUCUUGACGAUUUUCU
GCAGCGUCUUCAUCGCAGUAACUACAAGUCGGUGGUAUUCAGCCAGACGGAGCUCU
UUUUUCAACACAUUGAGGAAAACCUUCAAGGUGCAAACGAGUGCAUCAGCCUGAUU
UUGGACGAGCCCAACCAGCUGUUGAAUAGCCUCCACGAUCGACAUCUCGGACAUCG
CUUAAGCCUAUUUAUUUUCUAUUGGGGAGCACGCUGGCCACCCAGCUCCCGUGUAA
UUCGUUUUAGAGAGCCGCUUCGAGUGGUAGUCGUAACUCGUCCUCGCAAGAAGGCC
UUCCGCAUUUACUACAACCAGGCUAGACCCUGUAGCGACAGUCAGCUACAGUUGGU

UAAUUGGUACGACGGCGAUAACCUUGGUCUGCAACGAAUUCCCCUCCUCCCGACUG
CAUUAUCCGUGUACGCCAACUUUAAAGGUCGUACCUUUCGGGUGCCCGUAUUUCAU
UCUCCGCCGUGGUUUUGGGUAACGUAUUGCAAUAACAGCUUCGAAGAGGACGAGGA
GUUUAACAGCCUAGACAGCAUAGAGAAGAGAAAGGUUCGGGUCACUGGUGGCCGCG
AUCACCGCCUACUCAUGCUGCUAUCUAAGCAUAUGAACUUUCGGUUUAAGUAUAUC
GAAGCACCCGGUCGAACCCAGGGCUCAAUGAGGUCAGAAGAUGGCAAGGAUUCGAA
CGACAGUUUCACAGGGGGCAUUGGAUUGCUGCAAAGUGGACAAGCUGACUUUUUUU
UGGGAGAUGUCGGUCUAAGCUGGGAACGGCGGAAGGCCAUCGAGUUCUCUUUUUUC
ACACUGGCUGAUUCAGGAGCGUUUGCUACACACGCUCCCAGACGCCUUAAUGAGGC
CCUGGCGAUUAUGCGCCCGUUUAAGCAAGACAUCUGGCCCCAUCUAAUCCUUACGA
UAAUUUUCUCCGGCCCUAUUUUUUAUGGCAUUAUUGCCCUGCCUUAUAUUUGGCGU
CGACGAUGGGCGAACUCAGAUGUUGAACAUCUCGGAGAAUUAUAUAUCCAUAUGAC
GUACUUAAAAGAGAUAACCCCACGCUUAUUAAAGCUCAAACCCAGAACUGUGCUGU
CUGCCCACCAGAUGCCCCAUCAACUUUUUCAGAAGUGCAUAUGGUUCACUUUACGU
CUGUUUUUAAAACAAUAUGGCAUGCAAUGAACUACAUAACGGAUACCGAGCCAAGU
UUUUGACCAUAGUGUAUUGGAUAGCAGCGACCUAUGUUUUGGCCGAUGUAUAUUCA
GCUCAACUGACCAGCCAAUUUGCACGUCCAGCUCGCGAGCCACCAAUCAAUACUCU
UCAGCGCCUGCAAGCAGCGAUGAUUCAUGACGGUUACCGGCUAUAUGUGGAGAAGG
AAAGCAGUUCAUUGGAGAUGUUGGAGAAUGGGACAGAACUGUUUCGUCAGCUUUAU
GCUCUGAUGAGGCAGCAGGUGAUCAAUGACCCUCAAGGAUUUUUUAUUGACUCUGU
GGAAGCGGGAAUUAAACUAAUUGCAGAGGGCGGCGAGGACAAGGCAGUACUCGGAG
GGCGUGAAACACUGUUUUUCAACGUUCAGCAAUACGGAUCAAACAACUUUCAGCUC
AGUCAAAAACUUUACACUCGUUAUUCGGCUGUGGCUGUUCAAAUCGGAUGUCCCUU
UCUAGGUAGCCUCAAUAAUGUCUUGAUGCAGUUGUUUGAGAGCGGAAUCCUAGAUA
AGAUGACCGCUGCCGAAUACGCAAAGCAGUACCAGGAGGUAGAAGCCACGAGAAUA
UACAAGGGCAGCGUGCAGGCGAAAAACAGUGAGGCUUACAGUCGAACCGAAAGCUA
UGACAGCACGGUUAUCAGUCCGCUUAAUCUACGAAUGCUGCAGGGCGCUUUUAUCG
CUCUCGGAGUUGGUUCAUUGGCUGCAGGUGUAAUUUUGCUGUUAGAGAUAGUAUUU
AUAAAACUGGAUCAAGCGCGAUUGUGGAUGCUGUGCUCACGGCUGCAAUGGAUUAG
AUAUGACAGGAAAGUGUAA

**_Ir51a_ (1830 bp; Probe set ID PRD951) – sequence derived from Potter lab _wildtype_ strain, which has no predicted premature stop codons.**

AUGCAAGGAUUUCAAGAAGCCAAUGCACAGUUAACCACCAUGUACAACGUUCUGGU
AUUGUUCCUAUUGCUUUUCACUCGUGCCCAGAUGGAACCCCAUAGAAGAGGUCACA
ACAUGACUUUGCUGAGAUCGGUGCUGACAGUCAUCCGCGGCAGGGAGAAUUGGAAA
AAUACCCCCAUCUUCUUAGGCGGACAUUGUAAUUCGGAUGACCUGAAUAACUUGAU
GAGUUGGCUUCAAAAUACGAUGGAAGUAACUUGUCAUACGGUGGAUACAUCUACUU
CAGCCAAAAACGAAAACGCUCUCGGGUCACUUUAACAUAAAUGCGGAUAAUUCGCUU
GGUUUGUUGUUUUGCCAAAGCUCCCAUGAAUUGAUUUGGUUUAAUAUGGAUAAGAG
ACUUCGGCGAUUACGUGGUAUUCGCCUUAUUGUGAUUCUAUCAGACAAACGAAGUU
CAUCCAGCAAAGCUAUAAUGAGCACGUUUAAGAGACUGUGGCACUUUCAGUUCCUC
AGAGUUCUUGUUCUGCACAGAGAUCAGAUUUAUUCGUAUACUCCUUAUCCAGUCAU
UCGAUUCUUUAAGCUUGAUAGUGACGUCUAUCCGCUUUUCCCACCGAGUGCAAAAA
AUUUCCAAGGCUAUGUGGUGUCUACCCCAGUGGAAAACGACAUUCCGCGAGUGUUU
UUCGUGAAGGACAAGAAACCGGACGCAAGCAGAUCAGAGGAUUUGGAUAUCGCAC
AUUUGUGGAGUACCUGCAUCGCUACAAUGCCUCUUUACAUGUCAGUAAUUCUCAGC
AGGAGCAUGCCAUCAACAGCAGCGUGAAUAUGGGUCGGAUAAUUAAUCAGAUUGUG
GAUGGCCAAUUAGAAAUCUCCCUGCAUCCGUAUGUAGACGUUCCGGAAAAUAUGGG
AGAUAACAGUUAUCCCCUUCUAAUAGCUAGCAAUUGUCUUAUUGUUCCGGUCAGGA
ACGAGAUAUCUCGCUACAUGUAUCUACUAUUGCCUCUCAACCAAUCAAGCUGGAUA
CUACUGCUCGGUUCUGUAAUUUAUAUCAGCGGAGUGCUCUACUACAUUCAGCCUGG
UCUGCUGCACCGCACCUGGGAUCAGCGGAUUGGCCUCAACAUCCUAGAUAGCAUCA
GCCGAAUAAUUAAUAUAUGCUCUCCCUCCAGGAUUUACAACCCAUCCCUGAGGUAU
UUUAUAGUUUCGGUGCACCUUAGUAUUCUGGGCUUUGUGGUGACCAACCUCUACAG
CAUUAUGUUGGGCAGCUUCUUCACCACCUUGGUAGUGGGCGAGCAGGUGGACAGCA

UGCAGCAGCUCAUCCAGCAACAGCAAAAGGUACUUGUAAAAUAUUACGAAGUCAGU
ACAUUCUUACGGCAUGUGGAACCAGAUUUGGUGGACGGAGUAGCGCAACUAUUAGU
GGGCGUGAAUGCCAGUGAGCAGGUGUCCGCUCUUCUGGGAUUCAAUCGGAGCUAUG
CCUACCCCUUCACCCUGGAGAGAUGGGAGUUCUUCUCACUACAACAGCAAUACGCC
UUCAAGCCAAUCUUCAGAUUCUCAUCGGCAUGCCUGGGCUCCCCGAUUAUAGGCUA
UCCCAUGAAAGUGACUGCCACCUGCAGUCGUCAUUGAACAUGUUCAUCAUGCGGA
UCCAGGCCGCAGGACUUCUGCGGCACUGGGUAGUAUCCGAUUUUAACGAUGCAAUG
CGCGCUGGCUAUGUUCGACUUCUGGAAAACUUCCUAGGAUUUCACUCGUUAGAUGU
CGAUUCCUUGCGCCUGGGGUGGGCAGUACUCCUAUGUGGGUGGCUGCUAUCCACCU
UGAUUUUUCUUUGCGAGCGCUGGCGCUUUUACCACUAA

### *Ir60a* (2151 bp; Probe set ID PRD952)

AUGUGGUGUAAUAAUCCGGGUCUCAUCAUCAUCAUCUUUCUGGGGGCAGAUUUUAAA
CCUGUGCCAGGGGAUAGUGAAUCUUUCGAAUGAGACGGCCAACACGGUCAUCUUCA
UGCUGCCCGAAAAGGACUUGGGUCCCGAUGUGUGGAAGGCUGGAGUGGGCUGCCUC
GAUAGCUUCGCCCAAAUCUUCUUCUUUCGCAACCCGAAGGAGCGCUUCACCAGAGC
CUAUAACUUGAUGCUGGUGCACGCCUUUCAUCUGUCCAGUCCGGCGGAUCAGAUCC
AGGAAGGAUUCAGCAAACUGAUCAACGAAGCAGUCACGAAUCCAGGGCCGCCGGAC
AGGGAGGAACUCUUCCAAAUGCGCGUGGCCAGUGAUUACAACAUCACGAAUGGGAC
CGAAGACAAGGGAGAGCUGAUCCUAGCGGACAAUUAUGUGAUAGUGGUGGACUCGG
UGGAUCGAUUAAAAGAGCUAAUGAAGAAAAAAUUGUCGAAAUGCGAUCCUGGAAU
CCAGGAGCGCGUUUUCUGGUGCUCUUUCAUAAUGCGACUUGUCGGAAUCGACCGCU
UGGCGUAGCCUCCAAUAUUUUCAAGGACCUCAUGGAAAUGUUUUACGUGCACCGAG
UUGCUCUUCUCUAUGCCAACUCCACCAUGAACUACAAUCUGCUGGUCAAUGAUUAC
UACAGCAAUGUAAACUGCAGGAUUCUGAAUGUCCAGAGCGUGGGCCAGUGCCACGA
UGGCAAACUUUACCCCAAUAAUGCUGUCGUUAAGGCCUCCAUGCAGGACUACGUAU
CAGGAUUCAGUCCCAGGAACUGCACCUUUUUUGCCUGUUCCUCCAUCUCUGCUCCC
UUUGUGGAGGCCGACUGUAUCCUGGGACUAGAGAUGAGGAUCCUGGGGUUCAUGAA
AAAUCGACUGAAAUUCGAUGUAAACCAAACCUGCAGCCUGGAGUCACGUGGUGAAA
UGGAUGGCCCAGCUAACUGGACUGGAUUACUGGGGGAAAGUCCAGAACAACGAGUGC
GACUUUGUCUUCGGCGGCUAUUAUCCGGACAACGAGGUGGCGGACCAUUUUUGGGG
AUCCGAUACCUAUCUGCAGGAUGCGCACACGUGGUACAUAAAAAAUGGCCGACAGAA
GACCCGCCUGGCAGGCGUUGGUGGGUAUUUUCGAAGCCUACACCUGGAUUGGAUUC
AUCCUGAUUCUAAUAAUCAGCUGGCUGUUCUGGUUCACCCUGGUUAUGAUUCUUCC
GGAGCCGAAGUAUUACCAACAGUUGAGUCUUACGGCCAUUAAUGCCCUGGCCGUCA
CCAUAUCGAUAGCCGUCCAGGAACGACCCAUUUGCGAGACGACGAGGCUGUUCUUC
AUGGCCCUGACUUUGUAUGGCCUGAACGUGGUGGCUACAUACACGUCCAAGAUGAU
AGCCACCUUCCAGGAUCCCGGCUACCUUCACCAGCUGGACGAGCUGACGGAAGUGG
UGGCCGCGGGUAUUCCGUUUGGCGGCCACGAGGAGAGUCGCGACUGGUUCGAGAAU
GACGACGACAUGUGGAUCUUUAACGGAUACAACAUUUCGCCGGAGUUCAUUCCGCA
AUCGAAGAACCUGGAGGCGGUGAAGUGGGGCCAGCGGUGCAUCCUGAGCAACCGGA
UGUACACGAUGCAGAGUCCCCUGGCGGACGUCAUAUACGCCUUUCCCAACAACGUU
UUCAGCAGUCCGGUGCAGAUGAUCAUGAAGGCGGGAUUCCCCUUUCUCUUCGAGAU
GAACAGCAUCAUCCGCCUCAUGCGCGACGUGGGCAUUUUUCAGAAGAUCGACGCGG
ACUUUAGGUACAACAACACGUACCUCAACAGGAUCAACAAGAUGCGCCCCCAGUUC
CCGGAAACGGCUAUCGUCCUGACCACGGAGCACCUGAAGGGACCCUUCUUCAUCCU
GGUGGUGGGCAGCUGCUGGGCCGCCCUCACGUUCAUUGGCGAGCUAAUAAUCCACA
GAUGGCGAACCCAGCUGGUCAGCACGAGCGAGCAGCAGGACAGGAGGAGUGACAAG
AGGAGGAGGAGGAGGAGGAGGAGGAAGCCAGAAAAGGAUAACCGCUGGCAGCGACA
AGUUCAAGUGGCUCCAGUCGUUCGAUUUACGCCAGUCAAACGACGCAAAGUUUUCC
AGGGACAAACAAGUCAAAAGUAA

### *Ir62a* (1821 bp; Probe set ID PRD953)

AUGUAUCUUCAGUUUUUGUUCGCGCUGUUUUUGUCGCGCUACCAAAUCGUGGCCAC
CGAAAACUUUGACCGCGCUUUCGAGCUGGCUCUGUUUCGGACAGGAUCGGCCGCG
UGCAUCGCUUGCAUGCCAUCACUAUAGUGAACAGCCUGGGAUCUGUUGAUCCCAGC
UAUCUGGACGAUCUGCAUCGCGGCCUGAUGUGCAACAGCAGCAAUCACUUCUACAU

GCUGCCGCAGAUGACGGCCACGGACAAGGAUUCCUCGCACGUCCACUUCAGCAGCC
UGCAGGAUGAGGAGACCAUAUAUUUGGUCUUCGCCAGGGAUUCCAAGGAUGCGGUG
AUUUACUUGCAGGCCGAAAGAGCCAGAGGAAGGCGUUACACGAGGACUAUGUUCCU
GCUAAGAAAGCAGGAGUCGCAAAAGGACAUUAAAUACUUCUUUGAACUCCUGUGGA
AACUGCAGUUCCGGAGCGCUUUGGUAGUGGUGGCCGCCAGAAACUUCUACCAAAUG
GAUCCAUAUCCCACUGUUAGGGUGAUACGCAUGCGAAGAUUAUCCUCCUAUGAUCC
GCAUCAUGUUUUUCCGCCUGCAAAUCGAAAAAAUUUCAGAGGCUACAGGAUGCGAU
UGCCCGUACAGCAGGAUGUGCCCAAUACUUUUUGGUACAAGAAUCGCAGGACGAAG
GCCUGGGAACUGGCCGGAUUGGGCGGGAUCUUGAUCAAUCAACUGAUGAUGCACCU
GAACGUGACCAUGGAUUUGUUUAGAUUUGAAGUAAAUGGCUCUUCAUUGCUGAAUA
UGGCUGCGCUCACGGAUCUGAUUGUAAAGGGCAAGGUGGAGCUGAGUCCGCACUUG
UACGACACACUGCAAUCAAACACCAGUGUGGACUACUCAUAUCCCACGCAGGUGGC
ACCGCGCUGCUUCAUGAUCCCAUUGGACAAUGAAAUUUCGAGGAGCCUCUAUGUGU
UCCUGCCAUUUAGUUUAACCAUGUGGCUGUGCUUGCUGUUUGUUCUGCUCGUAGUG
CACUUCGUGUAUGUGCGCCGCCUGAUUCCAGACGGACAUUUUUGGGCCAUACUAGG
AGUGCCUGGUGCUGGUCAAGUCAGAUAUGGCAACCGCAAGCCGGUCAGGCGCUUUA
GUACCUUUCUGAUCCUCUUCGGAAUAUUCAUUCUUGGGCAAACGUACAGCACCAAA
CUAACUUCAUCUCUGACCGUGACUCUGAUCCGAAGACCAGACAACAGCUUGGAGGA
GCUAUUCCUGCUGCCGUAUCGCAUUCUAGUGCUGCCCACGGAUGUGUACGCCAUCG
UUGAUUCCCUGGGGUCAUGCCGAGCAGUUUAGCACAAAGUUUAGCUGCACGGAUGCC
GAGAACUUCAGCCAGAAGAGAAUCUCCAUGCAUCCGGAGUAUAUUUACCCCAUCAG
UACAAUUCGUUGGCGUUUCUUCGAUAUGCAGCAGCGCUUUCUGAGAAAGAAACGCU
UCUAUUUCUCCAAGAUUUGCCACGGCUCCUUUCCCUAUCAGUACCAGCUCAGGGUA
GACUCACAUCUCAAGGAUGCCCUCCAUCGAUUCCUGCUGCAUGUGCAGCAAGCCGG
GCUGCAUGAUCUUUGGCUGGAUACGUGCUAUCGCAAGGCCCAUCGCAUGGGCUAUC
UCAAGGAUUUCUCCACACUGGCGGAGCUGGAGGAGAAACUGAGGCUCCGCCCACUG
GCGCUCAAUCUGCUGGUGCCCGCGUUCAGUCUGUUUCUGUGCGGAAUGUUGGGCAG
CGGAAUCGCCUUCCUGGUGGAGAUCCGUCACAGCUUCGGGUGCAGACAAAAGCCAC
CGUCCAUUAACCGCAACCCCGGGGAUUAA

*Ir76a* (3281 bp; Probe set ID PRH240)

AUGGAAAACUUGCUGGUAGAGUCAUAUUACUUUAGCACUGUGCUUAGUUUUUUUGC
UCAGCAGUUCUUCGCGGACUCGCAUGCCACUUGUAUUUUUGGCACCCGGCGUUUG
AUUUUCGUCUGGAAACGGUUCAUCCAAUGCCAUUAAUAAUCAUGGACUGGCACAGA
UGGGCCAAUCGUUCCGAUCAAGAUGUUUAUGAUUACAAAAUAAAGGAGGAUGAGUU
CGAGGGGAAGGGAAUCCCCUACAACGAUUGGACUCUUAGACUGACUGUUGCCAUCG
AAAGGUCGCACUGUGAGCUAGUCAAAGAUAGUAUCACACUUAGAGUAGGUAUCCCG
CUGACAGUGAUGACAUGUGGCUCCAUCAAAACUCACUUGCAAGUGUUUAUCAUUCC
UGAACUUAUCUGUGAACUUUUUCCGAGUAUCCGGUGUCUUAAUGAAAUCGAGCGUU
AGCAAAAGCCGGUCUUUCCGAUCCGCUCCUGCAUCAGCCACAUUGUCCAAAAAUUG
AUUUACAUCCUCGAUAGGCAUCUUCCAAUGUUUUCCCAAUCUCAGAUGAUUUAGGU
UAUAGCAAUUUUGUACUAGCAACAUAAGUUGGCGACUUGGGUUUGCAGGUGAGUCA
AUGCAAUCCAGGGCCAGGCACUCGAGAUCUUUCAACUGGCCCAACUGGGACACCGA
UUGAGCUGGCCAGUUAUCACAGUCCAGAGCGUUGAGUUGCCUAAUCGCCAGGAUGU
GAUGCACUUGGUCACCUCUAACGCGGACGGAUAUGAGCUGUAGCUUUUGGAGGUUA
CUAACUCUUUUGGCCACCGACCUAUAAAAGUCGUUGUCCACUUCUGCUGUCCAACG
UGAGAUUAGGCGCAGGGAUCGGAGACCUUUAAAUUCACCCAACUCCGGGAAGCUCA
UAUCAAAGUUUUCGAGAGUAAGGUGAUCCAAGUUGGGAAACUUAUCUGCCAACAAC
GGGACCUGGUGAGAUCUCCUGCGAUCCGGUCCGAAAUUGAGAAGCAGUCGCUUCAG
GGAAGCCAUACUUUCAAACAUUUGUUGGAAGGAGGCGUUGCUGAGGUUUGGGUCGA
UUCCAUCCAAGUCCAAGGCCUCCAGUUUACUGAAGUGCCGCAAUUGAUCUAGCUGU
GCAGAUCUGGCAUCGAUCAGAGUGAGAUUGGUUAGACUGGGUAGCUCCAGGAGGAG
CUGCAUGAUGUGGCCCCUGUCCCCACUUUUCGGCGGCGACUCUGUAACGGCGUGCA
UAAAUAUGAUGACUGCCUGUCGCAGUUUGGGGCAGUGCCGUCCCAGGAGCCCCAGA
AAGGGGUAGGUAAACGGCUCAUCCCAGCUACCAUGGGGCACUUCGCACCUGGACAC
UUCGGAACCGCACAGCUGCAGCAGGAACUCCCAGUCCGGAAUGGUUUUCAGAAGCC

GCACAUUGAUUUUUUUGUGGCGCGUUCGCAAAAUGUCCGUCAGUACAACCUGGAAU
AGUGGAUGGGCCCGGCCCAGAUUUAGCUGGUCCUCCAGCACAUUUAUAUAUGAAAA
UAUCUGAUGGUAGCAGUCGUAGUUGAGAUCCAGGAAACCGGGUCCCCCACUCAUAC
UCUGCGGUCUUCCGACCAGGAAGGAGAAGGGGCAGCCUAAUCCCAGUUUGUCCAUA
CCUUCAUAGCAUUUCAGGAGCAGAUUCCCGAAUUCGCCAGAUAUUUUUAUCACGCC
AGCAUCUACUCCAUUUGGAGAUCUCUGCGAAAUCGUUUUAUGUUCGUUUACACUAA
AGAGUUUGAGGACAAAAAAGAUAGUUAUCUGUCCGGUUAUAUAUUCCAAGAUCAAC
CUAACAUCUUGGUAAUUACUUCACAAUACCUGAACUCAAGCACUUUUGAAAUUAAA
ACGAACCGAUUUGUCGGUCCGCGAAACUUUAACAAAAACCCGGAGCCUGUUGAGUU
UUACAUACUCCAGCGUUUCGAUGCCAAGGGUACGAAAGCUACUUGGGAAACCCAGA
GUGCAAUGUCCAGCAAGAUGCGAAACCUCAAGGGACGCGAGGUAGUCAUAGGCAUC
UUCGACUACAAGCCUUUUAUGCUAUUGGAUUAUUUAUGUGAGUUAGGAAAAGCCAC
CAUUAUAUUAUGAUCGUUUUAUGAACACGACGGAUGUAACCAUUGAUGGAACAGAU
AUUCAACUAAUGCUCAUUUUCUGCGAGUUGUACAACUGCACCAUUCAGGUGGACAC
AUCUGAACCAUACGACUGGGGUGACAUCUACUUGAAUGCCUCGGGCUAUGGUCUGG
UUGGAAUGAUCCUCGAUAGACGAAACGAUUACGGAGUAGGGGGCAUGUAUUUGUGG
UACGAGGCGUACGAGUAUAUGGACAUGACUCAUUUUCUGGGACGAUCUGGAGUAAC
CUGUCUGGUGCCCGCUCCGAACCGUUUAAUCAGCUGGACGCUCCUGCUCCGACCAU
UCCAGUUUGUCCUCUGGAUGUGCGUGAUGCUCUGCCUUCUGCUCGAGAGCUUAGCU
CUUGGUAUAACCCGUCGCUGGGAACACUCGUCGGUUGCGGCAGGCAAUUCCUGGAU
CAGUAGUCUGCGCUUCGGUUGCAUCAGUACGCUGAAACUUUUCGUAAAUCAGAGCA
CCAAUUAUGUGACCAGCUCGUAUGCUCUAAGAACCGUCCUAGUGGCCAGCUACAUG
AUCGACAUUAUAUUGACCACUGUGUACAGUGGCGGUCUGGCGGCCAUCCUCACUUU
GCCCACUUUAGAGGAAGCGGCGGACUCUCGCCAACGUCUCUUCGACCAUAAGCUCA
UUUGGACGGGGACUUCACAGGCCUGGAUUACCACCAUUGACGAGCGAUCGGCCGAC
GCCAGUUCUUCUUGGUUUAAUGGAGCAUUACCGAGUUUACGAUGCCAAUUUAAUAU
CCGCCUUCUCGCACACGGAGCAAAUGGGAUUUGUCGUCGAGCGCCUUCAGUUUGGU
CAUUUGGGCAACACCGAGCUAAUAGAAAACGAUGCCCUAAAGCGGCUGAAACUUAU
GGUGGAUGAUAUUUACUUUGCAUUUACGGUGGCGUUUGUGCCGCGACUGUGGCCGC
ACUUAAAUGCCUACAACGACUUCAUUCUGGCUUGGCAUUCCUCGGGCUUUGACAAA
UUCUGGGAAUGGAAGAUCGCCGCCGAAUACAUGAAUGCGCACCGCCAAAAUCGCAU
CGUGGCAUCUGAGAAAACGAACCUAGAUAUAGGACCUGUUAAACUUGGCAUUGAUA
AUUUUAUUGGCCUAAUCCUGCUUUGGUGCUUCGGCAUGAUUUGUAGCCUUCUGACA
UUUCUCGGAGAACUCUGGAGGGGACAGGGGUAG

**Ir93a (2607 bp; Probe set ID PRH241)**

AUGAAUCCUGGCGAAAUGCGGCCUUCGGCUUGCCUUCUGCUCCUGGCUGGACUGCA
GCUCUCUAUCCUGGUACCCACUGAGGCCAAUGACUUUUCGUCCUUCCUGAGCGCCA
AUGCAUCGCUGGCCGUUGUGGUGGAUCACGAGUAUAUGACGGUUCAUGGCGAGAAU
AUAUUGGCUCAUUUCGAGAAAAUCCUGAGCGACGUAAUACGGGAGAAUCUAAGGAA
CGGUGGCAUAAACGUAAAAUAUUUUAGCUGGAAUGCAGUGCGAUUGAAGAAGGAUU
UUUUGGCUGCCAUAACUGUUACGGAUUGCGAGAAUACAUGGAACUUUUACAAGAAC
ACUCAGGAAACUUCAAUUCUACUGAUCGCCAUUACGGAUUCCGACUGUCCCAGGCU
GCCCCUAAAUAGAGCUCUAAUGGUACCCAUCGUUGAGAACGGCGAUGAAUUCCCCC
AACUUAUUCUGGAUGCCAAGGUCCAGCAGAUUCUAAAUUGGAAGACCGCCGUUGUU
UUUGUGGAUCAAACCAUAUUGGAGGAGAACGCACUUCUGGUAAAAUCGAUUGUGCA
CGAAAGUAUAACCAACCACAUCACCCCAAUCUCCCUGAUCCUUUACGAGAUCAACG
ACUCCCUGAGGGGCCAACAGAAGCGAGUUGCUCUGCGCCAAGCUCUGUCUCAAUUC
GCUCCCAAAAAGCACGAGGAGAUGCGCCAGCAGUUCCUGGUCAUAUCUGCCUUUCA
CGAGGACAUCAUCGAAAUAGCCGAGACCCUGAACAUGUUUCACGUGGGCAAUCAGU
GGAUGAUUUUCGUGCUGGACAUGGUGGCUCGGGACUUCGAUGCCGGCACUGUGACC
AUAAACCUGGACGAGGGAGCCAACAUAGCCUUCGCCCUCAACGAAACGGAUCCCAA
CUGCCAGGACUCGCUAAACUGCACGAUCUCGGAAAUUAGUCUCGCUCUGGUCAACG
CUAUUUCCAAAAUUACCGUCGAGGAGGAGUCCAUAUAUGGUGAGAUCUCCGAUGAG
GAAUGGGAGGCCAUCCGCUUUACCAAGCAGGAAAAGCAGGCCGAGAUUCUGGAGUA
CAUGAAGGAAUUCCUGAAGACCAAUGCCAAGUGCUCCAGCUGCGCGAGAUGGCGCG

UGGAGACGGCCAUUACCUGGGGCAAAAGCCAGGAGAAUCGCAAGUUUCGCUCAACU
CCCCAACGCGACGCUAAGAACCGAAAUUUUGAGUUCAUCAACAUUGGCUAUUGGAC
ACCCGUGCUGGGAUUCGUCUGCCAGGAGCUCGCCUUUCCGCACAUCGAGCACCACU
UCCGCAACAUAACCAUGGACAUUCUGACCGUGCACAAUCCACCCUGGCAAAUCCUU
ACCAAGAACAGCAAUGGGGUCAUCGUGGAGCACAAGGGCAUUGUUAUGGAGAUCGU
CAAGGAGCUGAGUCGCGCCCUAAACUUCAGCUACUACCUUCACGAAGCCUCCGCAU
GGAAGGAAGAAGAUUCACUCAGCACAUCAGCGGGCGGAAAUGAAAGCGACGAGCUA
GUUGGUUCCAUGACCUUUCGUAUACCCAUCGAGUGGUGGAGAUGGUGCAGGGCAA
UCAGUUUUUCAUCGCUGCCGUGGCAGCCACCGUUGAGGAUCCCGACCAAAAGCCCU
UCAAUUAUACCCAGCCCAUCAGUGUGCAGAAGUACUCCUUCAUCACCCGCAAGCCG
GAUGAGGUGUCCCGCAUUUACUUGUUCACGGCACCCUUCACCGUGGAGACUUGGUU
CUGCCUAAUGGGCAUCAUUCUGCUGACUGCUCCCACGCUGUACGCCAUUAAUCGCC
UAGCUCCUCUGAAGGAGAUGCGAAUCGUGGGCCUGUCCACAGUUAAGAGCUGUUUU
UGGUAUAUAUUCGGGGCUUUGUUACAACAGGGAGGCAUGUACUUGCCCACAGCAGA
CAGUGGGCGCCUAGUGGUCGGCUUUUGGUGGAUCGUGGUUAUCGUGCUGGUGACCA
CCUAUUGCGGCAACCUUGUGGCCUUCCUCACGUUCCCCAAAUUUCAACCGGGCGUG
GACUAUUUGAAUCAACUAGAGGACCACAAGGACAUUGUACAGUAUGGAUUGCGAAA
CGGCACCUUCUUCGAGCGGUACGUUCAGUCGACAACGCGGGAGGACUUCAAACACU
ACCUGGAACGGGCGAAAAUCUACGGCAGCGCCCAAGAGGAGGACAUCGAGGCGGUG
AAGCGUGGCGAGCGCAUCAACAUCGAUUGGCGGAUCAAUCUGCAGUUGAUUGUUCA
GCGGCACUUCGAGCGGGAGAAGGAGUGCCACUUUGCUUUGGGCAGGGAGAGCUUCG
UGGACGAGCAGAUUGCCAUGAUUGUGCCGGCCCAGAGUGCGUAUCUGCACCUGGUA
AACCGCCACAUCAAGAGCAUGUUCCGGAUGGGCUUCAUCGAGCGCUGGCACCAGAU
GAACUUACCCAGCGCGGGCAAGUGCAACGGGAAGAGCGCCCAGCGCCAGGUUACCA
ACCACAAGGUGAACAUGGACGACAUGCAAGGGUGCUUUCUGGUCCUGCUCUUGGGC
UUCACGUUGGCUCUUUUAAUAGUGUGCGGCGAGUUCUGGUAUCGUCGCUUUCGGGC
CAGUCGAAAACGGCGUCAGUUCACCAACUGA.

### Orco (1461 bp; Probe set ID PRD954) – positive control

AUGACAACCUCGAUGCAGCCGAGCAAGUACACGGGCCUGGUCGCCGACCUGAUGCC
CAACAUCCGGGCGAUGAAGUACUCCGGCCUGUUCAUGCACAACUUCACGGGCGGCA
GUGCCUUCAUGAAGAAGGUGUACUCCUCCGUGCACCUGGUGUUCCUCCUCAUGCAG
UUCACCUUCAUCCUGGUCAACAUGGCCCUGAACGCCGAGGAGGUCAACGAGCUGUC
GGGCAACACGAUCACGACCCUCUUCUUCACCCACUGCAUCACGAAGUUUAUCUACC
UGGCUGUUAACCAGAAGAAUUUCUACAGAACAUUGAAUAUAUGGAACCAGGUGAAC
ACGCAUCCCUUGUUCGCCGAGUCGGAUGCUCGUUACCAUUCGAUCGCACUGGCGAA
GAUGAGGAAGCUGUUCUUUCUGGUGAUGCUGACCACAGUCGCCUCGGCCACCGCCU
GGACCACGAUCACCUUCUUUGGCGACAGCGUAAAAAUGGUGGUGGACCAUGAGACG
AACUCCAGCAUCCCGGUGGAGAUACCCCGGCUGCCGAUUAAGUCCUUCUACCCGUG
GAACGCCAGCCACGGCAUGUUCUACAUGAUCAGCUUUGCCUUUCAGAUCUACUACG
UGCUCUUCUCGAUGAUCCACUCCAAUCUAUGCGACGUGAUGUUCUGCUCUUGGCUG
AUAUUCGCCUGCGAGCAGCUGCAGCACUUGAAGGGCAUCAUGAAGCCGCUGAUGGA
GCUGUCCGCCUCGCUGGACACCUACAGGCCCAACUCGGCGGCCCUCUUCAGGUCCC
UGUCGGCCAACUCCAAGUCGGAGCUAAUUCAUAAUGAAGAAAAGGAUCCCGGCACC
GACAUGGACAUGUCGGGCAUCUACAGCUCGAAAGCGGAUUGGGGCGCUCAGUUUCG
AGCACCCUCGACACUGCAGUCCUUUGGCGGGAACGGGGGCGGAGGCAACGGGUUGG
UGAACGGCGCUAAUCCCAACGGGCUGACCAAAAAGCAGGAGAUGAUGGUGCGCAGU
GCCAUCAAGUACUGGGUCGAGCGGCACAAGCACGUGGUGCGACUGGUGGCUGCCAU
CGGCGAUACUUACGGAGCCGCCCUCCUCCUCCACAUGCUGACCUCGACCAUCAAGC
UGACCCUGCUGGCAUACCAGGCCACCAAAAUCAACGGAGUGAAUGUCUACGCCUUC
ACAGUCGUCGGAUACCUAGGAUACGCGCUGGCCCAGGUGUUCCACUUUUGCAUCUU
UGGCAAUCGUCUGAUUGAAGAGAGUUCAUCCGUCAUGGAGGCCGCCUACUCGUGCC
ACUGGUACGAUGGCUCCGAGGAGGCCAAGACCUUCGUCCAGAUCGUGUGCCAGCAG
UGCCAGAAGGCGAUGAGCAUAUCGGGAGCGAAAUUCUUCACCGUCUCCCUGGAUUU
GUUUGCUUCGGUUCUGGGUGCCGUCGUCACCUACUUUAUGGUGCUGGUGCAGCUCA
AGUAA.

## *Drosophila sechellia* experiments (using *D. melanogaster* sequence)

**Orco** – see above
**Ir25a** (2396 bp; Probe set ID PRL623)
CAAACGCCCCAUCUGAUCCUGGACACCACCAAAUCGGGCAUAGCCUCGGAAACGGU
GAAGAGCUUCACCCAGGCUCUGGGUCUGCCCACCAUUAGUGCCUCCUAUGGCCAGC
AGGGCGACUUGAGGCAGUGGCGCGACUUGGAUGAGGCGAAGCAGAAGUAUUUGCUG
CAGGUGAUGCCGCCGGCGGAUAUUAUUCCCGAGGCCAUUCGAAGUAUAGUGAUUCA
CAUGAACAUCACGAAUGCUGCCAUUCUGUACGAUGAUUCCUUUGUCAUGGACCACA
AGUACAAGUCCCUGCUGCAGAAUAUACAAACCCGUCAUGUGAUCACCGCCAUAGCC
AAGGAUGGUAAGCGGGAGCGCGAGGAGCAAAUCGAAAAGCUGAGGAACUUGGACAU
CAAUAACUUCUUUAUUCUGGGCACCCUGCAAUCGAUCCGCAUGGUCCUGGAGUCGG
UGAAGCCAGCGUAUUUCGAGCGCAACUUCGCCUGGCACGCCAUCACUCAGAACGAA
GGAGAGAUUAGCAGUCAGCGGGACAAUGCGACCAUUAUGUUUAUGAAACCCAUGGC
GUAUACGCAAUAUCGAGAUCGCUUGGGAUUACUGCGAACCACUUACAAUCUGAACG
AGGAGCCGCAGUUGUCAUCCGCGUUUUACUUCGAUCUGGCACUUAGGAGUUUCCUU
ACCAUCAAAGAAAUGUUACAAUCGGGCGCCUGGCCAAAGGAUAUGGAGUAUCUGAA
UUGUGACGAUUUCCAAGGUGGCAACACACCCCAAAGGAACUUGGAUCUUCGAGAUU
ACUUCACCAAGAUUACCGAACCGACUUCGUAUGGAACCUUUGAUCUCGUCACGCAA
UCCACUCAGCCAUUCAAUGGGCAUAGCUUCAUGAAAUUCGAAAUGGAUAUAAAUGU
GCUGCAGAUUCGUGGUGGCAGUUCCGUGAACAGCAAGUCCAUUGGCAAAUGGAUAU
CGGGUCUGAACUCGGAGCUCAUCGUCAAAGACGAGGAGCAGAUGAAGAAUCUCACU
GCAGACACUGUUUAUCGAAUCUUUACUGUAGUGCAAGCUCCUUUCAUAAUGCGCGA
UGAAACGGCUCCGAAAGGAUACAAAGGAUACUGCAUUGAUCUGAUCAACGAGAUAG
CCGCAAUUGUCCACUUCGAUUACACCAUCCAGGAGGUGGAGGACGGCAAGUUUGGC
AACAUGGACGAGAAUGGGCAAUGGAAUGGCAUUGUGAAGAAGCUGAUGGACAAACA
GGCGGACAUUGGCCUUGGCAGCAUGUCGGUGAUGGCCGAACGGGAGAUAGUCAUUG
ACUUCACCGUUCCGUACUACGAUCUGGUCGGGAUUACGAUCAUGAUGCAGCGACCC
AGUUCGCCAAGCUCGCUGUUCAAGUUCCUUACCGUGCUGGAAACGAACGUGUGGCU
UUGCAUCCUGGCUGCCUACUUCUUUUACCAGCUUUCUCAUGUGGAUCUUCGAUCGCU
GGAGUCCCUAUAGCUAUCAGAACAAUCGAGAGAAGUACAAGGACGACGAGGAGAAG
CGCGAGUUCAAUCUGAAGGAGUGCCUCUGGUUCUGCAUGACUUCAUUGACACCUCA
AGGCGGUGGCGAGGCUCCAAAGAAUCUGUCUGGCCGUUUAGUGGCCGCCACCUGGU
GGCUAUUCGGUUUUAUCAUUAUUGCUUCGUACACGGCCAAUUUGGCUGCCUUCUUG
ACCGUAUCACGUUUGGAUACGCCCGUUGAAAGCUUGGAUGACCUGGCGAAGCAGUA
CAAGAUCCUAUACGCUCCAUUGAAUGGCUCAUCUGCGAUGACAUAUUUCGAGCGUA
UGUCCAACAUAGAGCAGAUGUUUUACGAGAUUUGGAAGGAUCUGUCGCUGAACGAC
UCCCUGACCGCCGUGGAGCGCUCCAAGCUGGCUGUUUGGGAUUAUCCAGUGAGCGA
CAAGUAUACCAAGAUGUGGCAGGCCAUGCAGGAGGCGAAGCUACCGGCCACCCUCG
ACGAAGCGGUGGCCCGGGUUAGAAAUUCGACAGCUGCCACGGGUUUUGCCUUUCUG
GGCGAUGCCACCGAUAUACGCUACCUGCAGUUGACCAACUGUGAUUUGCAGGUGGU
GGGCGAGGAGUUCUCCCGGAAACCCUAUGCCAUAGCUGUUCAGCAGGGAUCGCAUC
UCAAGGAUCAGUUUAAUAAUGCAAUCCUGACCCUGCUCAACAAACGACAGCUGGAG
AAGCUCAAGGAGAAGUGGUGGAAGAACGACGAAGCUCUGGCCAAGUGCGAUAAGCC
GGAGGAUCAAUCGGAUGGCAUCUCGAUCCAGAACAUUGGCGGCGUCUUCAUUGUCA
UAUUCGUGGGCAUUGGAAUGGCCUGCAUCACGCUGGUCUUUGAGUACUGGUGGUAC
AGGUACCGCAAGAAUCCGCGGAUCAUCGAUGUGGCCGAAGCCAA.

The following genomic sequences from VectorBase were used by Molecular Instruments, Inc (Los Angeles, California) to produce custom probe sets targeting coding sequences:

**Anopheles coluzzii** experiments
**Orco** (10,821 bp; Probe set ID PRL382) – AGAP002560
CGTCGAGCGGAAAGGACGTATCAAGTCGATTCGTCTATCAGTGTCGGAACGATAGTGATA
GAAATCTACAGGCGCCAATCATAAACCGTTTACTGCTCGCGATAGGACACGCTTTGCTAA
CGTCTTGTGCATCGCGAAAACTAGTGATTGGAGTGTGGTTTTTGTAGCTGTTTGTCGTTCT
GTGCTCCACGTTGACTCTGTGTGTGTGTGCGCTGAATTCAACAAGACTTTTCTGCAACAG

CATCATTTGCAAAGAATAACCGGCGCGACTTACGCGGTCTGACTTGCTGGTGCGCTGCTT
TGTACGGCAAACGGCTACACAAGCGAATCGAATTATTTTCCTATCACGCTGCGCTTACCA
GCGCCTGCTGGTAGGCAAAGAATGTGCAAAGTTTCATTTGGCTTGGTTCGTCTGCTTTGC
TGTGAACGTGTGCACGGTTGCATCGCTAAGGTTTCGGTGTGAGCCGAGAAGTTGCAGATC
GAAACCCCTTTGTGTGTGTGTGTGCCGTGGGAAGCATTGTGTTTAGTGAGAAGTGAAAAG
AAAAGTGCTGAAAAGTAAGTATTCGATGTAGCTTCAACCTTTACACTTGACAAGGCGGAG
ATCTCAACCATTTTTCTGGTTCGAACGTTAAGCGGAACATTTGATCCTTTCAAAACAGTG
TGACTGTGTGTGTGTGTATGTACAATCGCCCCCCCATGTATTCCAAATATTTGCTTCTCGCAAAT
CAGGTCACAAAGCTGTGCTCAATGTTGGCACAATCAAAGCTTTCCCCATTCGTCTGGGCC
AAGCCTAGCTCAATTGTTTGTACGAGAGATTTTACCGTTTGGCTCCAAGCATTTGCAATT
GTCCTTTCCTGTACGCTTGCCGCCGCTTCACTGCGCCCTTCTAAGTACACAGAGCACACA
CAGCTTAACTCTTCCCTGTAGCTGTACGGCTTGGTGTCTGTTACCTAAAAATCGATAAATCAAT
TAGTCTCTTGCCGACCACACCAAACGACCACTACAAACGAGCCCGACTCGACCCGTTCAC
TTCCCGTTCGCAGATGCAAGTCCAGCCGACCAAGTACGTCGGCCTCGTTGCCGACCTGAT
GCCGAACATTCGGCTGATGCAGGCCAGCGGTCACTTTCTGTTCCGCTACGTCACCGGCCC
GATACTGATCCGCAAGGTGTACTCCTGGTGGACGCTCGCCATGGTGCTGATCCAGTTCTT
CGCCATCCTCGGCAACCTGGCGACGAACGCGGACGACGTGAACGAGCTGACCGCCA
ACACGATCACGACCCTGTTCTTCACGCACTCGGTCACCAAGTTCATCTACTTTGCGGTCA
ACTCGGAGAACTTCTACCGGACGCTCGCCATCTGGAACCAGACCAACACGCACCCGCTGT
TTGCCGAATCGGACGCCCGGTACCATTCGATTGCGCTCGCCAAGATGCGGAAGCTGCTGG
TGCTGGTGATGGCCACCACCGTCCTGTCGGTTGTCGGTATGTGTGTATGTGTGTGGCCGT
TTGGGAAAGTGTCTTTGCGGCAGAACCCCAATCTACTGTTACGCTTGACTGGGTTTTTGT
TTTTTTCTCGGTGGAGGGACGGGATAAAATATCTGAAAGAATAATTGAGTCAACCCACAG
GGGGATGCAAGACATCGCAGGCAGAGAGTTTGGGTTTGATTTATCACCGCACACCGAATA
TCTTCACGGTTCATAAGCTTCACCGCGGTGAAAAGGGAACTCCCCATTTCCCTGTTTTCT
TTTTTTTCTTCCTCTCGATAAATTACTCATCGCTTTTCGTTTTTTTTTTTTTTTGTTGTTGCTTCTTTCT
TCTTTCATCCCTACTAGCCTGGGTTACGATAACATTTTTCGGCGAGAGCGTCAAGACTGT
GCTCGATAAGGCAACCAACGAGACGTACACGGTGGATATACCCCGGCTGCCCATCAAGTC
CTGGTATCCGTGGAATGCAATGAGCGGACCGGCGTACATTTTCTCTTTCATCTACCAGGT
ACGTTGGCGGAATGTCCTGCGCGTCACAGTTGGCAGTCAGTGAGCGGCAACACGGC
GAAAAAATGGGACTAAAACCGGTCTTCACAGAGCCAACACATTCCTACAGCAATTGCATA
CCTTCGGGCGGTCGGGACTGGGCAATGCAGCTACAACATCCTCGCCTAAAGTTATGCAAT
TCGAGCGACAAATGTTGCCGTGTTAGGGCTTTTTGTGATAATAGTCGTTTTTTTGTCCTCTCGC
TTATCAAACTCTATCAACGGAGGAAATCCATTTTCGCTACAATGCCTACAGCTCAAGTTT
CAAGGTCAATCGAGCGGGTGGGGATCAACTTTTTTATTCATTTTGCTAACGCCCCATCAA
CAAATTCTATGTTCTCAATGGCAAAGATTACTGCCCGCACCAATCGCCCAACGAAACGGC
AAAAGAAAAGCGACGATTATGAAGATGTCCAAACCATTGCCCGCCCGACGCTTTATCTGA
TGATTTGCGGGATGGCTTTTACTTGTCTGCTACTTTCAGGCACAAAAGGAAATGAAACCA
GCGCAGGCTCGTTTGCCGGCTTGCGGAGGTTCTTCAGGCACTGAGGCTGAGTACTTAAAT
CGAACGATTTTTACGATTCTGGATCCAGTTTTATGATGTGGCCTGCATTACAGTGGCAATTATA
CCCTGATGTTCATTTCATTGCATTTTGTAAGTTTGTGCTGGTAACGCCCGTAACGATTAATTCT
TTTCAAAGAGATTCTTTCAAAGAGATTCAAAATGTGTATAACAAATGCTAACGAATGGAC
CGTACTTGGAGGGGTTGCGGAAAGTAACGTTTTAAAATATTCATCACAATCCTCTGCAAAC
TTGTGCTTAATTAATTGGTGCACAATAAGTTTAAACTGTGGCGGCAGATGTGTCGCTGTC
CGCTTCCTTCCTTCCCAGCAAGCTCGTGCGAAATAATTTATTCCATCATTTTAATACAGCCGTT
TGTGCATTTTAATTAGCAAAGCAATATAAAAAGCAGCTAACCATCCCCATTAAAACAAAG
TGCTTCCGGGCCCAATTGTTATGGCGGTGGAAGTAATGGTTTTACCAGTGGAAGTGTCCT
TTCCCATCGTGGGTACTTCGCGATATTCTTGTCTTATAACAAGTGCATACAGAAAAAAGG
AACAAATCCTCCTTGCTATGGTCTAAGGGCCAGCTTCGGTACCGCTTCCGCTTCGGGATG
TCATAAAGTTTGATGGGTGTTTTTAACATTACTTCCGCTCTTAACCACCTAATGGACTTTTCAT
GCTTGAGCTAAAGCTAAACCAGCCACCAGCGGTACGCACCGAGCCACGGTTGATTTCGGC
GGCGGCCTCATCCCCAGTTTTGCGCCACCAATATTGCCTTCATTAATCTGTACCCTCGGA
GCGTTAGGGCCCGCGGACGAGTCCTCGTTGTAATGCACCGCCATGCCACGGGACGGGATA
ATCCGTTGGGACGGCGCGAAAGCGACTATCGCGGACGGATTGGTTCGACCGTGCTACAAC
ACATTTTATGCTTCACAGATTTACTTCCTGCTGTTTTCGATGGTCCAGAGCAACCTCGCG

GATGTCATGTTCTGCTCCTGGTTGCTGCTAGCCTGCGAGCAGCTGCAACACTTGAAGGTA
GGTACGGTAGCAAACGTGGTTGTCTTTACATCCGCGTGCAGCATTATCCTTATCGACGTG
TAGTGTTAACGGTAAAAGAGGAAGCGATAAAAAAGCAACATTCTCTCACACCCTCGATCT
CTCTTTATTTTCTCTCTCTCTCTCTCTCTCTCTCTCTCTCTCTCTCTCTCTCTCTCTCTC
TCCATCTCCCTCGGGCAGGGTATTATGCGATCGTTGATGGAGCTTTCGGCCTCGCTGGAC
ACCTACCGGCCCAACTCTTCGCAACTGTTCCGAGCAATTTCAGCCGGTTCCAAATCGGAG
CTGATCATCAACGAAGGTATGTGAAACGTGTGCTCGTGGCAGACGGACTCAAAGAGAGCA
TAACACAATCCCCTGGTAGTTCATTTCAATGACCTTAACACTCGGCAAGCTAAGCGAGAC
AGTGGGGACAGTGAGAAAGAGAGAACAAGAAAAAAAACCATCATCCGTACGACATCATCG
CTACGTACCGGTATTTCAGGATGAGGAAATAAAACGCTAGGGGAATGAAAGTGCGACAGA
ATGATAAAACAATCCCCACCCAGGCCCCCAGCCTGGACGAACGGATGTAGTGTGCGAAGC
GAGCAAAAAAAGTCAAATAAATTGAAGTTTAAAAATAGATTTTCCCCGTCCATCCGTGGT
GGAGCGTAAAGCCCGGCGGACAACTTCGAGCACGGCGACCGTGCACAGTACTGTGC
CACAGTTGTAGGGACGGATAAGCTCCGTTCCTTTTTTATCCTTTTTTTTTGGAGATTTGTTTGC
GTTCGCATCGTTAGACGAGCTTAGTGCCGTGTTGCTCTAATTGCTATTTATTATAAAGCGCTTC
CAAATAGAAGATCGGTTCTCTCCATTTAATCTATCGCGCCTGTACGCCTGAAACTATGCA
CTGTGCTGTGAAACCGTCAAGCTCGAGCACGACGAATGGCCCACCGTACCACGCCCGTGG
TGCCCAAAGCGCAACGCGAATTGCATGTTAACAAACCTTTGCCTACCATCCAATCCGTGT
GAAATTGCCCGCTCTCTTTCTCTCTTTTGCGCTTTCGGTGTATCGAACGGTTTTGTCCCTTTTT
TTTACTTTGCTCTTGATCTCTTGCTGTGCTCACTTTCATCTCATGTTTTGCCTGACGGTGGTGG
GTTTTCGAAAAAGAGCGATTTCTTCTGCGTGTGTGTGTGGTTTTTTTAAATAACCGCTC
CAGGTCGTGTTGAACGCTGCAGGACCGATCGGAGCTAGTTTATTATCAGCTTTAGTGTTT
ATCCCACCCATGCCCCACATCACGTCTGTGGAGAGTGGGGGAAGCTTAAGTCCAATGTAA
TTTACCGTGTTTCTGTCGTTCGTCACCTTCTTCGTCGATGGAGATTGGTGCGGTTGGCAC
GATAAAAGCCCACTGCACGTTACGGACCGAGGGAAAGGTCTTTTTGTAGGCCTAGCAACG
GTCCTCATTCACCGCATGGGGGTGTAGCTCAGATGGTAGAGCGCTCGCTTAGCATGTGAG
AGGTACCGGGATCGATACCCGGCATCTCCAACCCACACAAAACGTTTTTTAAGAAGATTT
TTAGGGAAGATATTAACGCGGGTACACTGTGCTCCTCTAAGTTGGAAGAGTAGATGAGAT
GATGACAAGGGAGAAGGAACATGTGTACGTGTTTGATAGCAAACACACAAACAACAATAT
CATCTCTGATAATAATCTGATGTGTGATGTGTGTGTATTGTTGTTATGCTGCCTTTGCCATCTT
GTCCCTCTCTCTCCTGTTCAACTCCTAAAAGAATTGTTTGGAGTCCTCTCAGTTCCTCGT
AAAGATCCTTTCGAGATTCTTCTTTCCTTTTTATTATTTATTCCACGAGCCTCTGACATAAGTA
GCCTTCCGCTTATTTCCTTCTCCTTGCACTTGTCAGTTCCGTGTAGAGCGTCATTTTGAGGTTT
ACACATTTCCCACCGACGCCTGATTGTTACATTGTCATCTACATTGCTTTCCGTTTACCGTTCC
GCCCTTTTTTTTTAACGCTACCACAGAAAAGGATCCGGACGTTAAGGACTTTGATCTGAG
CGGCATCTACAGCTCGAAGGCGGACTGGGGCGCCCAGTTCCGTGCGCCGTCGACGC
TGCAAACGTTCGACGAGAATGGCAGGAACGGAAATCCGAACGGGCTTACCCGGAAG
CAGGAAATGATGGTGCGCAGCGCCATCAAGTACTGGGTCGAGCGGCACAAGCACGTTGTA
CGGTAGGTATGGTAATTTCTAAGGTGTGGTGTAAAGCCTCCAGGTTCCATGAAAAAGGGA
TACTTTACCACAGTAAGAGTTTGTTTTGCTGGACTTACATTCTTTGGAGCATTGTTTGGTGTTG
TGCTGAAACCGGTTGCAATATCGTTTTGCGAAGAAATTATGTGTAAAGCGTATTACAATCTCAT
TCCTCTGTTAATCTGTACCAATTGTGTCAGCCCCGACCGAAAGCAGGCCTAATTCGTACC
AGAAAAACCACAAGCTGTTTGTAAGCATCGATACGCCCGAAGCTTTCAATCCAGCCAAGG
CGCCACCTACTATTGACGTGACTTTTTGCACGTTCACACTCTCCCTCTCCCATTCTTTCTATAA
CCAATCGTCGCTCAGCCAGCATCGCCCGGAGTGAAGTTTTTTATTTGAACGATATCACCCG
TATCGATTTTCCACTAAACATGCTTAAATCGTTTCACAAAGCTCCCCCAAAATCCCATTTCACC
AATCCACCAATTTGAAGTCCGTCGTCCTTTGTGTCCTTGTGTTTGTGTGTTTGTGTGAGC
TGGAGACATGGGGGAGTGAGTAACCGAACAACCTCTTGCCGCTGCTTCACGATATCGAAC
AGCACCAAGATAAGCATCCCTTTTTTCCCTAGCCGATGTCTCCGATATCTCGATTCCGCTT
CCAGCGAGGCAAAGAAAAAGGCGAACTGGCTGACCTCACCCGGGGCGAGGAAAAAG
CGTAGGGATTACGTCGAGCAGCACGAGTTGTGATTTCTTCTTCTTCTGGTTCCATAAATC
GCTGACGGTTTCCATTACCGCCTGCGGAGTGCACACACGTGAAGGGAAAGCGAAACGTT
TAGATTCCAGCAGCAACGGCAGCACCAGAAGCAGCAGCAGCGCGGCAAATTGAATCATCC
TGACGCGATGAGTTGTCTGGGTTTTCGGGTCGGTGGCTTACAGCACCACACCATCTGCTG
CAGCTAATACAGCTGTAAATTTCGTTAGACATAGACTTGATTTTACAATATTACACACACACTT

```
ACACACACAGCTATAGATTTGTCGCTTGGCGTATGGCTCTGTACGGCGTGCCGTACATGC
CGCGAGCCGTGTTGCTGCTGGTTGCGATACGGATCACGTCCGATTCGATTCAGCCTGCGT
GTTTTTGGTGAAGATCCTTATCGGTGACCCACTTTCAGTGTGTCGAGAGCGAGGGTCACT
ATGGCGCCTGTCAGTTGGAAAGCTAGGCTCGATTCAAAGGGCCATTGTGCCAGTGTTCTT
TTTAAGATAGCGATAAGCTTTTGATCGAAATAGTAAATCAAACATTGTTTCTTTTTTCCTATTC
CAAACTGTTGCCAACCTCATTATTACGTTTTTGCAGCGGGTGTATAGTAAATTGCATACTTTAA
GGCGTGATTTTCAAATGTAGCGTTCCGTATGCAGAAACGCCATGGATTATGCAATTTAAA
CAATGCTGCTTCCTTAACATTCAAATAACGGCTTATTAAGGAACTTTTTGTGCAATTTGTTTTT
AACAGCAAATAGTTAGCTCAGAACGATCACATTTAGTATCGCTTCAACAAAGAACTCTTT
TAAACACACAATTTGTAATGCCATTCCCTCGAGAAAGTTTCTTGTCAGTCCTCCTCTGCA
TCACAGCAACAACCAAACCTGCTCATGTTTCCTGCTCGTTTCCTAGCTGTTTTGAACGTT
ATTTCCGATTCCTGTGCTTGCCCGCTTTTCTTACAATCAACCACAATGGTTCAGATTTCGCTCT
TATTTTATTGACCCACTGCTTTCGTGCTGAAGCCCGTGGAAACAATGCGCCAAGCTCAGC
ATCCAGCCATGCATGTAAAATGAGCCACGCGACAGATTTTAGACATCGCTTTCGCTCTGC
ACCGGAGGTGGTTTTATTCTTGTTTCCGATTCCCACGTCCATTCGTCCTGGGTCCGTCCG
CCGGGCCCGAAACCGTAAGCCGTGCGGGGAATTACGCAATCGAAACGAGCCAGAAA
ATGAGCACGCCAAATGCAAAGAAAATCCCCTTTTGAGTGGTGCTCCTGCCACCACTCATC
TCCCCAACTGGTGGGTGAAAAACCTTGTGCGCCCCTTCTCTTTCCAGAAAAAAAACGCCT
CGCTCGCACAAAACATGCTCGCCCGGTGAAGCTGCGTATGTCGCAGAAGCTCAAACCAA
CGCCGCCAGCAAGCATCAACAATTTCTATTCAAACACCCAACGCAGCGCCCAAACCGGGT
GCACTGTACTCAGTAGCGAAGATGCTCAGATTGTCCCGTGCGCTGCTTTCGATGCCCGTT
TCGGAGCGGGAAGCCATCGCTTGCCAACGTTGGCGATGTCTTTTAGCCGTGGATTTGAAT
TTTCTGAATATCACAGGCGGGCGCGGTTTGCCTGCAAGGTTGTTGCTTCCCACACGAGCA
TTGCTTTCCGTACCGCGGTGGGGCGAGTTTTCAACGCAACCTTCTACAAGCAACGCCACA
ACGCCTGGGAGCGATATTTAACAGAAACAAGAACATCCCGAACTTCAGCACATGCCGTGA
TTTGCCTGTTGGAAAAGCTTTTGTGAGCGTGTGAGTTGAACGAGCTCTATTTTCCCAGCG
ATGGGTGGCATTTGTGTGGCATGCTATCGTCAGCTTTTCTTGAATCTTTACCTCTCCATTCGCC
TCCATTAGTACACGCGTATGGAAAATGGGTGCAACGGATCAGAACGGATTTTCCGCGACA
GACTTAATAAAGGGAAAGCAACGCGTTTTTTGCATGTGTAGTGTTTATGAGCTTTATGCCGTTA
CTTTGCAATTAAAAATAGCAAAAAATAACAGTTTTTTTTTGTAAGCGGATTACAAAGAATGTAT
CAGAATATTACGTGAAACATTCATTTCATGCTGTTAACGCTCAAATAGAATAGTTTTGTAACAC
GGATTGCATACCTTGCCGGTATCGGTTACATTTTCGCCTAACAGTATGCAATCTGTTTAGCTTT
GTTGTTTAATGACTGCGTTGGTAGTACAATATTTATTTACACCGCGTAATTTATCTCACAAATT
GCAAAAAAATGTCAATCTGTATCGATTATTCACACAAATCAGATCCCGGAACCAGTGTAG
CCCAATGTGCTCTTTATTGAATTACCACGAACAAATCAACCTGATGCCCGGGTCCGTTGG
CAAACAGCTTGCGCCGAAGCCGCTCAGTGTTTCGTGCACTACCGTGCTGCCATTTTGCTG
CCCTCATCGAACAGATAAACAGAAGGGCAACTCTTGTGAGCATCGCAATGCCCGTCTGAA
GTTCCGTCGAAAATGGGCCTAAATTCAATTTGACGCATTTACCCGCGAACAATTGCGCGA
AGGCTGTCAAGTGTGTTCCACGAACTGCGACAACAAGCACACACACAAACACAAATGTTA
TCGTTTCGGCATGTTTCTCGGTACAAAGCGTGTGGCGCTATGTGGCATGCCGATTCCCAG
ACAGAGTGATCGATAGTAAATGTAGCCTATCCGGTAGCATTCAATTTCCTTTTCTATCCTCGCA
AACAAAGCCCATTCTGGGGAGGCGTGGTGAAGCTTTCAAAGGCATTGTGAAACAAATGTC
CTGGTTCGGAGGGATGCTGGGGAAAGCAAACACGGTGCCGCCATCGCTGCTACCGTCAAT
CGATCATGCATGATGTGATTAATATTTGTGTTATTCACCTGCGTATCTATGCGTCCGTCGTGTC
GTTCGATTTCCGGAGTCAAGGAAAAAGCGACTCCATTTGGGATTGGTTTTTGCAGCGAAA
AATCAAAACATTCGCACAAAACCGTCCTCCATTTCAAATGCCTACACTTGTCACTGTATATCTC
TCTTTCTCTCGTTTTGCCACGTTGCAGTCTCGTTTCAGCAATCGGAGATACGTACGGTCC
TGCCCTGCTGCTACACATGCTGACCTCCACCATCAAGCTGACGCTGCTCGCCTACCAGGC
AACGAAATCGACGGTGTCAACGTGTACGGATTGACCGTAATCGGATATTTGTGCTACGC
GTTGGCTCAGGTTTTCCTGTTTTTGCATCTTTGGCAATCGGCTCATCGAGGAGGTACGTGC
GCTCGGCGTGTTGCCGTGGGAAAGCATTCTCCCTGCCCCATATCGCTTCATTCTCCCAGA
TCACACATTTGCATCACAAAGCCAGCACACTTTTGCTTCGCCGCTGCCATCTCGGCTTCT
GAATGTTTTCACTCTCCCATACCTCTCCCGTGCAGAGCTCATCCGTGATGGAGGCGGCCT
ATTCCTGCCACTGGTACGACGGGTCCGAGGAGGCAAAAACCTTCGTCCAGATCGTTTGTC
AGCAGTGCCAGAAGGCGATGACTATTCCGGAGCCAAGTTTTTCACCGTTTCGCTCGATC
```

TGTTTGCTTCGGTAAGTGTAGCCTGGTGGCTGGCACAGAACAGGCTGGCAAAACAGGGAC
TTTGGCTCTAGCCTGATGGGTGGTATATGTGTGTCTATTTTTTGCTACCATTCTCGCATCCCTT
CCTTTCCAGGTTCTTGGAGCCGTTGTCACCTACTTCATGGTGCTGGTGCAGCTGAAGTAA
ACAGCCGTGGCCCGGAAGGATGTGTTTTTTTTCGCTCGTTCGGTTGTTTGTTTGTGCACA
CTTTCTCTTGGACATTTTCTCTACTGCAAAGGTTTAACAAACAGCAACAACAAATAATCC
CAAGTTTTCTTTTACAGATCTTTGCAAAATGATTAGATTTTAATAGATTAACAGTGCTTGATTA
TCTGTCCTGTAGCAACCGGGGCTGAAGAACGTTGATTTGGTAAAAGTACAAAAGGGACGT
TGGAAATTGAAACAACAGAAGAGTGATATTTATGCAAAGCTCACCAAGGGAAATCTATGT
ATGTGTGATTTGCGCTCATCAAGCACTGTATGTGCCTTTCAACTAGTGCAGCAATAAAGA
GTACAAATGTTTCTTAG.

***Ir25a*** (4005 bp; Probe set ID PRK149) – AGAP010272

CCGTCGCTTATAGACCGATTCCGCTATACTGTTCGGCTTGGGTATATGTGTGTATTACAAAAAT
ATCATGGATCCTAAGAACGGAAGGCGTTGGCTGGTTTTAATCCCTATTCAGCTTGCATCA
TATGCAATTATAGCGATCATGGGACAAACTACCCAAAATATTAATATATGTAAGCATTTTCATT
AATTAACTCCTTTTGCACTGGAAGTAGAGATTGTTGTGAACATTAGAAAAAGCTCAAATT
ATTTTAAAACGTAAATTTCACTTTATTACCAGTGTTTGTCAATGAGGTCGATAACAATTTGGCT
AATGTTGCTGTTGAAGTCGCACTGAACTATGTTAAAAAGAATCCTCAACTGGGGCTGTCC
GTTGATATGATGTACGTAGAAGGGAACCGCACCGACTCCAAGGATCTGCTACAAGCGCGT
AAGAACTATGAACACTTATGTCGTTGTTGTTATCTCTCTGGTGTTACTGCTTAAGGTTACAAA
CGTACTATTTAATTGTGTGTGATTTTTACCTTTTCAATGTCTAATTTGGTAAATGAGAAAGTGT
GCTCGAAGTATGGCCAGTCTTTGAGCGAAAATCGACCGCCACACTTACTACTGGATACCA
CGCTGACTGGAGTATCATCGGAAACGGTGAAATCGTTCAGTCTGGCACTTGGCATACCAA
CCGTATCGGCTTCGTTTGGACAGGAAGGAGATTTACGCCAGTGGAGAGACTTGACGCCAA
CCAAACGCGGCTATTTGCTTCAGGTAAATCATTATGTCTTCTGCAGGGTGCGGTTGCAAT
GGTAGTTTTCAGTGCCTTGCTTAGGCTGCAATCAAAATCGTTATGAAATGAAGCCAGAAC
GTTATGATTCAACATTCTTTATCCATGAAGGTTATGCCCCCGGCTGATATGATTCCGCAAGTGA
TTCGATCTATCATCATTTACATGAACATAACGAATGCTGCAATTCTGTACGACAACACATTCGT
AATGGATCACAAATATAAAGCTTTACTACAAAATATACCCACGCGCCACGTCATCACCAC
CATTGCTGACGATCGCGATAGAGCCAGCCAGATTGAAAAGCTTCGCAATTTGGACATAAA
CAATTTCTTTATACTAGGTTCTTTAGCATCGATCAAGCAAGTATTGGGTAAGTGGGTTAGTAGT
TGTCGTCGGAGGTGCGCGTTTTAATTAACGCTATTCTTGTTTTGGTCACTTGTTTTCACAGAAT
CGGCTAAAAATGAGTATTTTGAACGTAACTTTGCCTGGCATGTAATAACGCAAGAGCAAA
AGGATCTAACGTGCAATGTTGAAAATGCCACTATCATGTTTCTTCGTCCGATGTCTGATAGTTC
AAGCAAAGATCGATTGGGCAGTATACGCACAACATACAATCTTAAGCAGGAGCCTCAAAT
TACCGGATTTTTCTACTTCGACCTGACACTGCGCGCGTTAATTGCAATTAAGTAATATATCAAG
CTTTATATGGTTGGCTAGCATTAGATGCTGAACTGAGTTGTTTTTTTAAATTATTTTCTTTCAT
TTGCTAGAAATATTTTGCAGTCCGGATCCTGGCCATCAAACATGAAATACATCACGTGTG
AGGATTACGACGGGACAAACACACCAAATCACACAATCGACCTTAAAACGGCTTTCATTG
AGGTGACCGAGCCAACCACTTTCGGACCGTTTGAAATACCAAAAGGCGGAAAAATGCAAT
TCAACGGTAACACTTACATGAAGTTTGATATGGACATTAACGCCGTTTCCATCCGTAGCG
GCGCATCCGTTAATACGCGCAGTCTCGGAACATGGGAAGCGAGCTTAAATGCACCGATAA
ATGTAGCAAATGAGGCGGAAATAAAAAATCTTACTGCCGATGTTGTTTATCGTGTCTATA
CGGTTGTGGTAAGTGCGAGTGGATAAGATAATATATTGTAAGCGACAACAATGTTAACAA
GCGTAAACCCCGTTTATTTCAGCAAGCGCCATTCATAATGCGAGACCCAACGGCACCAAA
GGGCTTCAAGGGATACTGCATCGATCTGCTCAACAAAATTGCCGAGATCGTCGAATTTGA
CTATGAAATACGCGAAGTGGAGGACGGAAAGTTTGGCAACATGAATGAAAACGGCGAGTG
GAATGGTATCGTACGGAAGCTGATCGACAAGCAAGCAGATATCGGACTCGGCTCGATGTC
TGTAATGGCCGAGCGGGAAACAGTTATAGACTTTACCGTCCCGTACTATGATCTAGTTGG
GATTAGTATTATGATGCAATTGCCAAGCACGCCGAGTTCGCTGTTTAAATTTTTAACCGTGCTG
GAAACGAATGTGTGGCTCTGCATTTTGGCTGCCTACTTCTTTACCAGCTTTCTGATGTGG
ATTTTTGACCGCTATAGTCCATACAGCTACCAGAATAATCGAGAGAAGTACAAGAACGAC
GACGAAAACGGGAGTTCAATATTAAAGAATGTCTGTGGTTCTGCATGACATCGTTGACA
CCGCAAGGTGGCGGAGAAGCACCCAAAAATTTGTCCGGTCGCTTAGTCGCTGCCACATGG
TGGTTGTTTGGGTAAGTAGTAGCAGTATTGATAGAAGGAAAAATATGTTTCCATTGTGTG
CCGTTTCCACCCGCGTGGTTACACGGTAGGTGTAATTATCATTGCATTTCTATCATTATGTCAT

TATTCAACAGATTTATCATCATTGCCTCGTATACGGCTAATTTGGCAGCTTTCTTGACTGTATC
CCGACTAGACACGCCGGTTGAATCGCTGGATGATTTATCAAAGCAGTACAAAATTCTGTA
TGCTCCACTGAATGGATCATCGGCTATGACGTACTTCCAGCGAATGGCTGACATTGAAGC
TAAATTCTACGAGTAAGAAGTGCCATATACCGGCGCAAGTATCTAAGGATATGTTGCTTACTTA
TTTTACTACTTTCTGGCAAACGCAGGATTTGGAAGGAGATGTCGCTCAATGACTCTCTGA
CGGCGGTTGAGCGATCGAAGCTAGCTGTCTGGGACTATCCGGTCAGCGATAAATACACAA
AGATGTGGCAAGCCATGCTTGAGGCAGGTTTACCGAACAGTCTCGAAGAAGCCGTACAGC
GCATACGAAATTCGACATCTGCCTCTGGATTCGCATTTCTGGGCGACGCAACCGACATAC
GCTACCAAGTGTTGACAAATTGCGATTTACAGATGGTTGGCGAAGAGTTTTCTCGCAAAC
CGTACGCGATCGCCGTCCAGCAAGGATCACCGCTGAAGGATCAATTTAACAATGCGTACG
TATTAATTTCATTATTCCTTTTTGCGTTTTCTAACCTACACTTCTCTTCGCTGTTTAGTATACT
GATGCTACTGAACCGACGCGAGCTGGAAAAGTTGAAAGAACAATGGTGGAAGAATGATGA
CGTACAAACAAGTGCGAAAGCCAGATGACCAGTCGGATGGCATCTCGATACAAAACAT
CGGAGGCGTATTCATCGTAATATTTGTGGGGATAGGGATGGCGTGCATTACGTTATTGTT
TGAGTTTTGGTATTACAAGTATCGAAACAACTCCAAAGTGATCGATGTTGCCGAATCAAC
GGACCAGCAACACGGTGGAACAATAGTAAAAAATGTCCGTCCCGCTGGTAAGCTTATGAA
GCAAGATTCGTTAAAAGACTCAACAAAAGGTCACAATTATCAAAACCTCCGAACACGTAC
CTTGATGCCGAATTTGAGCAAGTTTCAACCTCGCTTCTAAAGTAATGCGTTTTTGACTGC
GCTTTGGAAGTACAACGAAACATACTAATGCGTCGTGAGATTTACGAGCATTAGTACGA
CTACGGATAGTTATGAACTGTATTTTTTTAATTTTAACGTATGATATGAATTTAATGGGGATAT
ATATGTAACTATCTTTACCAATCTTATATAACGATTTGTGCATTTTGAGCAATTCTTTGAATTA
TGCAACAAAACGCTAAAAATTAAATGAAA.

