## [Editor Report]

A combination of methods, including a new method for tagging genes, demonstrates that the chemosensory co-receptors of *Drosophila melanogaster* (Orco, IR8a, IR25a, IR76b) are expressed widely and highly overlapping. These findings challenge a long-standing dogma in the field and suggest that different types of receptors, that is, olfactory and ionotropic receptors, can be co-expressed in the same chemosensory neuron. Moreover, optogenetics and single sensillum recordings provide evidence that IR25a co-receptor might modulate the activity of typical Orco-dependent olfactory sensory neurons. The authors also provide evidence that this co-expression is conserved by examining two other fly species.

---

## [Decision Letter]

**Decision letter after peer review:**

Thank you for submitting your article "Chemoreceptor Co-Expression in *Drosophila* Olfactory Neurons" for consideration by *eLife*. Your article has been reviewed by 2 peer reviewers, one of whom is a member of our Board of Reviewing Editors, and the evaluation has been overseen by a Reviewing Editor and K VijayRaghavan as the Senior Editor. The reviewers have opted to remain anonymous.

Essential revisions:

Both reviewers support publication of this study. While they are by and large satisfied with the first part of the study (description of co-expression etc.), they have indicated that the second part (functional analysis) could be somewhat improved.

1. Please address all major points pertaining primarily but not exclusively to the functional significance of your findings.

2. While additional experiments are not obligatory, we recommend the inclusion of additional controls or alternative examples (i.e. traces) as indicated in the detailed reviews below.

*Reviewer #1 (Recommendations for the authors):*

At this point, the observation is very interesting and should certainly be reported in *eLife*. Nevertheless, the manuscript could be improved by addressing these more general points as this might help to interpret and understand in how far these findings should change our view of the olfactory system and receptor function. If this is observed in other insect species (e.g. by sequencing), this would suggest an evolutionary conserved and hence likely important feature of the chemosensory system.

– I would like to invite you to speculate a bit more about what your observation might mean and in how far it should change how we think about chemosensation.

– I do appreciate that cross-species analysis is challenging, but it would make your finding a lot more impactful.

– It would be great to find a way to test the importance of this co-expression in a behavioral paradigm (e.g. CO2 avoidance vs. attraction).

*Reviewer #2 (Recommendations for the authors):*

Line 47: please add "(Ebrahim et al., 2015" to "Kondoh et al., 2016; Papes et al., 2010).

Figure 4D: the red rectangle hides the traces. It would be nicer if it is lighter in color.

Figure 6 F: It would be better if the authors replace the up trace with a better trace.

Figure 2—figure supplement 2A: the example trace from the wildtype looks strange. I wonder if the authors could replace it with a better trace.

Figure 4—figure supplement 1B, C: Again, the red rectangles hide the traces. The author should use Ir25a-T2A-QF2>QUAS-CsChrimson not feed all-trans retinal as a control.

Line 1603-1604: It would be better if the authors mention which odorants they used exactly to identify each of ab2 and pb1-pb3 sensilla.

Key resources Table: It would be better if the authors add the catalog # of the chemicals in addition to the CAS #.

---

## [Author Response]

Essential revisions:Both reviewers support publication of this study. While they are by and large satisfied with the first part of the study (description of co-expression etc.), they have indicated that the second part (functional analysis) could be somewhat improved.1. Please address all major points pertaining primarily but not exclusively to the functional significance of your findings.2. While additional experiments are not obligatory, we recommend the inclusion of additional controls or alternative examples (i.e. traces) as indicated in the detailed reviews below.

We thank the two reviewers for their support of our work, and for their helpful and insightful comments on the manuscript. In this revised version, we updated the manuscript text and figures to address specific concerns of the reviewers. We have also included new experiments aimed at broadening the significance of our work (as requested by Reviewer 1). We performed fluorescence in situ analyses of *Orco* and *Ir25a* expression in the maxillary palps of *Drosophila sechellia*, as well as in situ analyses for the palps and antennae of the malaria mosquito *Anopheles coluzzii*. We find that *Orco* and *Ir25a* co-expression occur broadly in these insects, addressing a reviewer concern that chemoreceptor co-expression might have been unique to a lab strain of *Drosophila melanogaster*. We provide a point-by-point response to the reviewers’ comments in blue below.

Reviewer #1 (Recommendations for the authors):At this point, the observation is very interesting and should certainly be reported in eLife. Nevertheless, the manuscript could be improved by addressing these more general points as this might help to interpret and understand in how far these findings should change our view of the olfactory system and receptor function. If this is observed in other insect species (e.g. by sequencing), this would suggest an evolutionary conserved and hence likely important feature of the chemosensory system.

We thank the reviewer for their strong support of this work and for its publication in *eLife*. To address this reviewer’s comment regarding the broader impact of this work beyond *Drosophila melanogaster*, we have performed a series of new experiments shown in a new Figure (Figure 7). We performed fluorescence in situ hybridization experiments in the palps of *Drosophila sechellia*. As we found for *Drosophila melanogaster*, we also detect broad co-expression of *Orco* and *Ir25a* in this other fly species. To extend these results even further, we performed fluorescence in situ hybridization experiments in the antennae and palps of the malaria mosquito, *Anopheles coluzzii*. Similarly, we found co-expression of *Orco* and *Ir25a* in both of the mosquito olfactory organs. These findings suggest that *Orco* and *Ir25a* co-receptor expression likely occurs across many insect species and is not unique to an inbred lab strain of *Drosophila melanogaster*.

The new Figure 7 corresponds to a new section of the results at revised lines 509-526:

“Co-Receptor Co-Expression in Other Insect Olfactory Organs. To determine if co-receptor co-expression might exist in other insects besides *Drosophila melanogaster*, we used RNA in situ hybridization to examine expression of *Orco* and *Ir25a* orthologues in the fly *Drosophila sechellia* and in the mosquito *Anopheles coluzzii* (Figure 7). *Drosophila melanogaster* and *Drosophila sechellia* diverged approximately 5 million years ago (Hahn et al., 2007), while the *Drosophila* and *Anopheles* lineages diverged nearly 260 million years ago (Gaunt and Miles, 2002) (Figure 7A). Because co-receptor sequences are highly conserved, we could use our *D. mel. Orco* and *Ir25a* in situ probes (Figure 7B) to examine the expression of these genes in the maxillary palps of *Drosophila sechellia*. We found widespread co-expression of *Orco* and *Ir25a* (63% of all cells were double labeled), consistent with our findings in *Drosophila melanogaster* (Figure 7C). For *Anopheles coluzzii* mosquitoes, we designed *Anopheles*-specific *Orco* and *Ir25a* probes, and examined co-receptor co-expression in antennae (Figure 7D) and maxillary palps (Figure 7E). We observed broad co-expression of *AcOrco* and *AcIr25a* in the maxillary palp capitate peg sensilla (47% of all cells were double labeled), and narrower co-expression in the antennae (25% double labeled). Co-expression results for all tissues examined are summarized in Figure 7F, and cell counts can be found in Table 5. These results suggest that *Orco* and *Ir25a* co-receptor co-expression extends to other Drosophilid species as well as mosquitoes (see also (Ye et al., 2021; Younger et al., 2020)).”

– I would like to invite you to speculate a bit more about what your observation might mean and in how far it should change how we think about chemosensation.

As summarized above, we have expanded the Discussion section to raise additional biological ramifications of chemosensory co-receptor expression. Future experiments will allow such speculations to be refined, and possibly identify biological consequences not considered at this time.

– I do appreciate that cross-species analysis is challenging, but it would make your finding a lot more impactful.

As mentioned above, we performed a set of new experiments resulting in a new figure (Figure 7) which demonstrates that *Orco* and *Ir25a* co-expression occurs in insect species beyond *Drosophila melanogaster*. Our new experiments are further supported by the findings of the Vosshall lab (Younger, Herre et al., bioRxiv, 2020) that present compelling evidence that co-receptor co-expression similarly occurs in the mosquito *Aedes aegypti*. Furthermore, recent work in *Anopheles coluzzii* mosquitoes from the Zwiebel lab (Ye, Liu et al., bioRxiv, 2021) demonstrated co-expression of Ir76b-expressing neurons with *Orco*. Altogether, our data and these independent studies support co-expression of chemosensory receptors across multiple insect species.

– It would be great to find a way to test the importance of this co-expression in a behavioral paradigm (e.g. CO2 avoidance vs. attraction).

We agree that testing the behavioral implications for receptor co-expression is a fascinating and important question. Rigorously addressing this question with the type of experiments suggested by the reviewer requires generating complex genotypes (for example, *Gr21a-Gal4* driving *UAS-Ir25a* in a homozygous *Ir25a^-^; Gr63a^-^* double mutant background) and establishing behavioral apparatuses and assays not currently available in the lab (such as those modeled by Van Breugel et al., *Nature* 2018). In addition, we feel the primary findings of our work- the broad existence of co-receptor co-expression- would not be significantly strengthened by these additional experiments. So while we share the reviewer’s fascination to extend these findings into behavioral paradigms, we feel this would be beyond the scope of the current work.

Reviewer #2 (Recommendations for the authors):Line 47: please add "(Ebrahim et al., 2015" to "Kondoh et al., 2016; Papes et al., 2010).

Thank you for pointing this out. The reference has been added to the revised manuscript.

Figure 4D: the red rectangle hides the traces. It would be nicer if it is lighter in color.

Thank you for the suggestion. We have made the red rectangle lighter in this figure and the corresponding supplement.

Figure 6 F: It would be better if the authors replace the up trace with a better trace.

We have made the suggested change to Figure 6F.

Figure 2—figure supplement 2A: the example trace from the wildtype looks strange. I wonder if the authors could replace it with a better trace.

We have replaced the trace with another example in Figure 2—figure supplement 2A.

Figure 4—figure supplement 1B, C: Again, the red rectangles hide the traces. The author should use Ir25a-T2A-QF2>QUAS-CsChrimson not feed all-trans retinal as a control.

We have lightened the color of the rectangle. That is indeed the control genotype that we used, but we have made that more explicit in the figure legend and text for clarity.

Line 1603-1604: It would be better if the authors mention which odorants they used exactly to identify each of ab2 and pb1-pb3 sensilla.

For optogenetic ab2 experiments, the odorants are listed (lines 1796-1798). These same odorants were also used for non-optogenetic SSR experiments in ab2 (this information has been added in lines 1714-1716). Briefly, we distinguished the ‘large’ antennal basiconic sensilla in the following way:

– Ab1 can be ruled out due to having four spiking neurons

– Ab2 and ab3 can be distinguished using ethyl acetate (EA) and pentyl acetate (PA)

– Ab2 has high EA and low PA responses

– Ab3 has low EA and high PA responses

Similarly, we used EA and PA responses to distinguish the different palp basiconic sensilla (information added in lines 1709-1716):

– Pb1A has high EA and high PA responses

– Pb2A has low EA and low PA responses

– Pb3A has no EA and high PA responses

This is now more clearly describe in the modified methods on revised lines 1708-1716:

“Basiconic Single Sensillum Recordings

Flies were immobilized and visualized as previously described (Lin and Potter, 2015). Basiconic sensilla were identified either using fluorescent-guided SSR (for ab3 sensilla) or using the strength of the A and B neuron responses to the reference odorants 1% ethyl acetate (EA) (Σ #270989) and 1% pentyl acetate (PA) (Σ #109584) (for ab2 and pb1-3 sensilla) (de Bruyne et al., 1999; Lin and Potter, 2015). For example, pb1A has strong responses to both odorants, while pb3A does not respond to EA. Similarly, ab2 sensilla were distinguished from ab3 based on the A neuron responses: ab2A responds strongly to EA and weakly to PA, while ab3A neurons have the reverse response (weak EA and strong PA response).”

Key resources Table: It would be better if the authors add the catalog # of the chemicals in addition to the CAS #.

Thank you for the suggestion. This information has been added to the Key Resources Table.